# Cytosolic Ptbp2 modulates axon growth in motoneurons through axonal localization and translation of *Hnrnpr*

Saeede Salehi [1], Abdolhossein Zare [1], Gianluca Prezza[1,6], Jakob Bader [2], Cornelius Schneider[3], Utz Fischer [3], Felix Meissner [2,4,7], Matthias Mann [2,5], Michael Briese [1] ✉ & Michael Sendtner [1] ✉

The neuronal RNA-binding protein Ptbp2 regulates neuronal differentiation by modulating alternative splicing programs in the nucleus. Such programs contribute to axonogenesis by adjusting the levels of protein isoforms involved in axon growth and branching. While its functions in alternative splicing have been described in detail, cytosolic roles of Ptbp2 for axon growth have remained elusive. Here, we show that Ptbp2 is located in the cytosol including axons and growth cones of motoneurons, and that depletion of cytosolic Ptbp2 affects axon growth. We identify Ptbp2 as a major interactor of the 3' UTR of *Hnrnpr* mRNA encoding the RNA-binding protein hnRNP R. Axonal localization of *Hnrnpr* mRNA and local synthesis of hnRNP R protein are strongly reduced when Ptbp2 is depleted, leading to defective axon growth. Ptbp2 regulates hnRNP R translation by mediating the association of *Hnrnpr* with ribosomes in a manner dependent on the translation factor eIF5A2. Our data thus suggest a mechanism whereby cytosolic Ptbp2 modulates axon growth by fine-tuning the mRNA transport and local synthesis of an RNA-binding protein.

Axon growth is a fundamental process during neuronal development that is regulated by both extrinsic guidance cues and intrinsic signaling cascades[1]. In motoneurons, such mechanisms ensure that axons navigate over often-extensive distances to reach their target muscle[2]. A multitude of post-transcriptional RNA processing mechanisms such as alternative splicing, polyadenylation, and subcellular localization contribute to axon growth[3]. These processes are regulated by RNA-binding proteins (RBPs), which interact with their target RNAs in a sequence-dependent manner, thereby steering their processing and fate[4].

Polypyrimidine tract-binding protein 2 (PTBP2), also known as neuronal PTB, has been identified as a master regulator of neuronal differentiation and axonogenesis[5]. In developing mice, Ptbp2 is highly expressed in post-mitotic neurons in the nervous system. Expression starts around embryonic day 12 when most neurons are postmitotic, at the same time when its homolog Ptbp1 is downregulated[6–9]. This switch from Ptbp1 to Ptbp2 expression occurs also during differentiation of cultured embryonic stem cells into neurons[8]. The mechanism underlying this switch involves Ptbp1-dependent regulation of the alternative splicing of the *Ptbp2* transcript[6]. In neuronal progenitors, Ptbp1 induces skipping of exon 10 of *Ptbp2*, leading to its non-sense mediated decay. Downregulation of Ptbp1 during differentiation facilitates *Ptbp2* exon 10 inclusion and production of Ptbp2 protein.

[1]Institute of Clinical Neurobiology, University Hospital Wuerzburg, Wuerzburg, Germany. [2]Department of Proteomics and Signal Transduction, Max Planck Institute of Biochemistry, Martinsried, Germany. [3]Department of Biochemistry, Theodor Boveri Institute, University of Wuerzburg, Wuerzburg, Germany. [4]Experimental Systems Immunology, Max Planck Institute of Biochemistry, Martinsried, Germany. [5]NNF Center for Protein Research, Faculty of Health Sciences, University of Copenhagen, Copenhagen, Denmark. [6]Present address: Helmholtz Institute for RNA-based Infection Research (HIRI), Helmholtz Centre for Infection Research (HZI), Wuerzburg, Germany. [7]Present address: Institute of Innate Immunity, Department of Systems Immunology and Proteomics, Medical Faculty, University of Bonn, Bonn, Germany. ✉e-mail: Briese_M@ukw.de; Sendtner_M@ukw.de

Several lines of evidence indicate that Ptbp2 regulates neuronal development. Mice carrying a full *Ptbp2* null allele die at birth due to respiratory failure[10,11]. While the brains of *Ptbp2* null mice appear macroscopically normal, they show defects in neuronal progenitor proliferation as well as in neuronal differentiation and maturation[11]. *Ptbp2* knockout mice with Emx1-Cre-mediated conditional ablation of Ptbp2 in projecting neurons of the forebrain exhibit a series of neurodevelopmental abnormalities such as cortical atrophy appearing at P5 leading to postnatal lethality around P18-P21[10]. Cortical neurons cultured from *Ptbp2* knockout mice show impaired axon growth and branching defects[12]. These defects are accompanied by dysregulated alternative splicing of axonogenesis genes such as *Shootin1*[12]. Ptbp2 itself regulates alternative splicing by binding to CU-rich motifs within and around adult-specific exons, thereby repressing their splicing[11]. By doing so, Ptbp2 sustains axon growth by preventing premature usage of alternative splicing programs that support axon specification and maturation.

While functions of Ptbp2 in alternative splicing have been described in detail, cytosolic roles of Ptbp2 linked to axon growth have remained unclear. Here, we provide evidence that cytosolic Ptbp2 regulates axon growth of motoneurons. In addition to its nuclear localization, we detected Ptbp2 in the cytosol, including axons and growth cones, of cultured embryonic motoneurons. We show that Ptbp2 regulates axon growth of motoneurons through modulating the synthesis of the neuronal RBP hnRNP R. Proteomic analysis revealed that Ptbp2 is a major interactor of the 3′ UTR of *Hnrnpr*. We found that Ptbp2 binding to the *Hnrnpr* transcript occurs in the cytosol and axons of motoneurons and that transport of *Hnrnpr* mRNA into axons is dependent on Ptbp2. Additionally, we found that Ptbp2 promotes the translation of *Hnrnpr* by regulating its association with ribosomes. Accordingly, local synthesis of hnRNP R in axons was impaired upon depletion of Ptbp2. We present evidence that the function of Ptbp2 in *Hnrnpr* translation requires the translation factor eIF5A2, and that this mechanism is disturbed in Ptbp2-deficient axons. Importantly, over-expression of hnRNP R in Ptbp2-deficient motoneurons rescued their axon growth whereas overexpression of Ptbp2 failed to normalize the axon elongation defect of motoneurons cultured from a *Hnrnpr* knockout mouse. Together, our findings identify axonally produced hnRNP R as a mediator of Ptbp2 functions in axon growth.

## Results

### Ptbp2 is required for axon growth in motoneurons

First, we investigated the level of Ptbp2 expression in the developing mouse spinal cord from embryonic day E13 up to adult (6–12 weeks) stage. Ptbp2 levels were highest at E13 and continuously declined at later time points (Fig. 1a). To examine whether Ptbp2 mediates axon growth in motoneurons, we transduced cultured embryonic mouse motoneurons with lentiviruses expressing a short hairpin RNA (shRNA) directed against the *Ptbp2* 3′ UTR[13] (sh1*Ptbp2*) or against exon 4 (sh2*Ptbp2*) (Fig. 1b, c, Supplementary Fig. 1a). Following Ptbp2 depletion, cell survival was mildly reduced on day in vitro (DIV) 6 in agreement with previous studies on cortical Ptbp2-deficient neurons[10,12] (Fig. 1d). Also at this time point, a reduction in axon lengths upon Ptbp2 depletion became apparent (Fig. 1e, f and Supplementary Fig. 1b). Likewise, loss of Ptbp2 impaired growth cone size (Fig. 1g, h). To confirm the specificity of the *Ptbp2* knockdown phenotype, we performed a rescue experiment by co-expressing an EGFP-Ptbp2 fusion protein together with sh1*Ptbp2*. Both axon length and growth cone size were restored to normal levels following expression of EGFP-Ptbp2 in Ptbp2-deficient motoneurons (Fig. 1e–h). To investigate growth cone maturation of *Ptbp2* knockdown motoneurons, we performed immunostaining for the early presynaptic marker protein Synapsin 1 and the late synaptic marker Piccolo (Pclo). Interestingly, we observed reduction of Pclo but not Synapsin 1 in Ptbp2-depleted growth cones. Pclo is involved in presynaptic actin assembly at late

stages of development[14,15] whereas Synapsin 1 is a marker of synaptic vesicles that is already found in axon terminals of motoneurons at very early stages of development[16]. Thus, Ptbp2 appears to be involved in the late steps of axon terminal maturation (Fig. 1i–l).

### Cytosolic Ptbp2 is involved in axon growth

To investigate the subcellular distribution of Ptbp2, we fractionated motoneurons and motoneuron-like NSC-34 cells[17] into a cytosolic (Cyt) fraction, a nuclear soluble and organellar fraction (Nuc+org) and a nuclear insoluble chromatin (Chr) fraction. In agreement with its nuclear functions in pre-mRNA processing, Ptbp2 was abundant in the nuclear fractions of primary motoneurons and NSC-34 cells (Fig. 2a, b and Supplementary Fig. 1c,d). Nevertheless, Ptbp2 was also readily detectable in the cytosolic fraction of these cells. Consistent with these results, Ptbp2 immunofluorescence staining of motoneurons revealed Ptbp2 immunoreactivity in the nucleus as well as in the cytoplasm (Fig. 2c). Additionally, Ptbp2 immunoreactivity was detectable in axons and growth cones (Fig. 2c). To validate the specificity of the antibody, we immunostained motoneurons expressing EGFP-Ptbp2 with anti-Ptbp2 and anti-EGFP (Supplementary Fig. 1e). We observed that, in axons, the distribution of Ptbp2 punctae correlated with EGFP punctae.

To test whether the effect of *Ptbp2* knockdown on axon growth is mediated by the cytosolic function of Ptbp2, we prepared a lentiviral *Ptbp2* knockdown construct that co-expresses EGFP-tagged Ptbp2 harboring a deletion of its nuclear localization sequence (NLS) (EGFP-Ptbp2-ΔNLS) (Fig. 2d, e). Ptbp2 contains a NLS composed of two basic stretches (KR and KKFK) at its N-terminal end[18]. Deletion of these stretches has been shown to abolish the nuclear localization of Ptbp2[19]. We confirmed the cytosolic localization of EGFP-Ptbp2-ΔNLS in transduced motoneurons by immunostaining (Fig. 2f). Expression of EGFP-Ptbp2-ΔNLS in *Ptbp2* knockdown motoneurons could rescue the defect in axon growth indicating that cytosolic but not nuclear Ptbp2 functions mediate its role in axon growth (Fig. 2g–j).

### Ptbp2 is a major interactor of the *Hnrnpr* 3′ UTR

Previous studies have shown that the RBP heterogeneous nuclear ribonucleoprotein R (hnRNP R) is enriched in the nervous system and regulates axon growth[20]. In developing motoneurons, hnRNP R is located in the nuclei as well as in axons and growth cones[20]. Depletion of hnRNP R by morpholinos in zebrafish or by shRNA-mediated knockdown in cultured motoneurons results in axon elongation defects accompanied by axonal transcriptome alterations[20,21]. The *Hnrnpr* mRNA itself is localized in axons of cultured motoneurons suggesting that it is transported along axons for local translation[22]. Thus, hnRNP R is a crucial RBP for axon growth through axonal translocation of transcripts[21].

Given the roles of hnRNP R in axon growth, we investigated the possibility that Ptbp2 regulates the cytosolic processing of the *Hnrnpr* mRNA by interacting with its 3′ UTR. For this purpose, we sought to identify proteins interacting with the *Hnrnpr* 3′ UTR in an unbiased manner through mass spectrometry (MS)-based proteomics. To do so, we first assessed the alternative polyadenylation of *Hnrnpr* by 3′ rapid amplification of cDNA ends to determine 3′ UTR usage. We detected *Hnrnpr* transcript isoforms with three alternative 3′ UTRs of different lengths (short, medium, long; Supplementary Fig. 2a, b). Across different mouse tissues and in NSC-34 cells, *Hnrnpr* transcripts containing the short 3′ UTR were more abundant than transcripts containing the medium or long 3′ UTR (Supplementary Fig. 2c). In the mouse spinal cord, *Hnrnpr* transcripts containing the short 3′ UTR were expressed in a developmentally regulated manner with the highest levels at early developmental time points (Supplementary Fig. 2d). In NSC-34 cells, *Hnrnpr* transcripts containing the short 3′ UTR were more stable compared to transcripts with the medium and long 3′ UTR (Supplementary Fig. 2e). Following immunopurification of ribosomes

using the Y10b antibody and quantification of *Hnrnpr* transcript co-purification by qPCR, the vast majority of *Hnrnpr* transcripts bound to ribosomes harbored the short 3′ UTR indicating that hnRNP R protein is almost exclusively produced from this mRNA isoform (Supplementary Fig. 2f–h). To substantiate this finding, we quantified the levels of *Hnrnpr* isoforms in subcellular fractions of NSC-34 cells (Supplementary Fig. 2i). In agreement with our Y10b immunoprecipitation results, *Hnrnpr* transcripts containing the short 3′ UTR were more prevalent in the cytosolic fraction whereas isoforms harboring the medium or long 3′ UTR were confined to the nucleus. Together, these data suggest that hnRNP R protein synthesis and cytosolic *Hnrnpr* transcript localization are regulated by the short 3′ UTR.

To obtain insights into the post-transcriptional regulation of hnRNP R, we investigated the protein interactome associated with the short *Hnrnpr* 3′ UTR in NSC-34 cells with MS-based proteomics. In NSC-34 cells, both Ptbp1 and Ptbp2 are expressed (Fig. 3a). We generated a biotinylated RNA fragment encompassing the short 3′ UTR of *Hnrnpr* by in vitro transcription and used it as bait to isolate bound proteins from NSC-34 cell lysate (Fig. 3b). As a control, we used a biotinylated RNA corresponding to the antisense sequence of the short 3′ UTR. Differential enrichment analysis identified several proteins that specifically interacted with the short *Hnrnpr* 3′ UTR RNA but not its antisense control (Fig. 3c and Supplementary Data 1). According to the proteomics data, Ptbp2 was the most strongly enriched interaction partner. To validate the interaction, we performed immunoblotting for Ptbp2 following pulldown with the short *Hnrnpr* 3′ UTR RNA or its antisense control from NSC-34 lysate. In line with the proteomics results, Ptbp2 was selectively associated with the *Hnrnpr* 3′ UTR sense but not its antisense sequence (Fig. 3d, e). To show that Ptbp2 binds to the endogenous *Hnrnpr* mRNA, we performed RNA immunoprecipitation using

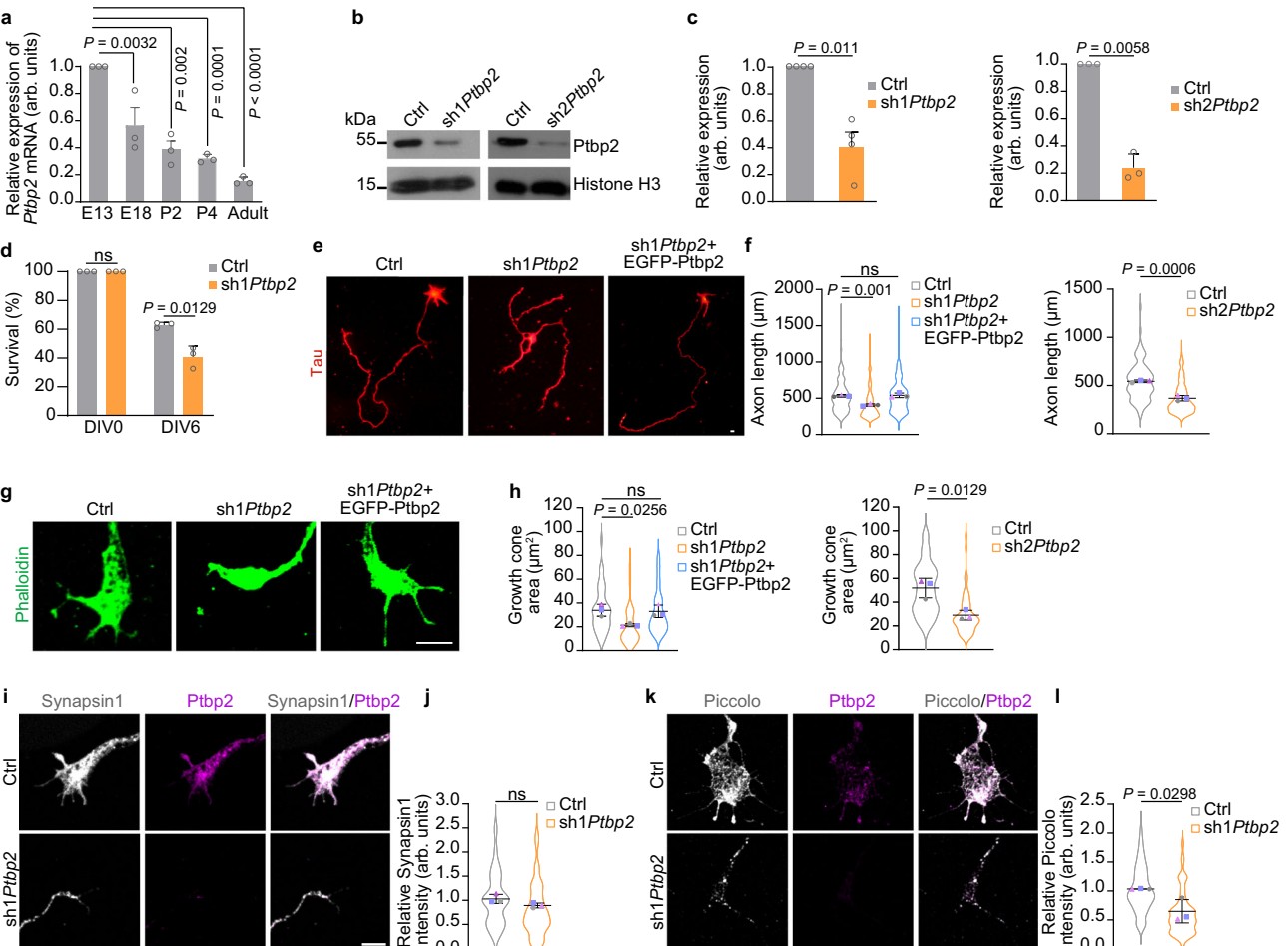

**Fig. 1 | Ptbp2 modulates axon growth in cultured motoneurons. a** *Ptbp2* mRNA expression detected by qPCR in the spinal cord of mice at different time points. *Nefl* mRNA was used for normalization. One-way ANOVA with Dunnett's multiple comparisons test. Data are mean ± standard deviation (s.d.) of $n = 3$ biological replicates. **b** Immunoblot analysis of Ptbp2 in control and Ptbp2-depleted motoneurons at DIV 6. Histone H3 was used as a loading control. **c** Quantitative analysis of Ptbp2 immunoblots as shown in (**b**). Two-tailed one-sample *t* test. Data are mean ± s.d. of $n = 4$ (sh1*Ptbp2*) and $n = 3$ (sh2*Ptbp2*) biological replicates. **d** Quantification of motoneuron survival at DIV 6. Two-way ANOVA with Sidak's multiple comparisons tests. ns, not significant ($P > 0.05$). Data are mean ± s.d. of $n = 3$ biological replicates. **e** Representative images of motoneurons transduced with a control lentivirus (Ctrl), a lentivirus expressing a shRNA targeting *Ptbp2* (sh1*Ptbp2*) and rescued motoneurons transduced with a lentivirus co-expressing sh1*Ptbp2* and knockdown-resistant EGFP-Ptbp2 (sh1*Ptbp2* + EGFP-Ptbp2) at DIV 6. Motoneurons were stained against tau as a specific axonal marker. Scale bar, 10 μm.

**f** SuperPlots representing the quantification of axon lengths. One-way ANOVA with Tukey's multiple comparisons test or unpaired two-tailed Student's *t* test. Data are mean ± s.d. of $n = 3$ biological replicates. Representative images (**g**) and SuperPlots (**h**) of growth cone sizes of control, *Ptbp2* knockdown and rescued motoneurons at DIV 6. Motoneurons were stained with phalloidin. Scale bar, 5 μm. One-way ANOVA with Tukey's multiple comparisons test or unpaired two-tailed Student's *t* test. Data are mean ± s.d. of $n = 3$ biological replicates. **i** Synapsin1 immunostaining in growth cones of control and Ptbp2-depleted motoneurons at DIV 6. Scale bar, 5 μm. **j** SuperPlots of Synapsin1 immunosignals in growth cones. Unpaired two-tailed Student's *t* test. Data are mean ± s.d. of $n = 3$ biological replicates. **k** Immunofluorescence imaging of Piccolo in growth cones of control and *Ptbp2* knockdown motoneurons. **l** SuperPlots of Piccolo immunosignals in growth cones. Unpaired two-tailed Student's *t* test. Data are mean ± s.d. of $n = 3$ biological replicates. Source data are provided as a Source Data file.

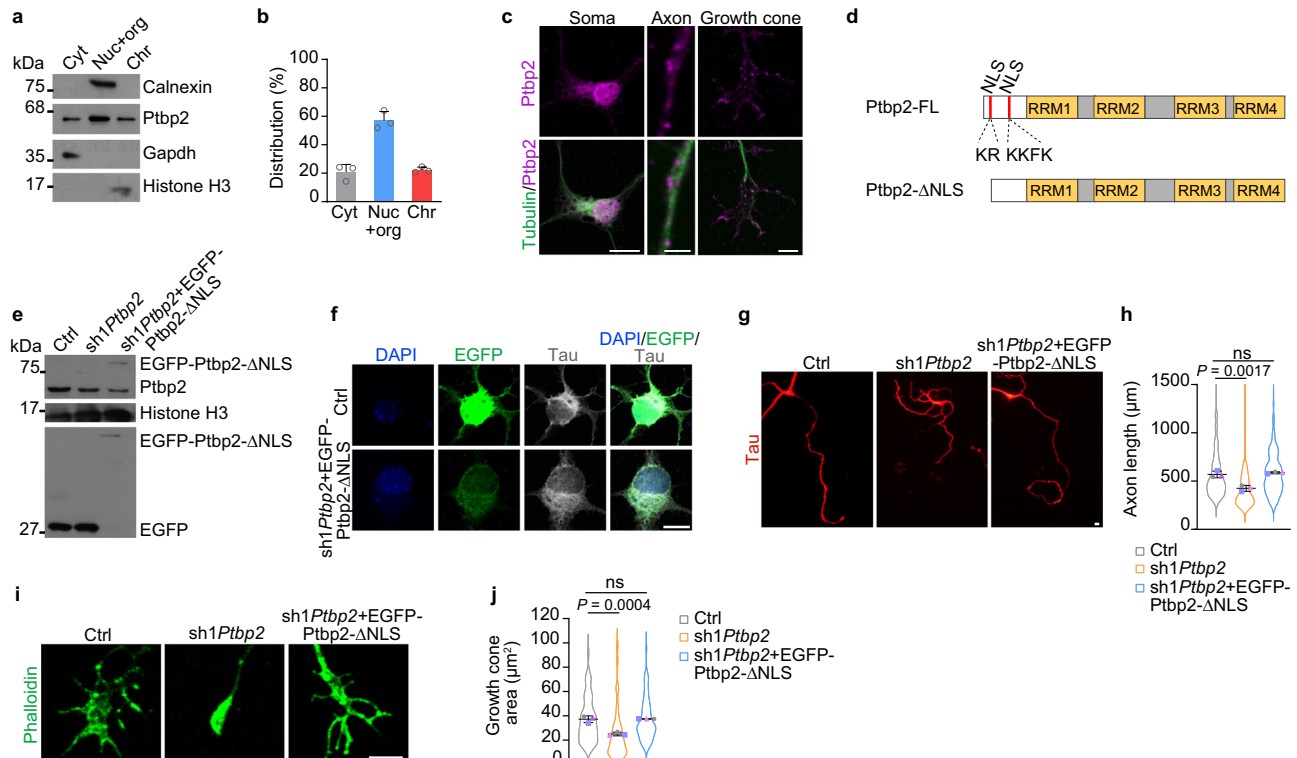

**Fig. 2 | Cytosolic Ptbp2 is involved in axon growth. a** Representative immunoblot of Ptbp2 levels in subcellular fractions of motoneurons. Cyt, cytosol; Nuc+org, nuclear soluble proteins and organelles; Chr, chromatin-associated proteins. **b** Quantification of Western blot signals in (**a**). Data are mean ± s.d. of $n = 3$ biological replicates. **c** Representative immunofluorescence images showing the distribution of Ptbp2 in the soma, axon and growth cone of a cultured motoneuron at DIV 6. Scale bars, 10 μm (soma), 2 μm (axon) and 5 μm (growth cone). The images are representative of at least three biological replicates. **d** Schematic representation of the constructs expressing full-length (FL) Ptbp2, or Ptbp2 harboring a deletion of the nuclear localization signal (NLS). Red bars represent the NLS located at the N-terminus of Ptbp2. **e** Immunoblot analysis of motoneurons transduced with control lentivirus, or lentiviruses expressing sh1*Ptbp2* or an EGFP-Ptbp2 NLS deletion mutant together with sh1*Ptbp2* (sh1*Ptbp2* + EGFP-Ptbp2-ΔNLS). The

immunoblot is representative of three biological replicates. **f** Immunofluorescence imaging of EGFP in motoneurons. Scale bars, 10 μm. The images are representative of three biological replicates. **g** Representative images of control, Ptbp2-depleted motoneurons and rescued motoneurons transduced with a lentivirus expressing sh1*Ptbp2* + EGFP-Ptbp2-ΔNLS at DIV 6. Motoneurons were immunostained against tau. Scale bar, 10 μm. **h** SuperPlots representing the quantification of axon lengths. One-way ANOVA with Tukey's multiple comparisons test. Data are mean ± s.d. of $n = 3$ biological replicates. Representative images (**i**) and SuperPlots (**j**) of growth cone sizes of control, *Ptbp2* knockdown and rescued motoneurons at DIV 6. Motoneurons were stained with phalloidin. Scale bar, 5 μm. One-way ANOVA with Tukey's multiple comparisons test. Data are mean ± s.d. of $n = 3$ biological replicates. Source data are provided as a Source Data file.

an anti-Ptbp2 antibody to isolate bound mRNAs from NSC-34 cells lysate. We observed *Hnrnpr* mRNA co-purification with the Ptbp2 antibody but not with IgG control (Fig. 3f, g). The association between *Hnrnpr* and Ptbp2 was also detectable in motoneurons showing an even stronger enrichment compared to NSC-34 cells (Fig. 3h, i). This, most likely, is due to absence of Ptbp1 expression in differentiated postmitotic motoneurons (Fig. 3a).

We then assessed the specificity of Ptbp2 binding along the *Hnrnpr* 3′ UTR by investigating a previously published Ptbp2 HITS-CLIP dataset from the developing mouse brain[11]. This identified a cluster of Ptbp2 CLIP tags within a pyrimidine-rich region of the *Hnrnpr* 3′ UTR (Supplementary Fig. 2j). To test whether Ptbp2 binds to this sequence, we generated lentiviral constructs for expression of the EGFP coding sequence fused to either the full-length short 3′ UTR of *Hnrnpr* (*EGFP-FL*) or a 3′ UTR with a deletion of the putative Ptbp2 binding site (PBS) (*EGFP-ΔPBS*) (Fig. 3j). Following lentiviral transduction of cultured motoneurons, we immunoprecipitated Ptbp2 and assessed co-purification of *EGFP* by qPCR. We observed that co-purification of *EGFP* containing the *Hnrnpr* ΔPBS 3′ UTR was reduced compared to the FL 3′ UTR, demonstrating the specificity of the Ptbp2 binding motif within the *Hnrnpr* 3′ UTR (Fig. 3k).

To evaluate whether alternative polyadenylation of *Hnrnpr* is regulated by Ptbp2, we assessed the relative abundance of the three 3′ UTR isoforms following *Ptbp2* knockdown in NSC-34 cells and

motoneurons. The levels of the three 3′ UTR isoforms were unchanged indicating that Ptbp2 does not alter *Hnrnpr* alternative polyadenylation (Fig. 3l).

## Ptbp2 regulates transport of the *Hnrnpr* mRNA into axons of motoneurons

Our results showing that Ptbp2 is present in axons of motoneurons and that Ptbp2 associates with the 3′ UTR of *Hnrnpr* point toward the possibility that Ptbp2 regulates the axonal transport of *Hnrnpr*. To evaluate this further, we visualized the localization of *Hnrnpr* mRNA in motoneurons by fluorescent in situ hybridization (FISH) and of Ptbp2 by immunostaining. Strikingly, the localization of Ptbp2-positive punctae strongly overlapped with the FISH signal for *Hnrnpr* in the cytoplasm as well as in axons (Fig. 4a, b). The *Hnrnpr* FISH signal was absent in motoneurons cultured from *Hnrnpr*−/− mice, indicating the specificity of the *Hnrnpr* probe (Supplementary Fig. 3a). Nearly all *Hnrnpr*-positive punctae also immunostained for Ptbp2 highlighting the importance of Ptbp2 for post-transcriptional regulation of *Hnrnpr* mRNA (Fig. 4c). Vice versa, ~60% of Ptbp2-positive punctae also contained an *Hnrnpr* FISH signal showing that Ptbp2 is present in excess of *Hnrnpr*. In contrast, we did not observe any cytosolic *Hnrnpr* FISH punctae (0 out of 180 punctae) that were positive for hnRNP A2/B1, a protein that was enriched for the anti-sense 3′ UTR (Supplementary Fig. 3b). This indicates that Ptbp2 and *Hnrnpr* are components of

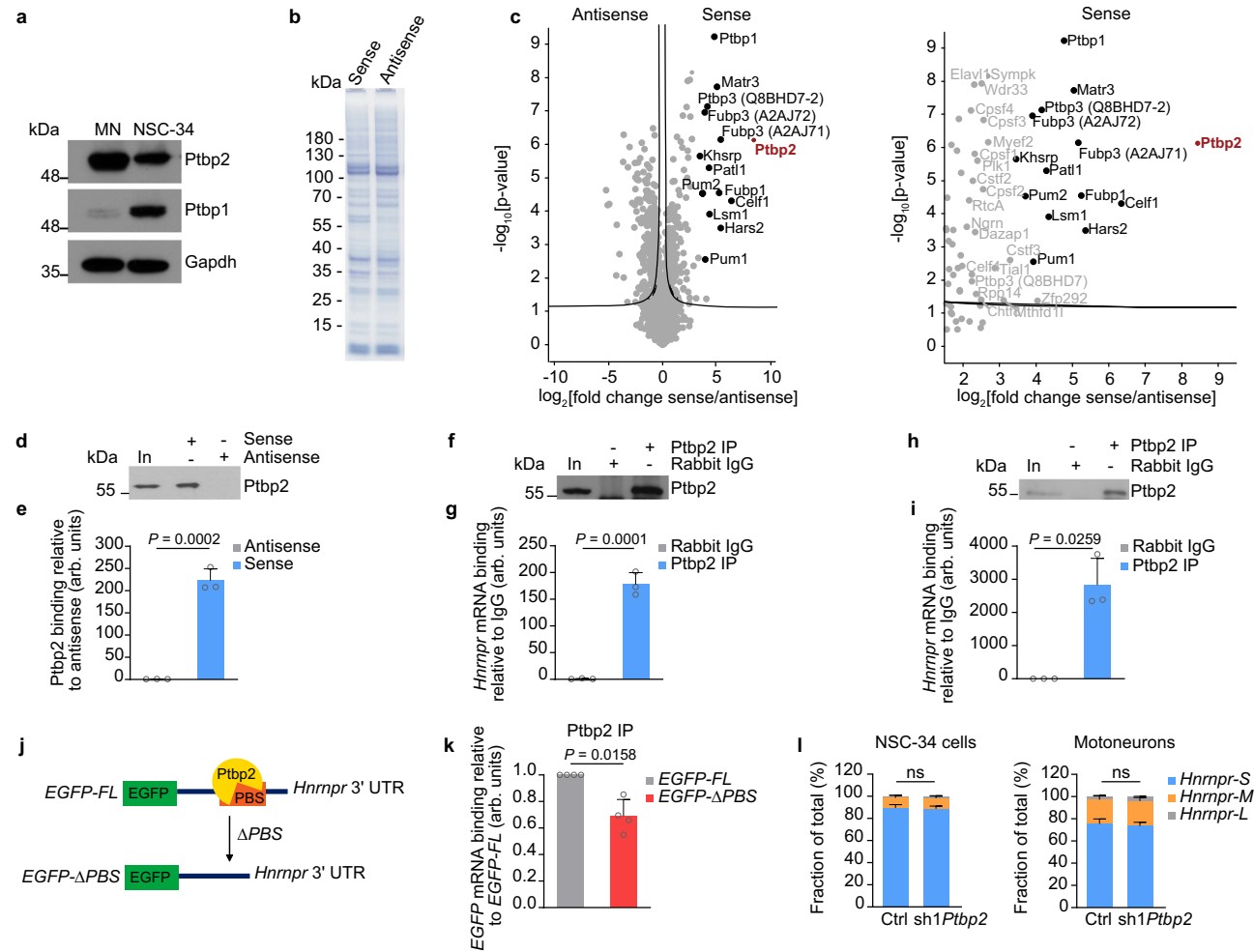

**Fig. 3 | Ptbp2 interacts with the *Hnrnpr* 3′ UTR. a** Immunoblot analysis of Ptbp2 and Ptbp1 protein levels in NSC-34 cells and mouse motoneurons (MN). Gapdh was used as loading control. The immunoblot is representative of two biological replicates. **b** Coomassie-stained SDS-PAGE gel following pulldown with biotinylated sense or antisense control *Hnrnpr* 3′ UTR RNA. The gel is representative of at least two biological replicates. **c** Volcano plots of the *Hnrnpr* 3′ UTR interactome following pulldown with sense *Hnrnpr* 3′ UTR and antisense control from NSC-34 cell lysate. Uniprot IDs in brackets specify protein isoforms in cases of multiple detected isoforms. $n = 4$ biological replicates. **d** Immunoblot analysis of Ptbp2 following pulldown with sense or antisense control *Hnrnpr* 3′ UTR RNA from NSC-34 lysate. In, input. **e** Quantification of the pulldown of Ptbp2 in (**d**). Data are shown as enrichment of Ptbp2 by pulldown with sense RNA as compared to the antisense RNA control. Unpaired two-tailed Student's *t* test. Data are mean ± s.d. of $n = 3$ biological replicates. **f** Immunoprecipitation of Ptbp2 from NSC-34 cell lysate. Immunoprecipitation with rabbit IgG antibody was used as a control. **g** qPCR

analysis of *Hnrnpr* co-immunoprecipitated with anti-Ptbp2 from NSC-34 cells. Two-tailed one-sample *t* test. Data are mean ± s.d. of $n = 3$ biological replicates. **h** Immunoblot of Ptbp2 immunoprecipitation from motoneuron lysate. **i** qPCR analysis of *Hnrnpr* co-precipitated by anti-Ptbp2 from motoneurons. Two-tailed one-sample *t* test. Data are mean ± s.d. of $n = 3$ biological replicates. **j** Schematic of the EGFP reporter constructs used to investigate the specificity of Ptbp2 binding with the *Hnrnpr* 3′ UTR. *EGFP-FL*, full length; *EGFP-ΔPBS*, deletion mutant; PBS, Ptbp2 binding site. **k** qPCR analysis of *EGFP* co-precipitated by anti-Ptbp2 from motoneurons transduced with lentiviruses for expression of *Hnrnpr* 3′ UTR *EGFP-FL* and *EGFP-ΔPBS* constructs. Two-tailed one-sample *t* test. Data are mean ± s.d. of $n = 4$ biological replicates. **l** Quantification of abundance of the three 3′ UTR isoforms of *Hnrnpr* mRNA in control and Ptbp2-depleted NSC-34 cells and motoneurons. Two-way ANOVA with Sidak's multiple comparisons tests. Data are mean ± s.d. of $n = 3$ biological replicates. Source data are provided as a Source Data file.

cytosolic mRNP particles in motoneurons. To further demonstrate the association between Ptbp2 and *Hnrnpr* mRNA in axons, we cultured motoneurons in microfluidic chambers allowing the separation of axons from the somatodendritic compartment (Fig. 4d, e). Following Ptbp2 immunoprecipitation from axonal and somatodendritic lysates, *Hnrnpr* mRNA co-purification was observed for both comportments (Fig. 4f). Consistent with this result, *Hnrnpr* mRNA was co-purified with Ptbp2 from the soluble nuclear fraction as well as from the cytosolic fraction of motoneurons (Supplementary Fig. 3c, d).

Next, we performed FISH in control and Ptbp2-depleted motoneurons and found that the axonal localization of *Hnrnpr* mRNA was strongly decreased while its presence in the cytoplasm of the soma and its nuclear levels were unchanged by Ptbp2 depletion (Fig. 4g, h). In dendrites, *Hnrnpr* was detectable only at low levels (Fig. 4h).

Overexpression of Ptbp2 could restore the axonal localization of *Hnrnpr* mRNA in motoneurons depleted of endogenous Ptbp2 (Fig. 4g, h). We also detected reduced *Hnrnpr* mRNA levels in the axonal but not somatodendritic compartment of Ptbp2-depleted motoneurons cultured in microfluidic chambers (Fig. 4i). Together, these results illustrate that Ptbp2 regulates the axonal transport of *Hnrnpr* mRNA in motoneurons.

## Ptbp2 modulates the axonal synthesis of hnRNP R

In order to assess the axonal translation of hnRNP R in motoneurons, we pulsed motoneurons with puromycin and labeled newly synthesized hnRNP R protein using the puromycylation followed by proximity ligation amplification (Puro-PLA) assay[23]. As puromycin truncates the nascent polypeptide chain following incorporation[23], we used an

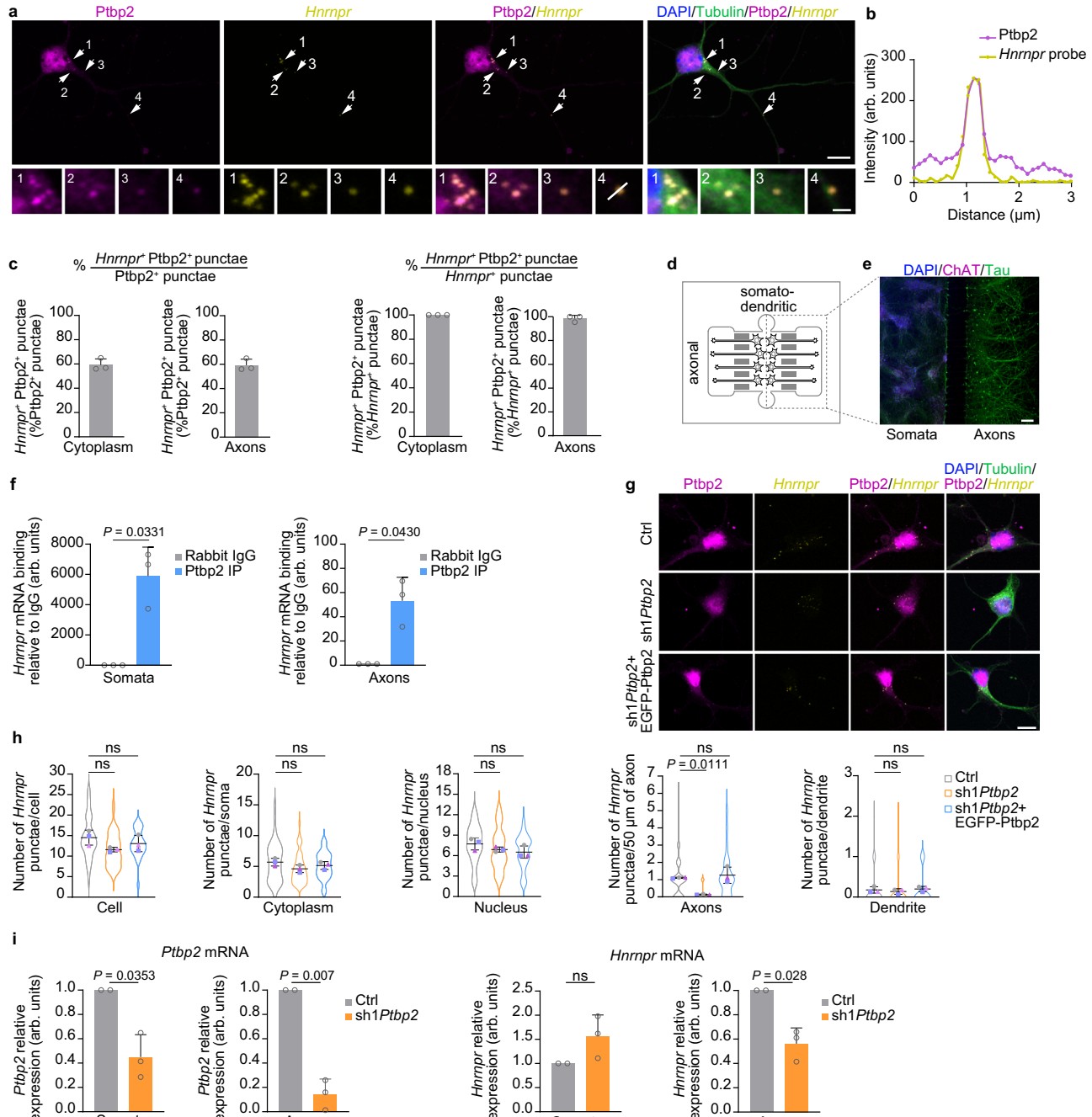

**Fig. 4 | Ptbp2 regulates axonal localization of *Hnrnpr* mRNA in motoneurons.**
**a** Representative images showing Ptbp2 immunofluorescence and *Hnrnpr* FISH in cultured motoneurons at DIV 6. An antibody against Tubulin was used for visualization of motoneuron morphology. Arrowheads indicate colocalization of Ptbp2 and *Hnrnpr* in granules. Scale bars, 10 μm and 1 μm (magnified areas).
**b** Fluorescence intensity profiles of Ptbp2 and *Hnrnpr* at the location indicated by arrow 4 in (**a**). **c** Quantification of the percentage of *Hnrnpr*⁺Ptbp2⁺ punctae in motoneurons. Data are mean ± s.d. of n = 3 biological replicates. **d** Schematic of a microfluidic chamber for compartmentalized motoneuron cultures allowing separation of the somatodendritic from axonal compartments.
**e** Immunofluorescence analysis of motoneurons cultured in a microfluidic chamber using antibodies against tau and ChAT. **f** qPCR analysis of *Hnrnpr* co-precipitated by antibody against the N-terminus of hnRNP R in combination with an antibody against puromycin (Fig. 5a). The Puro-PLA signal for hnRNP R was detectable in axons of motoneurons indicating that hnRNP R is locally translated (Fig. 5b). No Puro-PLA signal was observed when anti-Ptbp2 from somatodendritic and axonal compartments. Two-tailed one-sample *t* test. Data are mean ± s.d. of n = 3 biological replicates. **g** Representative images of Ptbp2 immunofluorescence staining and *Hnrnpr* FISH in control, Ptbp2-depleted and rescued motoneurons at DIV 6. **h** SuperPlots of the number of *Hnrnpr*-positive punctae in the whole cell, cytoplasm of the soma, nucleus, axon and dendrite of control, *Ptbp2* knockdown and rescued motoneurons. One-way ANOVA with Tukey's multiple comparisons test. Data are mean ± s.d. of n = 3 biological replicates. **i** qPCR analysis of *Ptbp2* and *Hnrnpr* mRNA from somatodendritic and axonal RNA of compartmentalized control and Ptbp2-depleted motoneurons at DIV 6. Statistical analysis was performed using a two-tailed one-sample *t* test. Data are mean ± s.d. of n = 2 (Ctrl) and n = 3 (sh1*Ptbp2*) independent experiments. Source data are provided as a Source Data file.

antibody against the N-terminus of hnRNP R in combination with an antibody against puromycin (Fig. 5a). The Puro-PLA signal for hnRNP R was detectable in axons of motoneurons indicating that hnRNP R is locally translated (Fig. 5b). No Puro-PLA signal was observed when

motoneurons were pre-incubated with the translation inhibitor cycloheximide or when puromycin was omitted showing the specificity of the labeling procedure (Fig. 5b). As additional control, we observed that the Puro-PLA signal was substantially higher when using

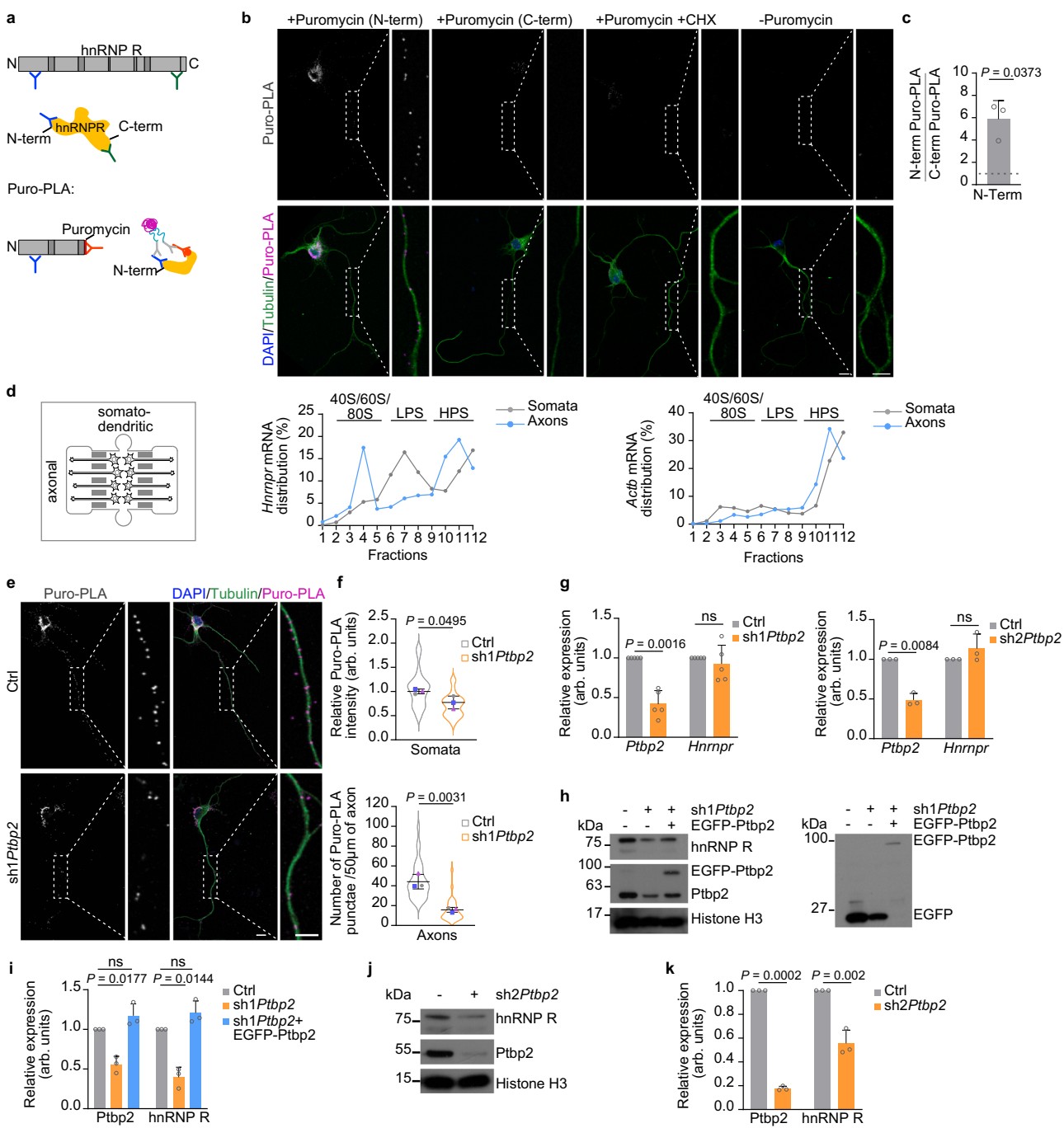

**Fig. 5 | Ptbp2 regulates the axonal translation of *Hnrnpr* mRNA in motoneurons. a** Schematic representing the binding sites for anti-hnRNP R antibodies targeted against the N- and C-terminus (blue and green, respectively). In Puro-PLA experiments, close proximity between an N-terminal hnRNP R antibody and an anti-puromycin antibody (orange) allows signal amplification. **b** Representative images of Puro-PLA signal in cultured motoneurons at DIV 7 using either N-terminal (N-term) or C-terminal (C-term) anti-hnRNP R antibodies. Labeling without puromycin or with puromycin in the presence of cycloheximide (CHX) was used as controls. Boxed regions are magnified to the right of each image. Motoneurons were immunostained for Tubulin to visualize their morphology and nuclei were labeled with DAPI. Scale bars, 10 μm and 5 μm (magnified areas). The images are representative of three biological replicates. **c** Ratio of Puro-PLA signal obtained with an N-terminal anti-hnRNP R antibody to Puro-PLA with a C-terminal antibody. Two-tailed one-sample t-test. Data are mean ± s.d. of *n* = 3 biological replicates. **d** Relative levels of *Hnrnpr* and *Actb* as a control in fractions of somatodendritic and axonal lysate subjected to sucrose density gradient ultracentrifugation. Data are

mean of *n* = 2 biological replicates. **e** Puro-PLA of hnRNP R of control and Ptbp2-depleted motoneurons. Scale bars, 10 μm and 5 μm (magnified areas). **f** Quantification of relative Puro-PLA intensity in somata and the number of Puro-PLA punctae in 50 μm of axons of control and *Ptbp2* knockdown motoneurons. Unpaired two-tailed Student's *t* test. Data are mean ± s.d. of *n* = 3 biological replicates. **g** Quantification of *Hnrnpr* and *Ptbp2* mRNA levels by qPCR in control and Ptbp2-depleted motoneurons at DIV 6. 18S rRNA was used as housekeeping transcript. Two-tailed one-sample *t* test. Data are mean ± s.d. of *n* = 5 (sh1*Ptbp2*) and *n* = 3 (sh2*Ptbp2*) biological replicates. **h** Immunoblot analysis of hnRNP R and Ptbp2 in control, Ptbp2-depleted and rescued motoneurons at DIV 6. Histone H3 was used as a loading control. **i** Quantitative analysis of Western blots as shown in (**h**) for Ptbp2 and hnRNP R. Two-tailed one-sample *t* test. Data are mean ± s.d. of *n* = 3 biological replicates. **j** Immunoblot analysis of hnRNP R and Ptbp2 in control and Ptbp2-depleted motoneurons. **k** Quantitative analysis of Western blots as shown in (**j**) for Ptbp2 and hnRNP R. Two-tailed one-sample *t* test. Data are mean ± s.d. of *n* = 3 biological replicates. Source data are provided as a Source Data file.

an antibody against the N-terminus of hnRNP R compared to an antibody against the C-terminus (Fig. 5b, c) suggesting that the hnRNP R Puro-PLA signal is due to the binding of anti-puromycin and anti-hnRNP R (N-terminus) antibodies to the same nascent polypeptide. To further validate the axonal translation of *Hnrnpr* mRNA, we cultured motoneurons in microfluidic chambers and performed polysome profiling from both the somatodendritic and axonal compartment by sucrose gradient ultracentrifugation. We observed that, in axons, the *Hnrnpr* transcript was present at higher levels in those fractions corresponding to polysomes (Fig. 5d). Thus, our data suggest that polysomes rather than monosomes mediate axonal *Hnrnpr* translation.

Next, we performed Puro-PLA for hnRNP R on control and *Ptbp2* knockdown motoneurons to evaluate whether Ptbp2 regulates the axonal translation of hnRNP R. We observed that the number of hnRNP R Puro-PLA punctae was significantly reduced in axons of *Ptbp2* knockdown motoneurons compared to controls (Fig. 5e, f). We also observed a reduction in hnRNP R Puro-PLA signal intensity in the somata of Ptbp2-depleted motoneurons, which, however, was much less pronounced compared to axons (Fig. 5e, f). To exclude the possibility that Ptbp2 deficiency induces a general defect in translation, we treated *Ptbp2* knockdown and control motoneurons with puromycin followed by immunostaining with an anti-puromycin antibody. No changes were observed in the puromycin immunosignal in the somata, proximal or distal axons of *Ptbp2* knockdown motoneurons relative to controls indicating that Ptbp2 depletion does not affect overall translation (Supplementary Fig. 4a, b). Together, our findings suggest that Ptbp2 is required for the local translation of hnRNP R in axons of motoneurons.

Having shown that axonal translation of hnRNP R is regulated by Ptbp2, we then asked whether Ptbp2 depletion also affects the steady-state protein level of hnRNP R in motoneurons. While *Hnrnpr* mRNA levels were unchanged upon knockdown of *Ptbp2* (Fig. 5g), hnRNP R protein levels were reduced by approximately half in Ptbp2-deficient motoneurons (Fig. 5h–k). In contrast to motoneurons, Ptbp2 depletion in NSC-34 cells altered hnRNP R protein levels to a much lower extent (Supplementary Fig. 4c–e) indicating that the role of Ptbp2 in regulating hnRNP R protein synthesis is particularly pronounced in postmitotic neurons. Expression of EGFP-Ptbp2 in *Ptbp2* knockdown motoneurons restored hnRNP R protein levels to normal (Fig. 5h, i), confirming the specificity of Ptbp2's role in hnRNP R translation.

## Ptbp2 controls translation of *Hnrnpr* by regulating its association with ribosomes

To investigate the underlying molecular mechanism whereby Ptbp2 regulates hnRNP R protein translation in motoneurons, we investigated whether Ptbp2 depletion affects the association of *Hnrnpr* mRNA with ribosomes. To do so, we immunoprecipitated ribosomes using the Y10b antibody from control and *Ptbp2* knockdown motoneurons and quantified co-purification of *Hnrnpr* by qPCR. We observed that Ptbp2 depletion significantly decreased the association of *Hnrnpr* with ribosomes (Fig. 6a). To substantiate this finding, we transduced motoneurons with the *EGFP-FL* and *-ΔPBS Hnrnpr* 3′ UTR lentiviral constructs and assessed *EGFP* co-purification following ribosome immunoprecipitation with Y10b. We observed that co-purification of *EGFP* containing the *Hnrnpr* ΔPBS 3′ UTR was reduced compared to the FL 3′ UTR, further demonstrating that Ptbp2 binding mediates the association of *Hnrnpr* with ribosomes (Fig. 6b). Additionally, we prepared a lentiviral construct for expressing an *EGFP* reporter mRNA fused to the PBS (*EGFP-PBS*). We found that the association of this reporter with ribosomes was enhanced compared to the control *EGFP* transcript (Fig. 6c–e). Thus, Ptbp2 binding is necessary and sufficient to promote mRNA recruitment to ribosomes. To test whether Ptbp2 itself associates with ribosomes, we immunoprecipitated Ptbp2 from motoneuron lysate and detected co-purification of the ribosomal protein Rps5 (Fig. 6f). Conversely,

Ptbp2 co-purified with ribosomes immunoprecipitated with Y10b (Fig. 6g). Together, these data indicate that Ptbp2 modulates hnRNP R translation through regulating the association of *Hnrnpr* mRNA with ribosomes.

Next, we investigated the translation stage at which Ptbp2 regulates hnRNP R protein synthesis. To assess the association of Ptbp2 with ribosomes in more detail, we performed sucrose density gradient ultracentrifugation of motoneuron lysate to separate polysomes from 80S monosomes and 40S and 60S ribosomal subunits (Fig. 6h). We observed that the ribosomal component Rps5 was equally present in all ribosome fractions along the gradient (Fig. 6h). While Ptbp2, similar to eIF2α, was present at higher levels in the 40S, 60S, and monosome fractions, it was detectable also in polysome fractions (Fig. 6h). Additionally, Ptbp2 was present in the light fractions containing mRNPs. We then subjected lysate from control and *Ptbp2* knockdown motoneurons to sucrose gradient ultracentrifugation to assess how Ptbp2 modulates the association of *Hnrnpr* with ribosomes. The polysome profile of Ptbp2-depleted motoneurons was similar to controls indicating that Ptbp2 downregulation does not globally affect translation (Fig. 6i, j). Strikingly, Ptbp2 depletion caused a shift in *Hnrnpr* mRNA distribution from heavy polysomes to monosomes and ribosomal subunit fractions (Fig. 6k). In contrast, the distribution of 18S rRNA along the gradient was unaffected by Ptbp2 loss in agreement with the polysome profile that was similar between control and *Ptbp2* knockdown motoneurons (Fig. 6l). As additional controls, the distribution of *Gapdh* mRNA was not altered (Fig. 6m) and the total levels of *Hnrnpr* mRNA were unchanged by Ptbp2 depletion (Fig. 6n). To determine the specificity of the *Hnrnpr* distribution along the polysome profile, we disrupted the association of mRNAs with polysomes by treating motoneuron lysate with EDTA. This caused a shift of the *Hnrnpr* distribution towards lighter fractions of the gradient in the EDTA-treated compared to untreated lysate (Supplementary Fig. 4f). Similarly, *Gapdh* and *Actb* mRNA were more abundant in the unbound and monosome fractions of EDTA-treated lysate (Supplementary Fig. 4g). Taken together, these data suggest that Ptbp2 regulates hnRNP R protein synthesis in motoneurons by modulating the association of *Hnrnpr* with translating ribosomes.

## Ptbp2-mediated association of *Hnrnpr* with ribosomes involves eIF5A2

Our data show that *Hnrnpr* mRNA shifts from polysome to monosome fractions upon knockdown of *Ptbp2*. To further elucidate the mechanism behind this observation, we first investigated whether Ptbp2 binds to the initiation factor eIF2α and whether eIF2α binding to ribosomes is regulated by Ptbp2. Whilst eIF2α was co-precipitated by anti-Ptbp2 in an RNA-independent manner (Supplementary Fig. 5a, b), we did not observe any alteration in eIF2α co-precipitation with Y10b upon *Ptbp2* knockdown (Supplementary Fig. 5c). This indicates that Ptbp2 associates with ribosomes already at the initiation stage, but does not affect protein production at this step.

To assess whether Ptbp2 has a role in translation elongation, we inspected an interactome database[24] and identified the translation factor eIF5A2 as a PTBP2 interactor. Two isoforms of eIF5A (eIF5A1 and 2) have been identified sharing 84% identity[25]. eIF5A1 is ubiquitously expressed, whereas the expression of eIF5A2 appears to be tissue-specific and enriched in the nervous system[26]. Both eIF5A1/2 are the only proteins containing hypusine (hydroxyputrescine lysine)[27] and regulate translation elongation of proteins containing poly-proline stretches[25–28]. hnRNP R contains several proline-rich sequences[29] pointing towards the possibility that its translation is dependent on eIF5A1/2. To investigate whether eIF5A2 is involved in the Ptbp2-dependent regulation of hnRNP R translation, we assessed the association between Ptbp2 and eIF5A1/2 by immunopurification. For this purpose, we expressed EGFP-Ptbp2 in motoneurons and immunopurified it using GFP-Trap beads. We found that eIF5A1/2 was co-

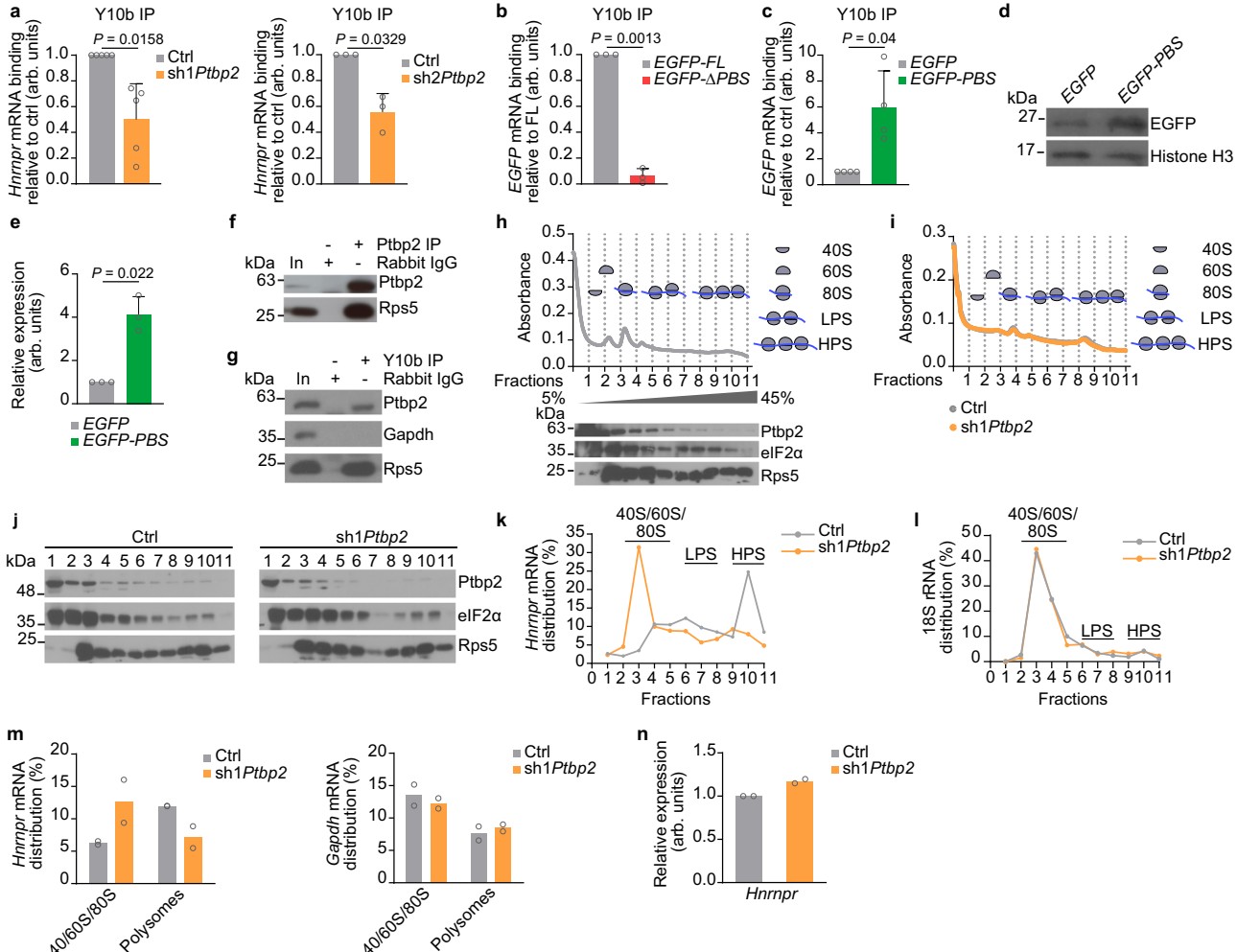

**Fig. 6 | hnRNP R translation efficiency is regulated by Ptbp2 in motoneurons.**
**a** qPCR analysis of *Hnrnpr* co-precipitated by the anti-ribosomal RNA antibody Y10b from control and Ptbp2 depleted-motoneurons. Two-tailed one-sample *t* test. Data are mean ± s.d. of *n* = 5 (sh1*Ptbp2*) and *n* = 3 (sh2*Ptbp2*) biological replicates. **b** qPCR analysis of *EGFP* co-precipitated by Y10b from motoneurons transduced with lentiviruses for expression of *EGFP* fused to full-length (*EGFP-FL*) or *ΔPBS Hnrnpr 3′* UTR (*EGFP-ΔPBS*). Two-tailed one-sample *t* test. Data are mean ± s.d. of *n* = 3 biological replicates. **c** qPCR analysis of *EGFP* co-precipitated by Y10b from motoneurons transduced with lentiviruses for expression of *EGFP* control or *EGFP* fused to the Ptbp2 binding site (PBS) present in the *Hnrnpr 3′* UTR. Two-tailed one-sample *t* test. Data are mean ± s.d. of *n* = 4 biological replicates. **d** Immunoblot analysis of EGFP in control motoneurons and motoneurons transduced with a lentivirus expressing *EGFP-PBS* at DIV 6. Histone H3 was used as a loading control. **e** Quantitative analysis of Western blots as shown in (**d**) for EGFP protein. Two-tailed one-sample *t* test. Data are mean ± s.d. of *n* = 3 biological replicates. **f** Immunoblot analysis of Rps5 co-immunoprecipitated by anti-Ptbp2 from motoneurons. The immunoblot is representative of three biological replicates. **g** Co-

immunoprecipitation of Ptbp2 by Y10b from motoneurons. Rps5 co-precipitation was used as a specificity control. The immunoblot is representative of two biological replicates. **h** Polysome analysis of motoneuron lysates by sucrose density gradient ultracentrifugation. LPS, light polysome fractions; HPS, heavy polysome fractions. Immunoblot analysis of Ptbp2, Rps5 and eIF2α. The immunoblot is representative of two biological replicates. **i** Polysome profiles of control and Ptbp2-depleted motoneurons. Absorbance profile is representative of two biological replicates. **j** Immunoblot analysis of Ptbp2, Rps5 and eIF2α across fractions from control and *Ptbp2* knockdown motoneurons. For fractions 1 and 2, 10% were loaded for analysis. The immunoblot is representative of two biological replicates. Relative levels of *Hnrnpr* (**k**) and 18S rRNA (**l**) in fractions of control and Ptbp2-depleted motoneurons. Data are the mean of *n* = 2 biological replicates.
**m** Quantification of the distribution of *Hnrnpr* and *Gapdh* mRNAs between ribosome subunits and polysomes in control and Ptbp2-depleted motoneurons. Data are mean of *n* = 2 biological replicates. **n** Quantification of the total levels of *Hnrnpr* mRNA. 18S rRNA was used for normalization. Data are mean of *n* = 2 biological replicates. Source data are provided as a Source Data file.

purified with EGFP-Ptbp2 but not with EGFP alone indicating that Ptbp2 interacts with eIF5A1/2 in motoneurons (Fig. 7a). To confirm the interaction between Ptbp2 and eIF5A1/2 in situ, we performed PLA using antibodies against eIF5A1/2 and Ptbp2. We found that the Ptbp2-eIF5A1/2 PLA signal was detectable not only in the cytosol of the somata but also in axons and growth cones of motoneurons (Fig. 7b and Supplementary Fig. 5d). Thus, Ptbp2-eIF5A1/2 complexes are present in axons in line with our finding that axonal *Hnrnpr* is translated locally in this neuronal compartment. To further assess the subcellular site of Ptbp2 interaction with eIF5A1/2, we performed immunoprecipitations from nuclear soluble and cytosolic fractions of motoneurons. Immunoprecipitation with anti-Ptbp2 showed that the

association of Ptbp2 with eIF5A1/2 was most prominent in the cytosolic fraction (Fig. 7c).

To investigate whether eIF5A1/2 is involved in *Hnrnpr* translation, we evaluated whether eIF5A1/2 is associated with *Hnrnpr* mRNA by RNA immunoprecipitation. Compared to immunoprecipitation with control IgG, *Hnrnpr* was enriched in the eIF5A1/2 co-precipitate (Fig. 7d, e). We also assessed our *Hnrnpr 3′* UTR protein interactome dataset from NSC-34 cells but were unable to identify eIF5A1/2. The most likely explanation for this result is that, for the pulldown experiments, only the *Hnrnpr 3′* UTR was used that does not associate with elongating ribosomes containing eIF5A1/2. In motoneurons depleted of Ptbp2, the association of *Hnrnpr* with eIF5A1/2 was

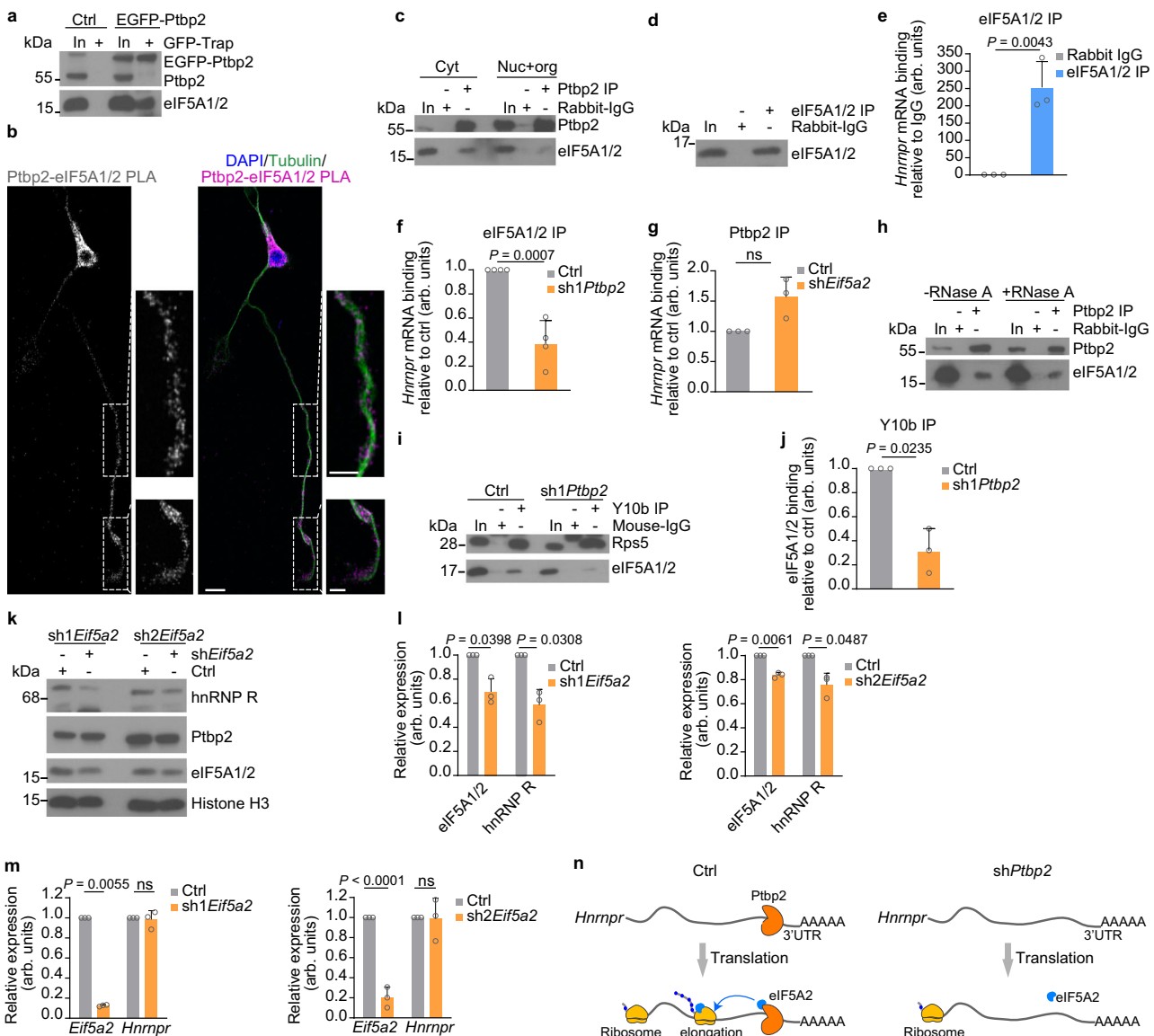

**Fig. 7 | Ptbp2 regulates hnRNP R translation through interaction with eIF5A.**
**a** Co-immunoprecipitation of eIF5A1/2 by GFP-Trap from motoneurons transduced with lentiviruses for expression of EGFP or EGFP-Ptbp2. The immunoblots are representative of two biological replicates. **b** Representative images of PLA signal in motoneurons at DIV 6 using anti-Ptbp2 and anti-eIF5A1/2 antibodies. Scale bars, 10 μm and 5 μm (magnified areas). The images are representative of three biological replicates. **c** Immunoblot analysis of eIF5A1/2 co-immunoprecipitated by anti-Ptbp2 from subcellular fractions of motoneurons. The immunoblots are representative of three biological replicates. **d** Immunoblot analysis of eIF5A1/2 immunoprecipitation from motoneurons. **e** qPCR analysis of *Hnrnpr* co-precipitated by anti-eIF5A1/2 from motoneurons. Two-tailed one-sample *t* test. Data are mean ± s.d. of *n* = 3 biological replicates. **f** qPCR analysis of *Hnrnpr* co-precipitated by anti-eIF5A/2 from control and *Ptbp2* knockdown motoneurons. Two-tailed one-sample *t* test. Data are mean ± s.d. of *n* = 4 biological replicates. **g** qPCR analysis of *Hnrnpr* co-precipitated by anti-Ptbp2 from control and *eIF5A2* knockdown motoneurons. Two-tailed one-

sample *t* test. Data are mean ± s.d. of *n* = 3 biological replicates. **h** Co-immunoprecipitation of eIF5A1/2 by anti-Ptbp2 from motoneuron lysate treated with RNase A for 15 min. The immunoblot is representative of three biological replicates. **i** Co-immunoprecipitation of eIF5A1/2 by Y10b from control and Ptbp2-depleted motoneurons. **j** Quantification of eIF5A1/2 co-purification by Y10b in (**i**). Two-tailed one-sample *t* test. Data are mean ± s.d. of *n* = 3 biological replicates. **k** Immunoblot analysis of hnRNP R, Ptbp2 and eIF5A1/2 in control and eIF5A2-depleted motoneurons at DIV 6. Histone H3 was used as a loading control. The immunoblot is representative of three biological replicates. **l** Quantification of Western blot data in (**k**). Two-tailed one-sample *t* test. Data are mean ± s.d. of *n* = 3 biological replicates. **m** Quantification of *Hnrnpr* and *Eif5a2* mRNA levels by qPCR in control and eIF5A2-depleted motoneurons. 18S rRNA was used for normalization. Two-tailed one-sample *t* test. Data are mean ± s.d. of *n* = 3 biological replicates. **n** Schematic representation of hnRNP R translation regulation by Ptbp2 through association with eIF5A2. Source data are provided as a Source Data file.

strongly reduced while the levels of eIF5A1/2 were unchanged (Fig. 7f and Supplementary Fig. 5e). To test whether, vice versa, eIF5A2 regulates binding of Ptbp2 to *Hnrnpr* mRNA, we have knocked down *Eif5a2* and found no statistically significant difference in Ptbp2 binding to *Hnrnpr* (Fig. 7g). Thus, Ptbp2 association with *Hnrnpr* occurs independent of eIF5A1/2.

Next, we tested whether the observed interaction of eIF5A1/2 with Ptbp2 is RNA-dependent. Following pre-treatment of motoneuron

lysate with RNase A, we observed that eIF5A1/2 still co-purified with Ptbp2, suggesting that their association is mRNA-independent (Fig. 7h). To investigate whether the association of eIF5A1/2 with ribosomes is regulated by Ptbp2 in motoneurons, we immunoprecipitated ribosomes with Y10b from control and *Ptbp2* knockdown motoneurons and assessed eIF5A1/2 co-purification. While co-purification of Rps5 with Y10b was unaffected by *Ptbp2* knockdown, co-precipitation of eIF5A1/2 was reduced upon loss of Ptbp2 (Fig. 7i, j).

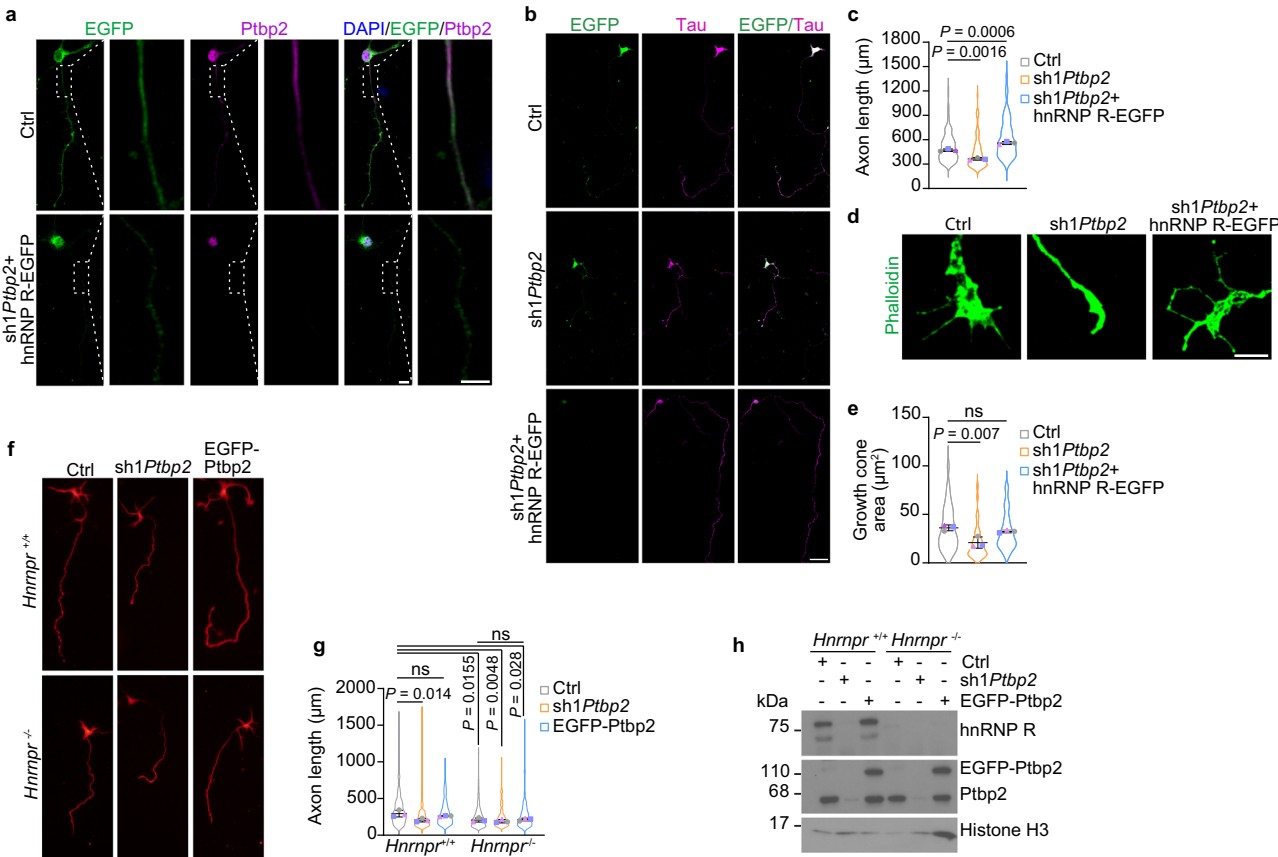

**Fig. 8 | The function of Ptbp2 in axon growth is dependent on hnRNP R.**
**a** Representative images of control and Ptbp2-depleted motoneurons expressing hnRNP R-EGFP (sh1*Ptbp2*+hnRNP R-EGFP) at DIV 7 immunostained for EGFP and Ptbp2. Scale bar, 10 μm and 5 μm (magnified areas). The images are representative of three biological replicates. **b** Representative images of control and Ptbp2-depleted motoneurons, and of Ptbp2-depleted motoneurons expressing hnRNP R-EGFP at DIV 7. Motoneurons were immunostained against tau and EGFP. Scale bar, 50 μm. **c** SuperPlots of axon length measurements. One-way ANOVA with Tukey's multiple comparisons test. Data are mean ± s.d. of *n* = 3 biological replicates. Representative images (**d**) and SuperPlots (**e**) of growth cone sizes of control and *Ptbp2* knockdown motoneurons, and of Ptbp2-depleted motoneurons

expressing hnRNP R-EGFP at DIV 6. Motoneurons were stained with phalloidin. Scale bar, 5 μm. One-way ANOVA with Tukey's multiple comparisons test. Data are mean ± s.d. of *n* = 3 biological replicates. **f** Tau immunostaining of cultured *Hnrnpr*[+/+] and *Hnrnpr*[-/-] motoneurons at DIV 6. Scale bar, 50 μm. **g** SuperPlots of axon lengths. One-way ANOVA with Tukey's multiple comparisons test. Data are mean ± s.d. of *n* = 3 biological replicates. **h** Immunoblot analysis of Ptbp2 and hnRNP R in *Hnrnpr*[+/+] and *Hnrnpr*[-/-] motoneurons transduced with control lentivirus, and in *Hnrnpr*[+/+] and *Hnrnpr*[-/-] motoneurons transduced with lentiviruses for *Ptbp2* knockdown or overexpression. The immunoblots are representative of three biological replicates. Source data are provided as a Source Data file.

Thus, eIF5A1/2 is an interactor of Ptbp2 and associates with ribosomes in a Ptbp2-dependent manner. Finally, we sought to test whether eIF5A2 regulates hnRNP R protein synthesis. Following knockdown of *Eif5a2* in motoneurons, hnRNP R protein levels were significantly downregulated (Fig. 7k, l) whereas the amounts of *Hnrnpr* mRNA or of Ptbp2 were unchanged (Fig. 7k–m). Taken together, our results suggest that Ptbp2 regulates hnRNP R translation in motoneurons through cooperation with eIF5A2 (Fig. 7n).

**The function of Ptbp2 in axon growth is dependent on hnRNP R**
Our results show that Ptbp2 regulates axon growth of motoneurons and indicate that hnRNP R acts downstream of Ptbp2 in this process. To test this possibility, we performed a rescue experiment in which we restored hnRNP R levels in Ptbp2-depleted motoneurons by over-expressing EGFP-hnRNP R (Fig. 8a). We found that re-expressing hnRNP R could restore axon length and growth cone size in Ptbp2-depleted motoneurons (Fig. 8b–e). Next, we investigated whether overexpression of Ptbp2 can rescue axon growth of hnRNP R-deficient motoneurons derived from an *Hnrnpr* knockout mouse. For this purpose, we cultured motoneurons from *Hnrnpr*[-/-] and *Hnrnpr*[+/+] mice and overexpressed EGFP-Ptbp2 by lentiviral transduction. We found that overexpression of Ptbp2 could not rescue the axon growth defect of

*Hnrnpr*[-/-] motoneurons (Fig. 8f–h). Additionally, we found that there was no further reduction in axon length of *Hnrnpr*[-/-] motoneurons following transduction with the sh1*Ptbp2*-expressing lentivirus. These results indicate that hnRNP R is required for Ptbp2-dependent regulation of axon growth.

## Discussion
Axonal mRNA transport and local mRNA translation contribute to the growth, differentiation, and maintenance of axons and synaptic connections[30]. These processes are regulated by RBPs, which interact with their target mRNAs through cis-regulatory elements in UTRs, forming messenger ribonucleoprotein (mRNP) complexes that are directed into axons and terminal growth cones[4,31]. Ptbp2 as a neuronal RBP has been characterized in detail for its functions in alternative splicing regulation, which contributes to neuronal development and axonogenesis[8,10–12,32]. Here, we show that, beyond its nuclear functions, Ptbp2 regulates axonal transport as well as axonal translation of *Hnrnpr* (Fig. 9). We found that Ptbp2 associates with *Hnrnpr* mRNA in the cytosol of motoneurons as part of mRNP particles, and that such particles are located in axons for local synthesis of hnRNP R. This mechanism is important for axon growth and extends our current understanding of Ptbp2 functions in neuronal development. It remains

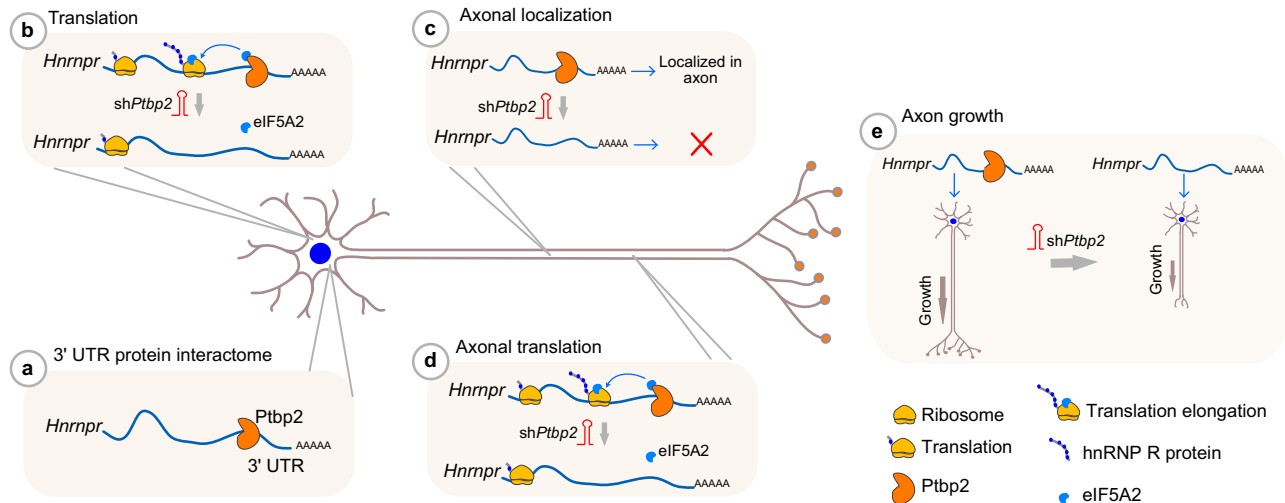

**Fig. 9 | Schematic summary of cytosolic Ptbp2 functions in axon growth.**
**a** Ptbp2 binds to the 3′ UTR of *Hnrnpr* mRNA in motoneurons. **b** Ptbp2 regulates hnRNP R translation in association with eIF5A2. Ptbp2 controls axonal localization (**c**) and translation (**d**) of *Hnrnpr* mRNA in motoneurons. **e** Ptbp2 modulates axon growth of motoneurons.

to be determined in future studies whether the reduced axonal levels of *Hnrnpr* mRNA in Ptbp2-depleted motoneurons are a consequence of defective transport or reduced local mRNA stability. Additionally, while our Puro-PLA data indicate reduced axonal translation of hnRNP R in *Ptbp2* knockdown motoneurons, we cannot rule out the possibility that the reduced axonal localization of *Hnrnpr* mRNA we observed in Ptbp2-deficient axons contributes to this effect. However, according to our qPCR data from compartmentalized motoneurons, the reduction at the *Hnrnpr* mRNA level is less than that observed by hnRNP R Puro-PLA in axons of Ptbp2-depleted motoneurons. Additionally, we also observed reduced hnRNP R translation in the somata of *Ptbp2* knockdown motoneurons. Since total *Hnrnpr* mRNA levels are not affected by *Ptbp2* knockdown, this finding supports the notion that Ptbp2 regulates axonal hnRNP R protein levels by promoting its local translation in addition to modulating its axonal mRNA localization.

A major finding of our study is the detection of Ptbp2 in axons and growth cones and the identification of its regulatory role in local synthesis of hnRNP R. While it is widely known that cytoskeletal components and regulators are locally synthesized in axons during axon elongation, branching, synapse formation, and synaptic transmission[33–35], only very few cases of locally translated RBPs have been identified so far. For example, a recent study has revealed that the axonal translation of KH splicing regulatory protein serves to promote decay of other axonal mRNAs following nerve injury and slow axon regeneration[36]. We thus hypothesize that the local synthesis of hnRNP R might induce remodeling of axonal mRNPs, further affecting the localization and translation of other mRNAs involved in axon development and synapse formation. Interestingly, axonal ribosomes themselves have recently been shown to be locally remodeled through the exchange of ribosomal proteins[37,38]. Our result showing that Ptbp2 associates with ribosomes and translation factors in axons indicates that it might not only regulate the local synthesis of hnRNP R but also modulate the activity and composition of axonal ribosomes in general, either separately or as part of mRNP particles. Whilst we observed that Ptbp2 associates with translation initiation complexes, we also showed that Ptbp2 binds to eIF5A1/2 in an RNA-independent manner, promoting the association of eIF5A1/2 with ribosomes and of eIF5A1/2 with *Hnrnpr* mRNA. eIF5A1/2 has previously been shown to interact with elongating ribosomes rather than translation initiation complexes[28,39,40]. This puts forward a model according to which Ptbp2 associates with the 3′ UTR of *Hnrnpr* and possibly other mRNAs, and stimulates their translation through modulating their association with translating ribosomes. According to a published RNA-seq dataset of

*Ptbp2* knockout brain[10], we found no indication for dysregulated splicing or reduced levels of transcripts encoding translation factors upon loss of Ptbp2. Together with our finding that global polysome profiles are not altered by Ptbp2 deficiency, this indicates that the reduced binding of eIF5A1/2 to ribosomes upon Ptbp2 depletion is not a consequence of dysregulated ribosome composition.

Our finding that Ptbp2-deficient motoneurons exhibit reduced axon growth and impaired growth cone maturation points towards the possibility that the presynaptic machinery for establishment of neuromuscular endplates might also be affected in motoneurons depleted of Ptbp2. This possibility is supported by the observation that the breathing of *Ptbp2*[−/−] mice is disturbed due to impaired synaptic innervation of the diaphragm muscle[10,11]. Several critical synaptic proteins including postsynaptic density protein 95, the neurotransmitter receptor NR2B, the major synaptic vesicle protein Synaptophysin, the trans-synaptic scaffolding protein Neuroligin1, calcium/calmodulin-dependent protein kinase Cask and the Glutamate Receptor GluR2 have been reported to be highly diminished in the E18 *Ptbp2*[−/−] brain[10]. Thus, deficits in synaptic transmission might contribute to the paralysis phenotype observed in pan-neural *Ptbp2* knockout pups[10]. In agreement with the synaptic defects observed in *Ptbp2*[−/−] mice, we observed reduced levels of Piccolo, a marker for synaptic maturation, in *Ptbp2* knockdown motoneurons. In contrast, levels of Synapsin I were unchanged by Ptbp2 deficiency indicating that early synapse formation is unperturbed.

Failure to form or maintain synaptic connections can result in neurodevelopmental or neurodegenerative disorders and it is possible that defects in Ptbp2-dependent axonal differentiation and signaling contribute to the pathomechanisms underlying such diseases. Multiple studies have identified mutations in *PTBP2* to be associated with autism spectrum disorder (ASD)[41–44]. In ASD, an imbalanced synaptic connectivity has been revealed as a primary causative event[45,46]. Mutations related to *PTBP2* could not only affect expression of PTBP2 but also dysregulate its localization in subcellular compartments leading to impairment of synaptic transmission and thus contribute to the ASD phenotype. In addition, the expression level of *PTBP2* mRNA in the substantia nigra of patients with Parkinson disease (PD) was reduced compared to healthy controls[47]. Disruption of striatal synaptic plasticity has a crucial role in the pathogenesis of PD[48] and it is possible that aberrant PTBP2 expression is involved in such impairments. Our finding that Ptbp2 exerts axon-specific functions thus not only expands our view of its repertoire of functions but also points towards

the possibility that defective axonal PTBP2 signaling mechanisms contribute to synaptic pathologies seen in several neurodevelopmental and neurodegenerative disorders.

## Methods

### Animals and ethical approval

All of the experimental procedures in this study involving mice were performed according to the regulations on animal protection of the German federal law and the Association for Assessment and Accreditation of Laboratory Animal Care, in agreement with and under the control of the local veterinary authority. Mice were housed in the animal facility of the Institute of Clinical Neurobiology at the University Hospital of Wuerzburg. The CD1 and *Hnrnpr* knockout mice were maintained on a 12 h/12 h day/night cycle under controlled conditions at 20–22 °C and 55–65% humidity with food and water in abundant supply.

### Isolation and enrichment of primary embryonic mouse motoneurons

Isolation and enrichment of primary mouse motoneurons were performed as previously described[49]. Briefly, lumbar spinal cords were isolated from E13 mouse embryos, and motoneurons were enriched by panning using a p75[NTR] antibody (clone MLR2, Biosensis). Cells were plated on coverslips or culture dishes coated with poly-DL-ornithine hydrobromide (PORN) (P8638, Sigma) and laminin-111 (23017-015, Thermo Fisher Scientific). Motoneurons were maintained at 37 °C, 5% $CO_2$ in neurobasal medium (Gibco) supplemented with 2% B27 (Gibco), 2% heat-inactivated horse serum (Linaris), 500 µM GlutaMAX (Gibco) and 5 ng/ml of brain-derived neurotrophic factor (BDNF). Medium was replaced one day after plating and then every second day. Motoneurons were cultured in compartmentalized microfluidic chambers (Xona Microfluidics, Cat. No. IND150, RD150) as described previously. In brief, motoneurons were plated into one side, which served as the somatodendritic compartment. During the 7 days culture, axons grew through the microgrooves to the other compartment (the axonal compartment) as a response to a BDNF gradient, which was applied at 20 ng/ml only to the axonal compartment.

### Cell lines

NSC-34 cells (Cedarlane, cat. no. CLU140) and HEK293TN cells (System Biosciences, cat. no. LV900A-1) were grown in high glucose Dulbecco's modified Eagle's Medium (Gibco) supplemented with 10% fetal calf serum (Linaris), 2 mM GlutaMAX (Gibco) and 1% penicillin–streptomycin (Gibco) in a humidified incubator at 37 °C with 5% $CO_2$. Cells were passaged when they were 80–90% confluent. Cells were tested negative for mycoplasma contamination.

### Plasmid construction

The *EGFP* coding sequence was PCR-amplified and cloned into the BamHI site of a version of the pcDNA3.1 plasmid from which the BGH terminator had been deleted. *Hnrnpr* 3′ UTRs (short, medium, long) were PCR-amplified from mouse genomic DNA and inserted downstream of the *EGFP* stop codon, leaving only a SacII restriction enzyme site in between. Due to its size, cloning of the long 3′ UTR was challenging and only one clone was obtained. It contained several mutations in highly repetitive regions (numbered from the start position of the short UTR: 3738 AT insertion, 7256 TTT deletion, 9034 CTG deletion, 10321 AAA deletion, 15214T deletion, 15527T insertion), which, however, were outside conserved regions.

shRNAs targeting *Ptbp2* and *Eif5a2* were cloned into a modified version of pSIH-H1 shRNA vector (System Biosciences) containing EGFP according to the manufacturer's instructions. The following target sequences were used for designing shRNA oligonucleotides: sh1*Ptbp2*: 5′-GCTGTTATCATTCCTTGGTTA-3′, sh2*Ptbp2*: 5′-GGTAACGA TAGTAAGAAATTT-3′, sh1*Eif5a2*: 5′-CACAACATGGATGTTCCAAAT-3′,

sh2*Eif5a2*: 5′-AGGCAGTTAGTAAGTTTAAA-3′. To generate Ptbp2 overexpression constructs, the mouse Ptbp2 coding sequence and the coding sequence of EGFP were PCR-amplified and inserted into pSIH-H1 digested with SalI (FD0644, Thermo Fisher Scientific) and NheI (FD0973, Thermo Fisher Scientific) using the NEBuilder HiFi DNA Assembly Cloning Kit (New England Biolabs). To generate the EGFP-Ptbp2 rescue construct, the sequence encoding EGFP-tagged Ptbp2 was into the sh1*Ptbp2*-expression construct. Since sh1*Ptbp2* targets the 3′ UTR of endogenous *Ptbp2* mRNA, EGFP-Ptbp2 lacking the 3′ UTR is resistant to knockdown. The sequence encoding EGFP-Ptbp2-ΔNLS was assembled by gene synthesis and PCR amplification and subcloned into the sh1*Ptbp2*-expression construct.

The construct for the rescue of Ptbp2-depleted motoneurons by overexpression of hnRNP R was created by cloning the sequence encoding EGFP-tagged hnRNP R into pSIH-H1 containing sh1*Ptbp2*. For EGFP-tagged hnRNP R, the exonic and intronic sequences of exon 1 to exon 3, the *Hnrnpr* coding sequence from exon 4 onwards, and the coding sequence of EGFP without its ATG start codon were PCR-amplified and inserted into pSIH-H1. Empty pSIH-H1 expressing EGFP was used as a control in all the experiments. The plasmids expressing the short full-length *Hnrnpr* 3′ UTR (FL) or ΔPBS 3′ UTR (ΔPBS) were generated by PCR-amplifying the FL or ΔPBS 3′ UTR sequences and inserting them into pSIH-H1 downstream of the EGFP stop codon. To create the plasmid expressing *EGFP* mRNA fused to the Ptbp2 binding site (PBS), we inserted the synthesized PBS into pSIH-H1 downstream of the EGFP stop codon. All plasmids were verified by DNA sequencing.

### Transfection

Transient transfections were performed with 1 µg of plasmid resuspended in 50 µl of Opti-MEM supplied with 2.8 µl of TurboFect reagent according to the manufacturer's reverse transfection protocol using the TurboFect Transfection Reagent (Thermo Scientific). Cells were collected at 48 h after transfection for RNA or protein extraction.

### Lentiviral transduction

Lentiviral particles were packaged in HEK293TN cells (System Biosciences, cat. no. LV900A-1) cells with pCMV-pRRE, pCMV-pRSV, and pCMV-pMD2G as described before[50]. Transduction was performed by incubation of motoneurons and NSC-34 cells with lentiviruses in a total volume of 50 µl for 10 min at room temperature before plating at DIV 0.

### RNA extraction

Total RNA was extracted from NSC-34 cells and motoneurons using the NucleoSpin RNA kit (Macherey-Nagel). For mouse tissue, samples were collected in microcentrifuge tubes and immediately snap-frozen by immersion in liquid nitrogen. Samples were kept at −80 °C until processing. Tissues were ground with mortar and pestle and RNA was isolated using the NucleoSpin RNA kit (Macherey-Nagel, 740955.50).

### cDNA synthesis and qPCR analysis

cDNA synthesis was performed using the First Strand cDNA Synthesis Kit (Thermo Fisher Scientific, K1612) using random hexamers. Reverse transcription reactions were diluted 1:5 in RNase-free water. qPCR reactions were set up with the Luminaris HiGreen qPCR Master Mix (Thermo Fisher Scientific, K0994) and run on a LightCycler 96 (Roche). qPCR primers were designed using the online Primer3Plus design tool and the sequences are listed in Supplementary Data 2. An annealing temperature of 60 °C was used for all primers. Two technical replicates were set up for each sample and averaged before normalization and statistical analyses. Relative expression was calculated using the ΔΔCt method.

### Absolute quantification of the UTR isoforms

For absolute quantification of the UTR isoforms, qPCR amplicons were cloned into the pJET vector (Clone JET PCR Cloning Kit, Thermo Fisher

Scientific, K1231) and sequenced to confirm their identity. Three independent serial dilutions of the plasmids were then made, with calculated concentrations ranging from $10^8$ to $10^1$ copies/2 µl. The crossing point (Ct) of each serial dilution was measured by qPCR and values from replicates were averaged. Standard curves were fitted by linear regression of data points composed of Ct values (y-axis) and the respective copy numbers (x-axis). The copy number of a sample was determined by interpolating the Ct value in the respective standard curve. Expression levels between tissues were normalized to 7SL lncRNA.

### RNA pulldown assay

Templates for the in vitro transcription (IVT) reaction were PCR-amplified from the pcDNA3.1 plasmid containing the short 3′ UTR with the KAPA HiFi HotStart ReadyMix kit (Roche, KK2601) using primers that included the T7 promoter sequence in the forward primer for generating sense RNA or reverse primer for generating antisense RNA. IVT was performed with the T7 RNA Polymerase kit (Thermo Scientific, EP0111) with a 1:10 ratio of biotin-16-UTP (Jena Bioscience, NU-821-BIO16) and 0.25 U/50 µl inorganic pyrophosphatase (Thermo Scientific, EF0221) and left overnight at 37 °C. Afterward, the DNA template was digested by the addition of 4 U of TURBO DNase (Ambion, AM2238) and incubation for 30 min at 37 °C. RNA was then purified on a NucleoSpin RNA column (Macherey-Nagel) and an aliquot was subjected to agarose gel electrophoresis for size confirmation.

Cell culture pellets were resuspended in lysis buffer (150 mM KCl, 2 mM MgCl₂, 20 mM Tris-HCl pH 7.4, 0.1% NP-40) containing 50 U/ml RNase Inhibitor, incubated for 20 min on ice, and cleared by centrifugation at $19,400 \times g$ for 15 min at 4 °C. 4 µg biotinylated RNA was incubated with 40 µl pre-washed streptavidin beads (Thermo Fisher Scientific, 88817) for 15 min at 4 °C with intermittent shaking at 800 rpm to allow streptavidin-biotin binding. Beads were then washed and the lysate was added. After incubation at 4 °C for 2 h on a rotating mixer, beads were collected and washed four times with lysis buffer (the last two washes with buffer without NP-40). For SDS-PAGE and Coomassie staining, proteins were eluted from the beads by resuspension in 1× Laemmli buffer (50 mM Tris-HCl pH 6.8, 1% SDS, 6% glycerol, 1% β-mercaptoethanol, 0.004% bromophenol blue) and boiling. For mass spectrometry analysis, beads of four technical replicates were snap-frozen in liquid nitrogen and stored at −80 °C until use.

### RNA immunoprecipitation

NSC-34 cells were grown in 10 cm dishes until 80–90% confluency. Cells were washed once with ice-cold Dulbecco's Phosphate Buffered Saline (DPBS, without MgCl₂, CaCl₂; Sigma-Aldrich, D8537) and collected by scraping. Cells were lysed in 1 ml lysis buffer (150 mM KCl, 2 mM MgCl₂, 20 mM Tris-HCl pH 7.4, 0.1% NP-40) for 20 min on ice and cleared by centrifugation ($19,400 \times g$) for 15 min at 4 °C. Magnetic Dynabeads protein A or G (Thermo Fisher Scientific) were bound to 1 µg antibody or 1 µg IgG control for 40 min at room temperature by rotation. Beads were washed twice with lysis buffer, and 300 µl lysate was added to the beads for and incubated for 2 h at 4 °C by rotation. Beads were washed twice with 500 µl lysis buffer. Total RNA was extracted by adding 300 µl buffer A1 (NucleoSpin RNA kit) and 300 µl absolute ethanol to both input and beads followed by RNA extraction according to the manufacturer's instructions. RNA was reverse-transcribed with random hexamers using the First Strand cDNA Synthesis Kit (Thermo Fisher Scientific). Reverse transcription reactions were diluted 1:5 in water and transcript levels were measured by qPCR. Relative RNA binding was calculated using the ΔΔCt method with normalization to input levels.

For RNA immunoprecipitation from primary mouse motoneurons, cells were grown on laminin-111-coated 6 cm dishes for 7 DIV. Cells were washed once with DPBS and directly lysed in lysis buffer (10 mM HEPES pH 7.0, 100 mM KCl, 5 mM MgCl₂, 0.5% NP-40). The lysates were incubated for 20 min on ice and centrifuged at $20,000 \times g$ for 10 min at 4 °C. The rest of the experiment was performed the same as for NSC-34 cells.

### Co-immunoprecipitation

Primary mouse motoneurons were grown on laminin-111-coated 6 cm dishes for 7 DIV. Cells were washed once with DPBS and lysed in lysis buffer (10 mM HEPES pH 7.0, 100 mM KCl, 5 mM MgCl₂, 0.5% NP-40) on ice for 15 min and cleared via centrifugation at $20,000 \times g$ for 15 min at 4 °C. Protein G or A Dynabeads (depending on the antibody species) were bound to either nonspecific IgG or antibody by rotating for 40-60 min at room temperature. 300 µl lysate was added to the antibody-bound beads and rotated for 2 h at 4 °C. Beads were washed twice with 500 µl lysis buffer and proteins were eluted in 1× Laemmli buffer. Proteins were size-separated by SDS-PAGE and analyzed by immuno-blotting. Antibodies used and their dilutions are listed in Supplementary Data 3.

### Ribosome pulldown

NSC-34 cells were grown in 10 cm dishes until 80% confluence. The medium was then replaced with fresh medium containing 100 µg/ml cycloheximide (Sigma-Aldrich, C4859) and cells were incubated for 10 min in a cell culture incubator. Cells were washed two times with ice-cold Hanks' Balanced Salt Solution (HBSS; Gibco) containing 100 µg/ml cycloheximide. For each plate, cells were collected into HBSS containing 100 µg/ml cycloheximide by scraping and divided into two microcentrifuge tubes. Cells were pelleted by centrifugation at $10,000 \times g$ for 2 min at 4 °C, and pellets were kept at −80 °C until use.

The protocol for ribosome pulldown was performed as previously described with some modification[51]. Briefly, a cell pellet was lysed in 1 ml of lysis buffer (300 mM KCl, 2 mM MgCl₂, 20 mM Tris-HCl pH 7.4, 2 mM DTT, 0.05% sodium deoxycholate, 50 U RiboLock RNase Inhibitor (Thermo Fisher Scientific), 100 µg/ml cycloheximide) for 20 min on ice and cleared by centrifugation at $19,400 \times g$, 4 °C for 15 min. The supernatant was transferred to a new tube and genomic DNA was digested by treatment with 2 µl of Turbo DNase (Ambion) at 37 °C for 10 min. Three 10 µl aliquots of Protein A magnetic beads (Thermo Fisher Scientific) were washed twice with lysis buffer and resuspended in 300 µl lysis buffer containing 100 µg/ml yeast tRNA (Sigma-Aldrich). Two bead aliquots were incubated with 1 µg of Y10b antibody (Abcam, ab37144) each and the third with 1 µg mouse IgG control (Santa Cruz Biotechnology, sc-2025) for 1 h at 4 °C on a rotating mixer. The antibody-bound beads were washed twice with lysis buffer and resuspended in 720 µl lysis buffer (Y10b pulldown and IgG control samples) or 720 µl lysis buffer supplemented with 50 mM EDTA (Y10b + EDTA sample). 280 µl of the cleared cell lysate was then added to each tube and samples were incubated for 3.5 h at 4 °C on a rotating mixer. 100 µl of the lysate was saved as an input sample and kept at 4 °C. After incubation, beads were washed twice with 200 µl lysis buffer and resuspended in 100 µl lysis buffer. RNA from the three samples and the input was extracted with 1 ml TRIzol (Ambion).

For ribosome pulldown from primary mouse motoneurons, cells were grown on laminin-111-coated 6 cm dishes for 7 DIV. The medium was replaced with fresh medium containing 100 µg/ml cycloheximide and cells were incubated for 10 min in a cell culture incubator. Cells were washed two times with ice-cold HBSS containing 100 µg/ml cycloheximide and collected by scraping. Cells were pelleted by centrifugation at $10,000 \times g$ for 2 min at 4 °C. The pellet was lysed in 700 µl of lysis buffer (300 mM KCl, 2 mM MgCl₂, 20 mM Tris-HCl pH 7.4, 2 mM DTT, 0.05% sodium deoxycholate, 100 µg/ml cycloheximide) for 20 min on ice and cleared by centrifugation at $20,000 \times g$, 4 °C for 10 min. 1 µg of Y10b antibody (Santa Cruz Biotechnology, sc-33678) or mouse IgG was bound to 10 µl Protein G Dynabeads in 200 µl lysis buffer containing 100 µg/ml yeast tRNA at room temperature for

40 min. 100 µl of lysate was used as input sample, and 300 µl of lysate was incubated with the antibody-bound beads for 2 h at 4 °C by rotation. Beads were washed twice with 500 µl lysis buffer. For qPCR, RNA was extracted from the input sample and beads with the NucleoSpin RNA kit (Macherey-Nagel). For western blot analysis, proteins were eluted in 1× Laemmli buffer. Antibodies used and their dilutions are listed in Supplementary Data 3.

## Sample preparation for LC-MS/MS

Beads were thawed and resuspended in 20 µl of 50 mM Tris-HCl pH 7.5, 2 mM MgCl₂, 375 U/ml Benzonase and incubated for 1 h at room temperature to degrade the RNA. Beads were kept in solution by shaking at 750 rpm throughout all steps. Proteins were denatured by adding 150 µl of 8 M urea, 50 mM Tris-HCl pH 7.5 solution, and 5 µl of 30 µM DTT and then digested by LysC treatment (0.25 µg/sample) for 2 h at room temperature. Samples were diluted fourfold in 50 mM Tris-HCl pH 7.5 to lower urea concentration to permit tryptic digestion and chloroacetamide at a final concentration of 5 mM was added to alkylate cysteines. Trypsin (0.25 µg/sample) was added and samples were digested overnight at room temperature in the dark. The digestion was terminated by the addition of trifluoroacetic acid (final 1% v/v) and the beads were pelleted by centrifugation. Half of the supernatant was further processed by desalting chromatography on three discs of C18 material using the STAGE-tip format[52]. Briefly, STAGE-tips were washed with 100 µl buffer B (50% v/v acetonitrile, 0.5% v/v acetic acid), conditioned with 100 µl methanol, washed twice with 100 µl buffer A (2% v/v acetonitrile, 0.5% v/v acetic acid), loaded with sample peptides, washed twice with 100 µl buffer A and subjected to peptide elution by 60 µl of buffer B. The eluate was evaporated to dryness in a vacuum concentrator. Finally, peptides were re-suspended in 10 µl 2% v/v acetonitrile, 0.5% v/v acetic acid, 0.1% v/v trifluoroacetic acid and stored at −20 °C. 2 µl were later used for mass spectrometry.

## LC-MS/MS

Peptides were separated on an EASY-nLC 1000 HPLC system (Thermo Fisher Scientific) via in-house packed columns [75-µm inner diameter, 50 cm length, and 1.9 µm C18 particles (Dr. Maisch GmbH)] in a gradient of buffer A (0.5% formic acid) to buffer B (80% acetonitrile, 0.5% formic acid). The gradient started at 5% B, increasing to 30% B in 65 min, further to 95% B in 10 min, staying at 95% B for 5 min, decreasing to 5% B in 5 min and staying at 5% B for 5 min at a flow rate of 300 nl/min and a temperature of 60 °C. A Quadrupole Orbitrap mass spectrometer (Q Exactive HF-X; Thermo Fisher Scientific) was directly coupled to the LC via a nano-electrospray source. The Q Exactive HF-X was operated in a data-dependent mode. The survey scan range was set from 300 to 1650 m/z, with a resolution of 60,000 at m/z 200. Up to the 12th, most abundant isotope patterns with a charge of two to five were isolated and subjected to collision-induced dissociation fragmentation at a normalized collision energy of 27, an isolation window of 1.4 Th, and an MS/MS resolution of 15,000 at m/z 200. Dynamic exclusion to minimize resequencing was set to 30 s.

## Proteomics data processing and bioinformatics analysis

To process MS raw files, we employed the MaxQuant software version 1.6.0.15[53], searching against the UniProtKB mouse FASTA database using canonical and isoform protein sequences. Default search parameters were utilized unless stated differently. A false discovery rate (FDR) cut-off of 1% was applied at the peptide and protein level. The search feature "Match between runs", which allows the transfer of peptide identifications in the absence of sequencing after nonlinear retention time alignment, was enabled with a maximum retention time window of 0.7 min. Protein abundances were normalized with the MaxLFQ label-free normalization algorithm[54]. For bioinformatic analysis and visualization, we used the open PERSEUS environment version 1.5.2.11[55]. Common contaminants and peptides only identified by

site modification were excluded from further analysis. Protein abundance values were log2-transformed. Data were filtered for proteins with at least three valid values in the four replicates of at least one group (sense or antisense RNA). Missing values were imputed from a normal distribution using the default parameters (width = 0.3 and down-shift = 1.8 standard deviations of parent distribution). The processed (log2-transformed, filtered, imputed) protein-level mass spectrometry data can be found in Supplementary Data 1. The Volcano plot was generated with the built-in PERSEUS tool using t-test statistics combined with a permutation-based FDR of 5% and an s0-parameter of 0.1 for integrating the fold change effect size[56]. Proteins above the cutoff line have a q value below 5%.

## 3′ RACE

RNA from mouse brain was reverse-transcribed using SuperScript III Reverse Transcriptase (Invitrogen) and the primer 5′-GGCCACGCG TCGACTAGTACTTTTTTTTTTTTTTTTTTTV-3′. A negative control in which all reagents except the reverse transcriptase were added was also set up. cDNA was then purified with the QIAEX II Gel Extraction Kit (Qiagen), following the manufacturer's protocol for DNA fragments ≤4 kb. The DNA was eluted from the silica particles in 100 µl water. A first PCR was performed with the AccuPrime Taq DNA Polymerase System (Invitrogen, 12339016), using the adapter primer 5′-CCACGC GTCGACTAGTACTTT-3′ and a *Hnrnpr* 3′ UTR-isoform-specific forward primer: 5′-GCCACCAGACAAGAAAAGGA-3′ (short 3′ UTR), 5′-ACTGCC TGAGGTGTGCTTAC-3′ (medium 3′ UTR) and 5′-AGGCAGGTCACTT-GATGCTC-3′ (long 3′ UTR). The thermal amplification protocol was the following: 98 °C for 5 min, 58 °C for 2 min, 68 °C for 40 min, followed by 31 cycles of (94 °C for 10 s, 58 °C for 10 s, 68 °C for 3 min), followed by 68 °C for 12 min. The PCR product was then diluted at 1:20 in water and used as the template for a second, nested PCR with the adapter primer and a nested UTR-isoform-specific forward primer: 5′-GGAGG-CAAGAGAAAGGCA-3′ (short 3′ UTR), 5′-TCAGTGCCAGAAAAAGTGA GGT-3′ (medium 3′ UTR), and 5′-TCCAGAGCTGCACAGTTAGC-3′ (long 3′ UTR). The thermal protocol was the following: 98 °C for 5 min, followed by 31 cycles of (94 °C for 10 s, 60 °C for 10 s, 68 °C for 3 min), followed by 68 °C for 12 min. The PCR products of the second PCRs were sequenced through Sanger sequencing by LGC Genomics (Berlin, Germany).

## Junction PCR

1 µg of brain RNA was reverse-transcribed with the Maxima H Minus Reverse Transcriptase (Thermo Fisher Scientific) using 100 U of the reverse transcriptase and the primer 5′-TTTTTTTTTTTTTTTTTTTTTCAG-3′. The reaction was kept at 50 °C for 1 h, followed by 5 min at 85 °C. The cDNA was then diluted 1:5 in water and used as a template for the PCR, which was performed with the LongAmp Taq DNA Polymerase (New England Biolabs) using the primers 5′-ATACCGCCGCGG CTTTGCTACTCCTTAACTGGC-3′ and 5′-GCAGTTTGCTGCCATTTGTA-3′. The thermal protocol was the following: 94 °C for 5 min, 58 °C for 2 min, 65 °C for 40 min, followed by 31 cycles of (94 °C for 20 s, 58 °C for 20 s, 65 °C for 6 min 30 s), followed by 65 °C for 10 min.

## mRNA stability assay

NSC-34 cells were seeded in 24-well plates and grown until 80% confluence. Cells were treated with Actinomycin D (10 µg/ml) or the vehicle DMSO (0.01%) for the indicated durations. At the respective time points, cells were lysed, and RNA was isolated with TRIzol reagent (Ambion). *Hnrnpr* 3′ UTR levels were measured by qPCR and normalized to 18S rRNA.

## Fluorescence in situ hybridization

FISH was performed with the ViewRNA ISH Cell Assay kit (Invitrogen, QVC0001). Motoneurons were grown for 5–6 DIV on glass coverslips (Marienfeld GmbH, 0111500) coated with PORN and laminin-111. Cells

were washed three times with RNase-free PBS and fixed for 10 min in paraformaldehyde lysine phosphate (PLP) buffer pH 7.4, containing 4% paraformaldehyde (PFA) (Thermo Fisher Scientific, 28908), 5.4% glucose and 0.01 M sodium metaperiodate. After three PBS washes, cells were permeabilized with the detergent solution provided with the kit for 4 min. Afterwards, the *Hnrnpr* probe (Thermo Fisher Scientific, VB1-3039780) was diluted at 1:100 in the supplied buffer, and cells were incubated with the diluted probe overnight at 40 °C. The following day, the supplied pre-amplifier, amplifier, and label probe solutions diluted 1:25 in the relevant buffers were added to the motoneurons sequentially for 1 h each at 40 °C. Three washing steps were performed with the supplied wash buffer before each incubation and after the last one. Next, cells were washed twice with RNase-free DPBS and co-immunostained with antibodies against Ptbp2 (Proteintech, 55186-1-AP, 1:250 dilution) and Tubulin (Sigma-Aldrich, T5168, 1:500 dilution).

## Puromycylation

Motoneurons were grown for 7 DIV on laminin-111-coated glass coverslips. Cells were incubated with 10 μg/ml puromycin (Sigma-Aldrich, P8833) supplemented in the medium for 8 min at 37 °C in cell culture incubator. In negative control experiments, cells were pretreated with 100 μg/ml cycloheximide for 30 min before the addition of puromycin to the medium. Cells were washed twice with prewarmed HBSS and fixed for 15 min in PLP. After fixation, cells were washed and permeabilized for immunostaining or a proximity ligation assay (PLA) using antibodies against puromycin (Sigma-Aldrich, MABE343, 1:200 dilution), N-terminal of hnRNP R (Sigma-Aldrich, HPA026092, 1:200 dilution) and C-terminal of hnRNP R (Abcam, ab30930, 1:100 dilution).

## Proximity ligation assay (PLA)

PLA was carried out using the Duolink In Situ Orange Starter Kit Mouse/Rabbit (Sigma-Aldrich, DUO92102) according to the manufacturer's recommendations. Briefly, Motoneurons were grown for 7 DIV on laminin-111-coated glass coverslips and washed twice with DPBS. Cells were fixed in PLP for 10 min, then permeabilized. After permeabilization and washing, cells were blocked in blocking buffer 1 h at 37 °C and incubated with primary antibodies diluted in blocking buffer overnight at 4 °C (Supplementary Data 3). PLA probes were applied in 1:5 dilution for 1 h at 37 °C, followed by ligation, amplification, and label probe binding for 30 and 100 min respectively. Cells were fixed again for 10 min at room temperature in PLP, washed with DPBS, and processed further for immunohistochemistry and 4′,6-diamidino-2-phenylindole (DAPI) staining. Antibodies used and their dilutions are listed in Supplementary Data 3.

## Immunofluorescence staining

Motoneurons were cultured on laminin-111- and PORN-coated glass coverslips for 6 or 7 DIV. Cells were washed twice with DPBS and fixed with 4% PFA at room temperature for 15 min followed by permeabilization with 0.3% Triton X-100 at room temperature for 20 min. After three washes in DPBS, cells were blocked in a blocking buffer containing 4% BSA at room temperature for 1 h. Primary antibodies diluted in blocking solution were applied onto coverslips and incubated at 4 °C overnight followed by incubation with secondary antibodies (Supplementary Data 3) at room temperature for 1 h and counterstaining with DAPI. Alexa Fluor 546 phalloidin (Invitrogen, A22283) was added at 1:50 in PBS during incubation with secondary antibodies. Coverslips were washed and mounted using FluorSave Reagent (Merck, 345789) and subsequently imaged. Images were taken on an Olympus confocal microscope Fluoview 1000 confocal system at 60× magnification. Intensity and region-of-interest analyses were performed using ImageJ as part of the Fiji package[57]. Antibodies used and their dilutions are listed in Supplementary Data 3.

## Image acquisition and data analysis

Images were acquired on an Olympus Fluoview 1000 confocal system equipped with the following objectives: 10× (NA: 0.25), 20× (NA: 0.75), 40× (oil differential interference contrast, NA: 1.30), or 60× (oil differential interference contrast, NA: 1.35). Fluorescence excitation was achieved with using 405, 473, 559, and 633 nm lasers. Images were obtained with the corresponding Olympus FV10-ASW (RRID:SCR_014215) imaging software for visualization. The resulting images (Olympus.oib format) were processed by maximum intensity projection and were adjusted in brightness and contrast using Image J software.

For FISH, maximum intensity projections were created from 0.3 μm z-stacks. To measure the percentage of colocalization between Ptbp2 and *Hnrnpr* punctae, Ptbp2, and *Hnrnpr* punctae were identified in the cell body outside the DAPI-stained area and in 50 μm-long proximal axonal regions semi-automatically utilizing the ImageJ threshold and particle analysis plugin after background subtraction.

*Hnrnpr* punctae were quantified in the nucleus, soma and 50 μm-long proximal axonal regions of each neuron semi-automatically utilizing the ImageJ threshold and particle analysis plugin after background subtraction. The nucleus was defined as the DAPI-stained area. In the soma, all punctae located in the cell body outside the DAPI-stained area were quantified. Axons were defined as the processes with a length at least three times as that of dendrites[58].

For intensity measurements of hnRNP R-PLA signals in the somata, mean gray values of images were measured from unprocessed raw data after background subtraction using ImageJ software. In the 50 μm-long proximal axonal regions, the number of punctae was quantified semi-automatically using the ImageJ threshold and particle analysis plugin after background subtraction. For quantification of immunofluorescence signals of Synapsin1 and Piccolo in growth cones, raw images were projected using ImageJ and mean gray values were measured after background subtraction. Puromycin intensity was measured in somata and 20 μm-long proximal and distal regions of axons.

For axon length measurements, motoneurons transduced with lentiviruses were immunostained at DIV 2, 4, and 6 with anti-tau and anti-EGFP antibodies. The images were acquired with a Keyence BZ-8000K fluorescence microscope equipped with a standard color camera using a 20× 0.7-NA objective. The length of the longest axon branch was quantified using ImageJ software. Axon collaterals were not considered for the analysis. Motoneurons were only scored when designated axons were at least three times longer than the corresponding dendrites ensuring an unambiguous distinction between axons and dendrites[58].

For growth cone size analysis, cells were plated on laminin-221 for 6 DIV and stained against tau and phalloidin. The area of the growth cone was measured using ImageJ software. Images from control and *Ptbp2* knockdown motoneurons were acquired with identical settings (laser intensity and photomultiplier voltage).

## Subcellular fractionation

Subcellular fractionation was applied as previously described[59]. Briefly, NSC-34 cells were seeded on 10 cm dishes and grown to 80–90% confluency. Cells were washed once with ice-cold PBS and incubated in 4 ml lysis buffer A (10 mM HEPES pH 7.0, 100 mM KCl, 5 mM MgCl₂, 35 μg/ml digitonin) on ice for 10 min at 4 °C. The supernatant was collected and centrifuged at 2000 × *g* for 5 min at 4 °C. Following centrifugation, the supernatant was collected as the cytosolic fraction (Cyt). The cells remaining on the dish were washed once with ice-cold PBS and dissolved in 4 ml lysis buffer B (10 mM HEPES pH 7.0, 100 mM KCl, 5 mM MgCl₂, 0.5% NP-40). The lysate was collected, incubated on ice for 15 min, and centrifuged at 20,000 × *g* for 15 min at 4 °C. The supernatant containing organellar and nuclear luminal proteins was collected as the nuclear soluble fraction (Nuc + org). Chromatin-bound

proteins that were precipitated in the pellet were dissolved in a 4 ml lysis buffer B. The same volume of each fraction was used for Western blotting and qPCR. Antibodies used and their dilutions are listed in Supplementary Data 3.

For subcellular fractionation of primary motoneurons, cells were plated in laminin-111- and PORN-coated 6 cm dishes and cultured for 7 DIV. Cells were washed once with ice-cold PBS followed by incubation in 700 μl lysis buffer A containing 200 μg/ml digitonin per well for 10 min at 4 °C. The supernatant was transferred to a 1.5 ml tube and centrifuged at $2000 \times g$ for 5 min at 4 °C to obtain the Cyt fraction. The cells remaining were washed once with ice-cold PBS and lysed in 700 μl lysis buffer B. The lysate was transferred into a 1.5 ml tube and kept on ice for 15 min followed by centrifugation at $20,000 \times g$ for 15 min at 4 °C. The supernatant containing the Nuc+org fraction was transferred into a new tube and the pellet was dissolved in 700 μl lysis buffer B as the Chr fraction. The same volume of each fraction was used for Western blotting. Antibodies used and their dilutions are listed in Supplementary Data 3.

### Protein extraction and western blotting
Total protein was extracted from primary motoneurons and NSC-34 cells with RIPA buffer (50 mM Tris-HCl pH 7.4, 150 mM NaCl, 1% NP-40, 0.05% sodium deoxycholate, 0.1% SDS). Protein concentration was quantified using a BCA protein assay kit (Thermo Fisher Scientific, 23227). Equal amounts of proteins were size-separated by SDS-PAGE gel electrophoresis followed by transfer onto nitrocellulose membrane and immunoblotting with the indicated antibodies. Antibodies used and their dilutions are listed in Supplementary Data 3.

### Sucrose gradient fractionation
Approximately $2 \times 10^7$ primary motoneurons were plated in a laminin-111- and PORN-coated T-75 Flask and cultured for 7 DIV. Cells were pretreated with 100 μg/ml cycloheximide for 10 min at 37 °C before washing with ice-cold PBS containing 100 μg/ml cycloheximide. Cells were then lysed in polysome lysis buffer (20 mM Tris-HCl pH 7.4, 100 mM KCl, 5 mM MgCl₂, 1 mM Leupeptin, 1 mM Pepstatin, 1 mM Aprotinin, 0,1 mM PMSF, 0,1 mM AEBSF, 0.5% NP-40, 100 μg/ml cycloheximide, protease inhibitor, 40 U/ml RNAsin (Promega)). After 10 min incubation on ice, lysates were centrifuged at $10,000 \times g$ for 10 min at 4 °C. 20 μl of cleared lysate were kept as input lysate and the remaining sample were loaded onto 5–45% sucrose gradients in gradient buffer (20 mM Tris-HCl pH 7.5, 100 mM KCl, 5 mM MgCl₂) and centrifuged at 34,500 rpm for 2 h at 4 °C in a SW 41Ti swing-out rotor (Beckmann Coulter). Gradients were harvested with a Biocomb PGFip Piston Gradient Fractionator that continually monitored the optical density at 254 nm. Aliquots of the ribosomal fractions were used to extract total RNA with the NucleoSpin RNA kit (Macherey-Nagel) for further analysis. For western blot analysis, proteins were precipitated using methanol/chloroform precipitation[60]. Antibodies used and their dilutions are listed in Supplementary Data 3.

### Statistics and reproducibility
All statistical analyses were performed using GraphPad Prism version 9 for Windows (GraphPad Software, San Diego, California USA) and statistical significance was considered at test level $P < 0.05$. Quantitative data are presented as mean ± s.d. unless otherwise indicated. No statistical method was used to predetermine sample size. No data were excluded from the analyses. Two groups were compared using unpaired two-tailed student's $t$ test or two-tailed one-sample t-test. For multiple independent groups, one-way or two-way analysis of variance with post hoc multiple comparisons test was used. When the data did not adhere to a normal distribution, statistical analysis was performed using SuperPlots[61]. Most experiments were carried out independently at least three times and individual data points are presented in all graphs. Details of replicate numbers,

quantification and statistics for each experiment are specified in the figure legends.

### Reporting summary
Further information on research design is available in the Nature Portfolio Reporting Summary linked to this article.

## Data availability
Source data are provided with this paper. The raw mass spectrometry proteomics data have been deposited to the ProteomeXchange Consortium via the PRIDE[62] partner repository with the dataset identifier PXD035196. Previously published HITS-CLIP dataset for Ptbp2[11] are available at Gene Expression Omnibus (GEO) under accession code GSE47564. Source data are provided with this paper.

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

## Acknowledgements

This work was supported by the Deutsche Forschungsgemeinschaft [BR4910/1-2 (SPP1738) and BR4910/2–2 (SPP1935) to M.B., SE697/4-2 (SPP1738) and SE697/5–2 (SPP1935) to M.S., Fi573/15-2 (SPP1935) and Fi573/20-1 to U.F.] and a Grant from the Schilling Stiftung im Stifterverband

für die Deutsche Wissenschaft to M.S. This publication was supported by the Open Access Publication Fund of the University of Wuerzburg.

## Author contributions
S.S. performed the majority of the experiments, data analysis and statistical analysis. A.Z. contributed to several experiments and assisted with imaging as well as data analyses. G.P. performed the RNA pull-downs for proteomics and carried out the alternative polyadenylation analyses. J.B., F.M., and M.M. performed proteomics. C.S. and U.F. performed sucrose density gradient ultracentrifugation. S.S. wrote the manuscript. M.B. and M.S. reviewed and revised the manuscript. M.S. and M.B. supervised this study and provided financial support. All authors approved the final version of the manuscript.

## Funding

## Competing interests
The authors declare no competing interests.
