## [Peer Review File · Nature Communications]

Cytosolic Ptbp2 modulates axon growth in motoneurons through axonal localization and translation of HnrnpREVIEWER COMMENTS

Reviewer #1 (Remarks to the Author):

Ptbp2 is an RNA-binding protein with well-known nuclear roles in splicing regulation during neuronal development. In this manuscript, Salehi et al. describe new cytoplasmic roles for Ptbp2 in regulating the axonal growth of motor neurons. They show that Ptbp2-deficient neurons have reduced axonal lengths which can be rescued by Ptbp2 overexpression. The authors carry out a pulldown of biotinylated short 3' UTR of Hnrnpr and identify Ptbp2 as a strong interactor through mass spectrometry in NSC-34 cells (with motor-neuron-like characteristics). The authors also argue that Ptbp2 regulates the transport of Hnrnpr into axons and could show via FISH that Hnrnpr puncta are reduced in Ptbp2-deficient axons and can be rescued by overexpression of Ptbp2.

Using a Puro-PLA assay, the authors also demonstrate that local translation of hnRNP R in axons of motor neurons is reduced upon Ptbp2 knockdown. Furthermore, through a ribosomal pulldown assay, Salehi et al. show that Ptbp2 impacts HnRNP R translation via regulating Hnrnpr association with ribosomes. Ptbp2 depletion shifts Hnrnpr profile in a sucrose density gradient ultracentrifugation assay to lighter ribosomal fractions indicating reduced binding to polysomes. Moreover, the authors identify eIF5A2 as an interactor of Ptbp2 and a regulator of the axonal translation of Hnrnpr. Finally, they demonstrate that overexpression of HnRNP R can rescue axonal growth deficits in Ptbp2-deficient neurons, and that the opposite is not true, showing that Ptbp2 exerts its effects via regulation of HnRNP R. Overall this manuscript describes previously unknown roles of Ptbp2 in the cytoplasm and provides several lines of evidence that Ptbp2 exerts its effects via HnRNP R.

Major comments:

The levels of Ptbp2 seem to be reduced but not completely knocked down, moreover, the degree of depletion seems to be quite variable across the experiments which can compromise the conclusions made. For instance in figure 4 h and 5 h, the levels look very similar to the controls. I understand these experiments were done in primary mouse neurons where knocking down genes is challenging, but there are available Ptbp2-KO mouse models. Could the authors obtain some cross-validation from these models, perhaps in collaboration with other teams that have cultured primary neurons from knockout models (such as here <https://www.ncbi.nlm.nih.gov/pmc/articles/PMC3896118/>), or obtain some brain tissue from such a model for cross-validating the change in hnRNPR expression?

Given that Ptbp2's role in RNA processing is well-known, the authors should perform an experiment to exclude the option that reduction in HnRNP R protein observed was due to splicing or 3' end processing changes upon Ptbp2 knockdown that could for instance (but not limited to) increase the levels of the longer Hnrpnr isoform over the shorter "translated" isoform? Have the authors looked at the distribution of Hnrpnr isoforms in Ptbp2-deficient motor neurons? Similarly, is it possible that Ptbp2 indirectly impacts association of eIF5A2 with ribosomes by altering splicing of one of the factors in the translation initiation complex? To truly exclude any nuclear impact of Ptbp2, overexpressing a cytoplasmic-only Ptbp2 that lacks NLS in a Ptbp2-KO background would be a possibility.

Some reasoning would be helpful to understand the motive for investigating eIF5A2 and separating its role from eIF2a. It is at present unclear why the authors shifted focus from eIF2a to eIF5A2 from figure 5 to figure 6? They show in figure 5 that Ptbp2 binds to eIF2a in an RNA-independent manner.

Could the authors discuss in more detail their proposed order of events through which Ptbp2 regulates HnRNP R translation? If Ptbp2 binds to the 3' UTR of Hnrpnr, when and how does it bind to eIF5A2 and subsequently bring it to Hnrpnr? eIF5A2 was not present in the mass spectrometry data in figure 2 that was done in NSC-34 cells and using the 3' UTR of Hnrpnr alone, indicating that Ptbp2 does not bind to eIF5A2 directly when bound only to 3' UTR. When and how do the authors believe Ptbp2 regulates the binding of eIF5A2 to the ribosome?

Other comments:

From figure 1, Ptbp2-deficient neurons seem to have more neurites. Considering previously described roles of Ptbp2 in axonogenesis (Zhang et al. (2019) doi: 10.1016/j.neuron.2019.01.022), is it possible there is reduced axonal length due to a failure of neurite fate commitment to axons? Hence, could the authors speculate why overexpression of Ptbp2 does not increase the length of the axon further than WT?

Across all figures, it is unclear why there are no error bars in the control conditions. Are the authors scaling to the control per biological replicate? I believe it would aid analysis if they were to scale to the average of the controls instead. Additionally, across all figures, it is advisable to show the individual data points to clarify the spread of the data, boxplots alone are insufficient. This is sometimes done (figure 1 c) but not always (figure 1 e).

In figure 3 c, correlation of the number of puncta is not an appropriate comparison and does not indicate colocalisation. A better measure would be using the percentage of colocalised puncta. As an

additional control, it would be interesting to show Hnrnp2r colocalisation with another RBP that is not expected to bind to it, for example any RBP/protein enriched in the anti-sense mass spectrometry data from figure 2. And how is cytoplasm measured in f? In h and i, some of the groups lack boxes (especially knockdown), making it difficult to estimate the extent of change. It is not possible to judge if most dots are at 0 or close to the median. This could be clearer if the authors showed violin rather than box plots. It is also unclear how the authors are choosing axon vs dendrite in Ptbp2-deficient neurons. Finally, their conclusion suggests that only the transport into axons is affected and not into dendrites, but could this be a statistical limitation due to the lower number of measured dendrites? A violin plot could show more clearly if the distribution of puncta numbers decreases only for axons, and not dendrites.

In Figure 4, the reduction in PLA intensity in axons may not have to do with a reduction in local translation but rather with less Hnrnp2r puncta overall in the axons due to a transport deficit (as they showed earlier). To address this, it is possible to include the number of PLA puncta measured relative to the number of Hnrnp2r puncta observed via FISH. Moreover, in f, intensity alone is not an accurate measure. If the overall length of the axon is reduced, one expects the number (and hence intensity) of Hnrnp2r puncta to also be reduced. Intensity per unit area, or better, number of puncta per unit area might be a more relevant readout as it accounts for the reduced length. Considering HnRNP R is reduced by half in i, and part e of the figure shows there is no impact of Ptbp2 knockdown on HnRNP R translation in the soma, does that mean that half of the HnRNP R in neurons is present in the axons? Finally, the authors should show that hnRNP R (oe) has no effect on the efficiency of shPtbp2 knockdown.

Reviewer #2 (Remarks to the Author):

In the submitted article entitled “Ptbp2 modulates axon growth in motoneurons through axonal localization and translation of Hnrnp2r”, Salehi et al. investigate the cytoplasmic function of the RNA-binding protein Ptbp2, specifically in axons of cultured mouse motor neurons and NSC-34 cells. They show that Ptbp2 plays a role in axon outgrowth. Interestingly, Ptbp2 interacts with the mRNA of the RNA-binding protein hnRNP R and regulates its localization and translation in axons. Importantly, Ptbp2 is shown to directly interact with eIF5A, through which it facilitates translation of hnRNP R mRNA. Finally, the authors show that it is indeed this regulation of hnRNP R that impacts axonal growth.

Overall, Salehi et al. identify a novel cytoplasmic function of Ptbp2 in axonal localization and translational control and investigate the underlying mechanism in detail. The experiments are clearly described and properly designed to address the proposed research questions. The scientific background is sufficiently presented. The article is generally well written and clear. In the remainder, there are open questions and controls that need to be addressed as well as some text changes are suggested.

1. For the plotting of single cell microscopy data, on top of the provided information about the number of replicates, it would be informative how many cells were analyzed for a replicate. The authors may

want to consider plotting the average of each replicate and color-code the individual cells by individual replicates to clearly show the reproducibility between replicates, possibly as superplots (as suggested here <https://doi.org/10.1083/jcb.202001064>). For the same reason, the authors may want to consider to do the statistical analysis on the average of each replicate rather than the pooled cells.

2. The authors should consider providing a control for the specificity of their Hnrnpr FISH probes (e.g. by using the Hnrnpr^{-/-} culture or an antisense probe)? Additionally, a more detailed analysis of colocalization of Hnrnpr mRNA with Ptbp2 would be welcome. For instance, what is the percent of colocalization in soma vs. axon?

3. Could the authors make sure the breaks on the y-axis of Fig. 3h and i are depicted correctly? For instance, where does the top blue dot in Fig. 3i belong? Perhaps the authors would like to use a different y-axis scale (i.e. 10 as maximum) instead of breaking the axis. In Fig. 3h, are all orange dots outliers to a boxplot or is this a dotplot instead. In the second case, please unify the plot for all conditions (or use superplots as suggested above).

4. In Fig. 7a-c, the authors show that Ptbp2 regulates the association of Hnrnpr mRNA with ribosomes. This point would be solidified if the same result were obtained using the EGFP reporter with the deleted binding site in the Hnrnpr 3'UTR, to show the effect depends on the direct binding of Ptbp2.

5. For Fig. 8, could the authors show whether these findings are not only true for axon length but also for growth cones, as suggested in Fig. 1?

6. In the discussion, the authors elaborate on the possibility that Ptbp2 may influence synaptic maintenance in the axon. Could the author provide an initial experiment (e.g. immunostaining of synaptic markers) or at the least discuss whether this might be the case in Hnrnpr^{-/-} mice as well?

7. The methods section needs a general part on microscopy and image analysis. For instance, how were the FISH experiments imaged, how were the FISH spots quantified, how was the Puro-PLA intensity measured? This is only briefly mentioned for immunostainings.

8. The authors may want to consider increasing the brightness on some of their microscopy pictures to increase visibility, e.g. in Fig. 2a.

9. The order of panels is at times confusing. Perhaps some panels can be reorganized to follow a more logic pattern.

Additional issues / editing:

- Fig.1a: The authors should clearly state in the text that mRNA levels were investigated, and *Ptbp2* should consistently be written in italics when refereeing to mRNA. Moreover, could the authors specify the age or age-range that is considered to be adult?
- In rescue experiments, it is not clear from the figure label/legend that the over expression condition is shRNA treated as well (e.g. Fig. 1eg, Fig. 3e-l, Fig 7b). The plots would be more intuitive if this information were included. Moreover, could the authors specify if the co-expressed EGFP-*Ptbp2* used for rescue experiments was mutated to be resistant to sh*Ptbp2*?
- In Fig.3e-l, if *Ptbp2* mediates axonal *Hnrnpr* mRNA transport, would its depletion in the axon not result in the accumulation of the mRNA in other fractions? As *Hnrnpr* mRNA does not accumulate in other fractions while it is depleted from axons, would a likely interpretation of this data not be a transcription or stability effect rather than transport? This may be in line with the reduction of mRNA levels seen in Fig. 4g. Could the authors discuss this option? Also to clarify this point, could the authors label Fig.3f with “soma” instead of cytoplasm as indicated in the text, if this is what was analyzed.
- The statement in line 330 that the translation of *Hnrnpr* is dependent on eIF5A1/2 is not fully supported at that point. We suggest to move this statement to the end of the paragraph after the relevant experiments have been presented.
- Could the author comment whether there is any known or observed phenotype for the *Hnrnpr*^{-/-} mouse. Are there overlapping defects with *Ptbp2* mutant mice?
- We ask that the authors reevaluate their discussion about axonal co-transport, as co-transport of *Ptbp2* and *Hnrnpr* mRNA is not directly shown. The authors may want to simply phrase this as localization instead.

Reviewer #3 (Remarks to the Author):

The manuscript entitled 'Ptbp2 modulates axon growth in motoneurons through axonal localization and translation of Hnrnp-R' reports an axon-localized pool of the splicing factor PTBP2 in motor neurons and explores its function in this compartment.

The argument for PTBP2 to carry an axonal function is core to the study. Yet, the findings supporting the idea that the neuronal defect and hnrnp phenotype in Ptbp2 KD is due to lack of axonal pool are currently too weak.

A series of essential controls and additional experiments are needed to support the conclusions proposed. The authors show that PTBP2 is required for axonal growth in cultured motor neurons and that PTBP2 binds to hnrnp-R mRNA 3'UTR (unclear if this binding is nuclear or also cytoplasmic). They show that in addition to co-localize in the nucleus, hnrnp mRNA and PTBP2 protein also co-localize in axons (although good control for the immunostaining is needed, see specific comments below). They do not provide experiments going beyond a correlation. Nuclear PTBP2 binds to 3'UTR of many mRNAs and regulates a lot of mRNAs coding for RNA-binding proteins, all likely to be important for transport of mRNAs. The loss of hnrnp-R transcripts in axons in the KD could therefore be very indirect. Same indirect correlation is true for the binding of PTBP2 with ribosomes and eIF5A, lacking findings supporting its functional significance. The manuscript is potentially very important because it shows that PTBP2 is in the cytoplasm and axons and that in the cytoplasm, it can form a complex with the translation machinery. But is this binding meaningful or is the phenotype observed an indirect consequence of PTBP2's known role in splicing of essential cytoplasmic proteins important for transport and local translation? Finally, results shown in Figure 7 contradict the conclusions proposed by the authors.

All together this study needs a lot of additional data to provide a convincing set of evidence supporting the message promoted by the authors. I would recommend rethinking the study in depth; do a careful assessment of the nuclear and cytoplasmic PTBP2 variants and use variants (either natural or synthetic – ie, adding NES) to dissect the relative contribution of nuclear and cytoplasmic proteins in neuronal maturation.

Specific weaknesses:

- How quickly after shPTBP2 lentiviral treatment are the axons showing a defect in culture? The time of transfection/infection is unclear and imaging is done at DIV6. Published mouse mutant studies already showed that PTBP2 is required for neuronal survival during development and axonal integrity. Is this different? Different time of treatment and timelapse studies of neurons after treatment is required.
- There are at least 8 splice variants of PTBP2 in mouse. It is surprising that the Western is detecting only one band in neurons. Is it possible that the antibody is missing a few due to location of its epitope?
- Is the EGFP-Ptbp2 protein transported in axons? Often GFP-tagged proteins show modified transport. Images of GFP fluorescence in the 'rescued' neurons and quantification of GFP puncta in axons in Fig. 1d-g is required. This is a key control.

- Control of the specificity of immunostaining in neurons is required showing absence of reactivity in neurons carrying Ptbp2 deletion (in Fig. 3d, although nuclear signal is lower in shPtbp2 treated cells, cytoplasmic signal is still quite high). This can be done using CRISPR. If the deletion approach is too demanding, the authors need at least to compare GFP with anti-PTBP2 signals in the EGFP-PTBP2 expressing culture and ascertain co-localization.
- One possibility that would support the antibody staining results is that Ptbp2 mRNAs are transported in axons and locally translated. The authors need to perform a FISH against Ptbp2 mRNA as these RNA detection techniques are now routine.
- The data showing interaction between the hnRNP-R 3'UTR and PTBP2 are very solid. However, is this interaction happening in both nucleus and the axon? This needs to be checked doing IP on axonal extracts. Compartmentalized cultures are now routine in the field.
- In Fig 2f, IgG IP control shows a PTBP2 band too. Why is that? What is the reason for the PTBP2 bands to be different between NSC-34 and primary culture? How is the full Western looking (see question on variants above)?
- Fig 3 shows quite a lot of PTBP2 proteins left in the KD neurons, especially in the cytoplasm. Authors need to show whether rescue can be obtained with a cytoplasmic version of the PTBP2 protein (deletion of NLS and addition of NES element to the GFP construct and imaging of GFP localization as well as hnRNP-R mRNA).
- Fig 4 shows drop in proteins in the axons. As the authors show that the number of mRNA is reduced substantially in the KD, the drop in protein is not surprising. This does not mean 'that Ptbp2 is required for the local translation of hnRNP R in axons of motoneurons.'. This is a vast over-interpretation of the results. The observation that mRNA level is not as reduced as protein level can here again be very indirect as 'sick' neurons do show overall reduced translation. Only the experiment done later shows that overall polysomes are not generally affected in the KD.
- Binding of PTBP2 with ribosomes is very interesting but do not infer a functional requirement of this binding for translation. The authors need to be much more careful in their conclusions.
- The fact that PTBP2 KD affects more polysomes than monosomes suggests that it affects cell body translation rather than axonal (normal axons and synapses are richer in monosomes and the Schuman lab showed that monosomes translate mRNAs in these compartments). Ribosomal distribution needs to be measured in control and KD neurons.
- The co-precipitation of eIF5A with PTBP2 is expected as PTBP2 is binding to 3'UTR of ribosome-decorated mRNAs. Translation requires folding of the mRNA and direct interaction between cap proteins and 3'UTR complexes.
- If the authors are right and PTBP2 phenotype is not due so much to hnRNP-R mRNA reduction but rather caused by lack of translation of this mRNA, the rescue of Ptbp2 KD by overexpression of a hnRNP-R vector is contradicting the conclusion completely as this vector is producing transcripts that should not be properly translated in absence of PTBP2. Figure 7 contradicts completely the model proposed by the authors.

Response to reviewers manuscript NCOMMS-22-30890-T

Reviewer #1 (Remarks to the Author):

Ptbp2 is an RNA-binding protein with well-known nuclear roles in splicing regulation during neuronal development. In this manuscript, Salehi et al. describe new cytoplasmic roles for Ptbp2 in regulating the axonal growth of motor neurons. They show that Ptbp2-deficient neurons have reduced axonal lengths which can be rescued by Ptbp2 overexpression. The authors carry out a pulldown of biotinylated short 3' UTR of Hnrnpr and identify Ptbp2 as a strong interactor through mass spectrometry in NSC-34 cells (with motor-neuron-like characteristics). The authors also argue that Ptbp2 regulates the transport of Hnrnpr into axons and could show via FISH that Hnrnpr puncta are reduced in Ptbp2-deficient axons and can be rescued by overexpression of Ptbp2.

Using a Puro-PLA assay, the authors also demonstrate that local translation of hnRNP R in axons of motor neurons is reduced upon Ptbp2 knockdown. Furthermore, through a ribosomal pulldown assay, Salehi et al. show that Ptbp2 impacts HnRNP R translation via regulating Hnrnpr association with ribosomes. Ptbp2 depletion shifts Hnrnpr profile in a sucrose density gradient ultracentrifugation assay to lighter ribosomal fractions indicating reduced binding to polysomes. Moreover, the authors identify eIF5A2 as an interactor of Ptbp2 and a regulator of the axonal translation of Hnrnpr. Finally, they demonstrate that overexpression of HnRNP R can rescue axonal growth deficits in Ptbp2-deficient neurons, and that the opposite is not true, showing that Ptbp2 exerts its effects via regulation of HnRNP R. Overall this manuscript describes previously unknown roles of Ptbp2 in the cytoplasm and provides several lines of evidence that Ptbp2 exerts its effects via HnRNP R.

Author's response: We appreciate the reviewer's positive assessment of our manuscript and of the novelty of our findings regarding a cytosolic function of Ptbp2 in axon growth.

Major comments:

The levels of Ptbp2 seem to be reduced but not completely knocked down, moreover, the degree of depletion seems to be quite variable across the experiments which can compromise the conclusions made. For instance in figure 4 h and 5 h, the levels look very similar to the controls. I understand these experiments were done in primary mouse neurons where knocking down genes is challenging, but there are available Ptbp2-KO mouse models. Could the authors obtain some cross-validation from these models, perhaps in collaboration with other teams that have cultured primary neurons from knockout models (such as here), or obtain some brain tissue from such a model for cross-validating the change in hnRNPR expression?

Author's response: We followed up on the reviewer's suggestion by trying to obtain *Ptbp2* KO mouse models from different sources. However, in each case, the line was not available. Nevertheless, in order to solidify our results we designed an additional shRNA targeting the *Ptbp2* transcript in exon 4 (new Fig. 1b,c). Using this shRNA (termed sh2 in the revised manuscript), we confirmed some key results that were obtained with the shRNA (termed sh1 in the revised manuscript) targeting the 3' UTR of *Ptbp2*. We show that motoneurons transduced with sh2*Ptbp2* have shorter axons (new Fig. 1f), smaller growth cones (new Fig. 1h), reduced levels of hnRNP R protein but not *Hnrnpr* mRNA (new Fig. 5g,j,k) and reduced association of *Hnrnpr* mRNA with ribosomes (new Fig. 6a). These additional data with sh2*Ptbp2* together with our previous result that the effects of *Ptbp2* knockdown on axon growth

and hnRNP R levels can be rescued by re-expressing *Ptpb2* indicate that deficiency of *Ptpb2* is responsible for this phenotype.

Given that *Ptpb2*'s role in RNA processing is well-known, the authors should perform an experiment to exclude the option that reduction in HnRNP R protein observed was due to splicing or 3' end processing changes upon *Ptpb2* knockdown that could for instance (but not limited to) increase the levels of the longer *Hnrnpr* isoform over the shorter "translated" isoform? Have the authors looked at the distribution of *Hnrnpr* isoforms in *Ptpb2*-deficient motor neurons?

Author's response: We thank the reviewer for raising the point that *Ptpb2* deficiency might alter the alternative polyadenylation of *Hnrnpr* mRNA. To test this possibility, we performed qPCR for quantification of the three alternative *Hnrnpr* 3' UTR isoforms from *Ptpb2* knockdown motoneurons (new Fig. 3I). We did not observe any changes in the relative abundance of the 3' UTRs indicating that *Ptpb2* itself does not regulate *Hnrnpr* alternative polyadenylation. We updated the main text as following:

"To evaluate whether alternative polyadenylation of *Hnrnpr* is regulated by *Ptpb2*, we assessed the relative abundance of the three 3' UTR isoforms following *Ptpb2* knockdown in NSC-34 cells and

motoneurons. The levels of the three 3' UTR isoforms were unchanged indicating that Ptbp2 does not alter *Hnrnpr* alternative polyadenylation (Fig. 3I)."

Fig. 3I

Similarly, is it possible that Ptbp2 indirectly impacts association of eIF5A2 with ribosomes by altering splicing of one of the factors in the translation initiation complex?

Author's response: We thank the reviewer for pointing out this possibility. We scanned a published RNA-seq dataset of *Ptbp2* KO brain (Li et al. 2014 eLife, doi: 10.7554/eLife.01201) and found no indication for dysregulated splicing or reduced levels of transcripts encoding translation factors. We included this information in the Discussion of the revised manuscript as following:

"According on to a published RNA-seq dataset of *Ptbp2* KO brain¹¹, we found no indication for dysregulated splicing or reduced levels of transcripts encoding translation factors. Together with our finding that global polysome profiles are not altered by Ptbp2 deficiency, this indicates that the Ptbp2-dependent binding of eIF5A1/2 to ribosomes is not a consequence of dysregulated ribosome composition."

To truly exclude any nuclear impact of Ptbp2, overexpressing a cytoplasmic-only Ptbp2 that lacks NLS in a Ptbp2-KO background would be a possibility.

Author's response: We thank the reviewer for the excellent suggestion. We tested whether Ptbp2-depleted motoneurons can be rescued with a cytoplasmic Ptbp2 mutant by preparing a lentiviral *Ptbp2* knockdown construct that co-expresses an EGFP-Ptbp2-ΔNLS variant. Ptbp2 contains a NLS composed of two basic stretches (KR and KKFK) at its N-terminal end (Pina etv al. 2018 Biochemistry, doi: 10.1021/acs.biochem.8b00256). Deletion of these stretches has been shown to abolish the nuclear localization of Ptbp2 (Romanelli et al. 2002 FEBS J, doi: 10.1046/j.1432-1033.2002.02942.x). We found that EGFP-Ptbp2-ΔNLS could rescue the effect of *Ptbp2* knockdown on axon growth and growth cone size in motoneurons (new Fig. 2d-j). We updated the manuscript as following:

"To test whether the effect of *Ptbp2* knockdown on axon growth is mediated by the cytosolic function of Ptbp2, we prepared a lentiviral *Ptbp2* knockdown construct that co-expresses an EGFP-Ptbp2 NLS deletion mutant (Fig. 2d,e). Ptbp2 contains a NLS composed of two basic stretches (KR and KKFK) at its N-terminal end¹⁶. Deletion of these stretches has been shown to abolish the nuclear localization of

Ptbp2¹⁷. We confirmed the cytosolic localization of EGFP-Ptbp2-ΔNLS in transduced motoneurons by immunostaining (Fig. 2f). Expression of EGFP-Ptbp2-ΔNLS in *Ptbp2* knockdown motoneurons could rescue the defect in axon growth and growth cone size indicating that cytosolic but not nuclear Ptbp2 functions mediate its role in axon growth (Fig. 2g-j).”

Some reasoning would be helpful to understand the motive for investigating eIF5A2 and separating its role from eIF2a. It is at present unclear why the authors shifted focus from eIF2a to eIF5A2 from figure 5 to figure 6? They show in figure 5 that Ptbp2 binds to eIF2a in an RNA-independent manner.

Author’s response: We agree with the reviewer that this point needs further clarification. Our data show that *Hnrrnr* mRNA shifts from polysome to monosome fractions upon knockdown of *Ptbp2* (Fig. 6i in the revised manuscript). To further elucidate the mechanism behind this observation, we first tested whether depletion of *Ptbp2* affects the association of eIF2a with ribosomes by Y10b immunoprecipitation but could not see any change (new Supplementary Fig. 5c). This indicates that translation elongation rather than initiation might be affected by *Ptbp2* knockdown. We followed up on this possibility by inspecting a protein interaction database (BioGRID) and identified eIF5A2 as *Ptbp2* interactor. eIF5A2 has previously been shown to be associated with elongating ribosomes rather than

translation initiation complexes (doi: 10.1002/jcb.20658, doi: 10.1016/j.molcel.2017.03.003, doi: 10.1038/nature08034). Furthermore, eIF5A2 promotes synthesis of poly-proline-containing proteins, including hnRNP R, and thus we selected it for further analysis. To make this point more clear, we have adjusted the Results section as following:

“Our data show that *Hnrnpnr* mRNA shifts from polysome to monosome fractions upon knockdown of *Ptbp2*. To further elucidate the mechanism behind this observation, we first investigated whether *Ptbp2* binds to the initiation factor eIF2 α and whether eIF2 α binding to ribosomes is regulated by *Ptbp2*. Whilst eIF2 α was co-precipitated by anti-*Ptbp2* in an RNA-independent manner (Supplementary Fig. 5a,b in the revised manuscript), we did not observe any alteration in eIF2 α co-precipitation with Y10b upon *Ptbp2* knockdown (new Supplementary Fig. 5c). This indicates that *Ptbp2* associates with ribosomes already at the initiation stage, but does not affect protein production at this step.

To assess whether *Ptbp2* has a role in translation elongation, we inspected an interactome database²³ and identified the translation factor eIF5A2 as a PTBP2 interactor.”

Supplementary Fig. 5c

Could the authors discuss in more detail their proposed order of events through which *Ptbp2* regulates HnRNP R translation? If *Ptbp2* binds to the 3' UTR of *Hnrnpnr*, when and how does it bind to eIF5A2 and subsequently bring it to *Hnrnpnr*?

Author's response: We thank the reviewer for the comment. Our data show that knockdown of *Ptbp2* reduces the association between eIF5A1/2 and *Hnrnpnr* mRNA (Fig. 7g in the revised manuscript), between *Hnrnpnr* mRNA and ribosomes (Fig. 6a in revised manuscript) and between eIF5A1/2 and ribosomes (Fig. 7i-j in the revised manuscript). To test whether, vice versa, eIF5A1/2 regulates binding of *Ptbp2* to *Hnrnpnr* mRNA, we have knocked down *eIF5A2* and found no statistically significant difference in *Ptbp2* binding to *Hnrnpnr* (new Fig. 7g). Thus, *Ptbp2* association with *Hnrnpnr* occurs independent of eIF5A1/2. Since eIF5A1/2 associates with elongating polysomes (doi: 10.1002/jcb.20658, doi: 10.1016/j.molcel.2017.03.003, doi: 10.1038/nature08034), our data suggest that *Ptbp2* binds to the 3' UTR of *Hnrnpnr* and subsequently navigates it to translating ribosomes via eIF5A1/2. We have updated the manuscript and adjusted the schematic in new Fig. 7n as following:

“To test whether, vice versa, eIF5A1/2 regulates binding of *Ptbp2* to *Hnrnpnr* mRNA, we have knocked down *eIF5A2* and found no statistically significant difference in *Ptbp2* binding to *Hnrnpnr* (new Fig. 7g). Thus, *Ptbp2* association with *Hnrnpnr* occurs independent of eIF5A1/2.”

Fig.7g**Fig.7n**
eIF5A2 was not present in the mass spectrometry data in figure 2 that was done in NSC-34 cells and using the 3' UTR of *Hnrnpr* alone, indicating that Ptbp2 does not bind to eIF5A2 directly when bound only to 3' UTR.

Author's response: eIF5A2 was only identified at low abundance in the *Hnrnpr* 3' UTR interactome datasets and therefore removed for analysis. The most likely explanation for this result is that, for the pulldown experiments, only the *Hnrnpr* 3' UTR was used that does not associate with elongating ribosomes containing eIF5A1/2. We have included this information in the text as following:

"We also assessed our *Hnrnpr* 3' UTR protein interactome dataset from NSC-34 cells but were unable to identify eIF5A1/2. The most likely explanation for this result is that, for the pulldown experiments, only the *Hnrnpr* 3' UTR was used that does not associate with elongating ribosomes containing eIF5A1/2."

When and how do the authors believe Ptbp2 regulates the binding of eIF5A2 to the ribosome?

Author's response: Our data show that knockdown of *Ptbp2* reduces the association between eIF5A1/2 and ribosomes (Fig. 7i-j in the revised manuscript) and that Ptbp2 interacts with eIF5A1/2 in an RNA-independent manner (Fig. 7f in the revised manuscript). This suggests that Ptbp2 promotes eIF5A1/2 interaction with ribosomes and we have clarified this point in more detail as following:

"Thus, eIF5A1/2 is an interactor of Ptbp2 and associates with ribosomes in a Ptbp2-dependent manner."

Other comments:

From figure 1, Ptbp2-deficient neurons seem to have more neurites. Considering previously described roles of Ptbp2 in axonogenesis (Zhang et al. (2019) doi: 10.1016/j.neuron.2019.01.022), is it possible there is reduced axonal length due to a failure of neurite fate commitment to axons?

Author's response: We appreciate the reviewer's feedback. It does not seem to be the case that Ptbp2 causes disturbed neurite fate commitment, since axons were clearly detectable in Ptbp2 deficient motoneurons. Axons were defined as those processes that were at least three times longer than the

corresponding dendrites ensuring an unambiguous distinction between axons and dendrites. We have adjusted the Methods section explaining our analysis in more detail in the section “Image acquisition and data analysis” as following:

“For axon length measurements, motoneurons transduced with lentiviruses were immunostained at DIV 2, 4 and 6 with anti-Tau and anti-EGFP antibodies. The images were acquired with a Keyence BZ-8000K fluorescence microscope equipped with a standard colour camera using a 20× 0.7-NA objective. The length of the longest axon branch was quantified using ImageJ software. Axon collaterals were not considered for the analysis. Motoneurons were only scored when designated axons were at least three times longer than the corresponding dendrites ensuring an unambiguous distinction between axons and dendrites.”

Hence, could the authors speculate why overexpression of Ptpb2 does not increase the length of the axon further than WT?

Author’s response: We thank the reviewer for pointing this out and helping us to be more specific about our experimental setup. Our overexpression experiment is actually a rescue of the *Ptpb2* knockdown by re-expression of Ptpb2 and not an overexpression of Ptpb2 in wildtype motoneurons. The levels of Ptpb2 expression in the sh1*Ptpb2*+EGFP-Ptpb2 condition were not higher than in WT motoneurons (See the Western blot in Fig. 5H). To make this important point clearer, we changed the label of the graphs to indicate that Ptpb2 was expressed in *Ptpb2* knockdown motoneurons as following: “sh1*Ptpb2*+EGFP-Ptpb2” (Fig. 1e,f,g,h; Fig. 4g,h and Fig. 5h,i in the revised manuscript). Similarly, we changed the labels for the hnRNP R rescue experiment to “sh1*Ptpb2*+EGFP-hnRNP R” (Fig. 8b,c,f-h in the revised manuscript).

Fig. 4g

Fig. 4h

Fig. 5h

Fig. 5i

Fig. 8b

Fig. 8f

Fig. 8c

Fig. 8g

Fig. 8h

Across all figures, it is unclear why there are no error bars in the control conditions. Are the authors scaling to the control per biological replicate? I believe it would aid analysis if they were to scale to the average of the controls instead.

Author's response: For those graphs not containing error bars for the control condition (Fig. 1c; Fig.3k; Fig. 5f,h; Fig. 6a-c,o; Fig. 7e,g,h,j,l,m; Supplementary Fig. 2g,h and Supplementary Fig. 4a,c in the revised manuscript), experimental replicates were performed on different days such that the absolute measured values (for example, band intensities of Western blots) were different across experiments and thus could not be directly compared between replicates. For this reason, we decided to normalize experiments separately by setting the control condition to 1 in each case. Accordingly, we used a one-sample t-test for the statistical analysis.

Additionally, across all figures, it is advisable to show the individual data points to clarify the spread of the data, boxplots alone are insufficient. This is sometimes done (figure 1 c) but not always (figure 1 e).

Author's response: We thank the reviewer for the suggestion. We have now used SuperPlots across all figures to clarify the spread of the data (please also see comment by reviewer #2) (Fig. 1f,h; Fig. 4h; Fig. 5f and Fig. 8c,g in the revised manuscript). We also performed statistical analysis on the average of the replicates rather than pooled cells.

Fig. 1f

Fig. 1h

Fig. 4h

Fig. 5f

Fig. 8c

Fig. 8g

In figure 3 c, correlation of the number of puncta is not an appropriate comparison and does not indicate colocalisation. A better measure would be using the percentage of colocalised puncta.

Author's response: As suggested by the reviewer, we re-analyzed the *Hnrnpr* FISH and Ptpb2 immunostaining data and quantified the percentage of colocalized punctae in the cytoplasm of the somata and axons. We observed that ~60% of Ptpb2-positive punctae also contained an *Hnrnpr* FISH signal showing that Ptpb2 is present at the same position as *Hnrnpr*. Strikingly, nearly all *Hnrnpr*-positive punctae also immunostained for Ptpb2 highlighting the importance of Ptpb2 for post-transcriptional regulation of *Hnrnpr* mRNA. These data have been inserted as new Fig. 4c.

As an additional control, it would be interesting to show Hnrnpr colocalisation with another RBP that is not expected to bind to it, for example any RBP/protein enriched in the anti-sense mass spectrometry data from figure 2.

Author's response: We thank the reviewer for the suggestion. We have selected hnRNP A2/B1 as control protein enriched in the proteomics data for the anti-sense (control) 3' UTR and performed immunostaining together with *Hnrnpr* FISH (new Supplementary Fig. 3b). We did not observe any cytosolic *Hnrnpr* FISH punctae that were positive for hnRNP A2/B1. We updated the manuscript as following:

"In contrast, we did not observe any cytosolic *Hnrnpr* FISH punctae (0 out of 180 punctae) that were positive for hnRNP A2/B1, a protein that was enriched for the anti-sense 3' UTR (Supplementary Fig. 3b)."

Supplementary Fig. 3b

And how is cytoplasm measured in f?

Author's response: For cytoplasm, all punctae located in the cell body outside the DAPI-stained area were quantified. We added this information in the Methods section "Image acquisition and data analysis" of the revised manuscript as following:

"*Hnrnpr* punctae were quantified in the nucleus, soma and 50 μm -long proximal axonal regions of each neuron semi-automatically utilizing the ImageJ threshold and particle analysis plugin after background subtraction. The nucleus was defined as the DAPI-stained area. In the soma, all punctae located in the cell body outside the DAPI-stained area were quantified."

In h and i, some of the groups lack boxes (especially knockdown), making it difficult to estimate the extent of change. It is not possible to judge if most dots are at 0 or close to the median. This could be clearer if the authors showed violin rather than box plots. **Author's response:** We thank the reviewer for the suggestion. We have re-plotted the figures as SuperPlots (please also see our response to reviewer #2). For these plots, statistical analysis was performed on the average of the replicates rather than pooled cells. We have updated the figure (Fig. 4h in the revised manuscript).

Fig. 4h

It is also unclear how the authors are choosing axon vs dendrite in Ptbp2-deficient neurons.

Author's response: We selected axons based on their length, which is at least three times as long as that of dendrites (Jablonka et al. 2007 J Cell Biol, doi: 10.1083/jcb.200703187). We included this information in the Methods section "Image acquisition and data analysis" of the revised manuscript as following:

"For axon length measurements, motoneurons transduced with lentiviruses were immunostained at DIV 2, 4 and 6 with anti-Tau and anti-EGFP antibodies. The images were acquired with a Keyence BZ-8000K fluorescence microscope equipped with a standard colour camera using a 20× 0.7-NA objective. The length of the longest axon branch was quantified using ImageJ software. Axon collaterals were not considered for the analysis. Motoneurons were only scored when designated axons were at least three times longer than the corresponding dendrites ensuring an unambiguous distinction between axons and dendrites⁶⁰."

Finally, their conclusion suggests that only the transport into axons is affected and not into dendrites, but could this be a statistical limitation due to the lower number of measured dendrites? A violin plot could show more clearly if the distribution of puncta numbers decreases only for axons, and not dendrites.

Author's response: We addressed the reviewer's comment in two ways. First, we now include the number of dendrites used for the statistical analysis in the figure legend. The number of dendrites analyzed is in the same range as the number of axons. Second, we plotted the number of punctae as SuperPlots. Our data show that the number of *Hnrnp*-positive punctae in dendrites is much lower compared to axons, explaining the statistical limitation mentioned by the reviewer. Thus, the low amounts of *Hnrnp* in dendrites makes it difficult to draw conclusions on dendritic transport of *Hnrnp*.

In Figure 4, the reduction in PLA intensity in axons may not have to do with a reduction in local translation but rather with less Hnrnp puncta overall in the axons due to a transport deficit (as they showed earlier). To address this, it is possible to include the number of PLA puncta measured relative to the number of Hnrnp puncta observed via FISH.

Author's response: We have followed the reviewer's suggestion and quantified the number of Puro-PLA punctae per axon length rather than the Puro-PLA intensity (Fig. 5f in the revised manuscript). We agree that reduced axonal *Hnrnp* mRNA levels as measured by FISH can contribute to the reduced

hnRNP R Puro-PLA signal we observed in axons of *Ptbp2*-depleted motoneurons. However, we think that a direct comparison between the number of FISH and Puro-PLA punctae might not be meaningful as each technique has a different stringency of detection. Particularly, the FISH technique we are using is highly specific, only giving a signal when multiple FISH probes are annealed side by side on the target RNA. Therefore, we performed additional experiments and measured the level of *Hnrnpr* mRNA in the somatodendritic and axonal compartments of control and *Ptbp2*-depleted motoneurons cultured in microfluidic chambers (Fig. 4i in the revised manuscript). Axonal levels of *Hnrnpr* mRNA were reduced by ~44% in *Ptbp2* knockdown compared to control motoneurons. This reduction in axonal *Hnrnpr* mRNA is less than that observed by Puro-PLA (~64%). Additionally, there was no reduction or a tendency to increased level of *Hnrnpr* mRNA levels in the somatodendritic compartment (Fig. 4i in the revised manuscript) or in whole *Ptbp2*-depleted cells (Fig. 5g in the revised manuscript), while we observed a significant reduction of hnRNP R Puro-PLA signal intensity also in the somata of *Ptbp2*-depleted motoneurons. Thus, our data indicate that *Ptbp2* regulates the axonal translation of *Hnrnpr* mRNA in addition to its axonal localization. We adjusted the main text and included the possibility that the *Ptbp2*-dependent *Hnrnpr* mRNA transport might contribute to its effects of local translation in axons as following:

“Additionally, while our Puro-PLA data indicate reduced axonal translation of hnRNP R in *Ptbp2* knockdown motoneurons, we cannot rule out the possibility that the reduced axonal localization of *Hnrnpr* mRNA we observed in *Ptbp2*-deficient axons contributes to this effect. However, according to our qPCR data from compartmentalized motoneurons, the reduction at the *Hnrnpr* mRNA level is less than that observed by hnRNP R Puro-PLA in axons of *Ptbp2*-depleted motoneurons. Additionally, we also observed reduced hnRNP R translation in the somata of *Ptbp2* knockdown motoneurons. Since total *Hnrnpr* mRNA levels are not affected by *Ptbp2* knockdown, this finding supports the notion that *Ptbp2* regulates axonal hnRNP R protein levels by promoting its local translation in addition to modulating its axonal mRNA localization.”

Fig. 5f

Fig. 4i

Fig. 5g

Moreover, in f, intensity alone is not an accurate measure. If the overall length of the axon is reduced, one expects the number (and hence intensity) of *Hnrnpr* puncta to also be reduced. Intensity per unit area, or better, number of puncta per unit area might be a more relevant readout as it accounts for the reduced length.

Author's response: We agree with the reviewer and have quantified the number of Puro-PLA punctae per 50 μ m-long proximal axonal regions of each neuron (new Fig. 5f). Following re-analysis, we observed a significant reduction in the number of axonal Puro-PLA punctae in *Ptpb2* knockdown motoneurons:

Fig. 5f

Considering hnRNP R is reduced by half in i, and part e of the figure shows there is no impact of Ptbp2 knockdown on hnRNP R translation in the soma, does that mean that half of the hnRNP R in neurons is present in the axons?

Author's response: The reviewer raises an important point that overall steady-state levels of hnRNP R are reduced on Western blots (Fig. 5g,h in the revised manuscript) whereas we previously found hnRNP R synthesis affected only in axons of Ptbp2 knockdown motoneurons. However, we re-analyzed the data as SuperPlots as requested by reviewer #2 and, as a result, now found a significant reduction in Puro-PLA intensity also in the soma, which, however, was much less pronounced compared to axons (Fig. 5e in the revised manuscript). Nevertheless, this newly analysed dataset shows that Ptbp2 indeed has also a role for hnRNP R translation in the soma, explaining the reduction in hnRNP R levels upon Ptbp2 knockdown. We updated the figure and text as following:

“We observed that the number of hnRNP R Puro-PLA punctae was significantly reduced in axons of Ptbp2 knockdown motoneurons compared to controls (Fig. 5e,f). We also observed a reduction in hnRNP R Puro-PLA signal intensity in the somata of Ptbp2-depleted motoneurons, which, however, was much less pronounced compared to axons (Fig. 5e,f).”

Fig. 5f

Finally, the authors should show that hnRNP R (oe) has no effect on the efficiency of shPtbp2 knockdown.

Author's response: We thank the reviewer for suggesting this important control. The lentiviral construct for expressing hnRNP R and sh1*Ptbp2* simultaneously is very large (~14.6 kbp) such that the efficiency of viral transduction is much lower compared to the construct only expressing sh1*Ptbp2*. For this reason, monitoring the *Ptbp2* knockdown efficiency upon expression of exogenous hnRNP R is not possible by Western blotting of transduced cultures. Instead, we performed *Ptbp2* immunostaining to show that *Ptbp2* levels are reduced in sh1*Ptbp2*-containing motoneurons expressing exogenous hnRNP R (new Fig. 8a).

Fig. 8a

Reviewer #2 (Remarks to the Author):

In the submitted article entitled “Ptbp2 modulates axon growth in motoneurons through axonal localization and translation of Hnrnp2”, Salehi et al. investigate the cytoplasmic function of the RNA-binding protein Ptbp2, specifically in axons of cultured mouse motor neurons and NSC-34 cells. They show that Ptbp2 plays a role in axon outgrowth. Interestingly, Ptbp2 interacts with the mRNA of the RNA-binding protein hnRNP R and regulates its localization and translation in axons. Importantly, Ptbp2 is shown to directly interact with eIF5A, through which it facilitates translation of hnRNP R mRNA. Finally, the authors show that it is indeed this regulation of hnRNP R that impacts axonal growth. Overall, Salehi et al. identify a novel cytoplasmic function of Ptbp2 in axonal localization and translational control and investigate the underlying mechanism in detail. The experiments are clearly described and properly designed to address the proposed research questions. The scientific background is sufficiently presented. The article is generally well written and clear. In the remainder, there are open questions and controls that need to be addressed as well as some text changes are suggested.

Author’s response: We thank the reviewer for appreciating the novelty of our findings. To further substantiate our results, we performed a series of new experiments according to the reviewer’s suggestion.

1. For the plotting of single cell microscopy data, on top of the provided information about the number of replicates, it would be informative how many cells were analyzed for a replicate. The authors may want to consider plotting the average of each replicate and color-code the individual cells by individual replicates to clearly show the reproducibility between replicates, possibly as superplots (as suggested here <https://doi.org/10.1083/jcb.202001064>). For the same reason, the authors may want to consider to do the statistical analysis on the average of each replicate rather than the pooled cells.

Author’s response: We thank the reviewer for this suggestion to present our data in a more precise format. We followed this advice and plotted single cell microscopy data as SuperPlots according to the reference provided by the reviewer. However, rather than colour-coding individual cells and representing them as scatter plots (see below for individual replicates), we chose to plot data distribution as Violin plots for better visibility, as is also recommended in the reference provided by the reviewer. Additionally, We have now performed statistical analysis on the average of the replicates rather than pooled cells (Fig. 1f,h; Fig. 4h; Fig. 5f and Fig. 8c,g in the revised manuscript).

Data of Fig. 1f

Fig. 1f

Fig. 1h

Fig. 4h

Fig. 5f

Fig. 8c

Fig. 8g

2. The authors should consider providing a control for the specificity of their *Hnrnpr* FISH probes (e.g. by using the *Hnrnpr*^{-/-} culture or an antisense probe)?

Author's response: We followed the reviewer's guidance and checked the specificity of the *Hnrnpr* FISH probe on motoneurons cultured from *Hnrnpr*^{-/-} mice (new Supplementary Fig. 3a). We observed a complete loss of the FISH signal in *Hnrnpr*^{-/-} motoneurons indicating the specificity of the probe. We updated the manuscript as following:

"For the *Hnrnpr* probe, we observed a punctate FISH signal in the nucleus as well as in the cytoplasm and in axons of motoneurons, which was absent in motoneurons cultured from *Hnrnpr*^{-/-} mice, indicating the specificity of the *Hnrnpr* probe (Fig. 4a and Supplementary Fig. 3a)."

Supplementary Fig. 3a

Additionally, a more detailed analysis of colocalization of *Hnrnpr* mRNA with *Ptbp2* would be welcome. For instance, what is the percent of colocalization in soma vs. axon?

Author's response: We thank the reviewer for pointing this out. We re-analyzed our *Hnrnpr* FISH and *Ptbp2* immunostaining data and quantified the percentage of co-localized punctae in the cytoplasm of motoneurons. We observed that ~60% of *Ptbp2*-positive punctae also contained an *Hnrnpr* FISH signal showing that *Ptbp2* is present in excess of *Hnrnpr*. Strikingly, nearly all *Hnrnpr*-positive punctae also immunostained for *Ptbp2* highlighting the importance of *Ptbp2* for post-transcriptional regulation of *Hnrnpr* mRNA. These new data have been inserted as Fig. 4c.

Fig. 4c
3. Could the authors make sure the breaks on the y-axis of Fig. 3h and i are depicted correctly? For instance, where does the top blue dot in Fig. 3i belong? Perhaps the authors would like to use a different y-axis scale (i.e. 10 as maximum) instead of braking the axis. In Fig. 3h, are all orange dots outliers to a boxplot or is this a dotplot instead. In the second case, please unify the plot for all conditions (or use superplots as suggested above).

Author's response: We followed the reviewer suggestion. We removed the breaks of the y-axes in Fig. 4h in the revised manuscript. For better visualization, we used SuperPlots.

Fig. 4h
4. In Fig. 7a-c, the authors show that *Ptbp2* regulates the association of *Hnrnp1r* mRNA with ribosomes. This point would be solidified if the same result were obtained using the EGFP reporter with the deleted binding site in the *Hnrnp1r* 3' UTR, to show the effect depends on the direct binding of *Ptbp2*.

Author's response: We thank the reviewer for this excellent suggestion. We repeated the ribosomal RNA IP using the Y10b antibody on motoneurons transduced with the EGFP-FL and EGFP- Δ PBS *Hnrnp1r* 3' UTR constructs, the latter lacking the *Ptbp2* binding site (new Fig. 6b). We observed that deletion of the *Ptbp2* binding site strongly reduced the association of the EGFP mRNA with ribosomes. We updated the main text as following:

“To substantiate this finding, we transduced motoneurons with the EGFP-FL and Δ PBS *Hnrnp1r* 3' UTR lentiviral constructs and assessed EGFP mRNA co-purification following ribosome immunoprecipitation with Y10b. We observed that co-purification of EGFP mRNA containing the *Hnrnp1r* Δ PBS 3' UTR was reduced compared to the FL 3' UTR, further demonstrating that *Ptbp2* binding mediates the association of *Hnrnp1r* with ribosomes (Fig. 6b).”

Fig. 6b

5. For Fig. 8, could the authors show whether these findings are not only true for axon length but also for growth cones, as suggested in Fig. 1?

Author's response: We followed the reviewer's advice and quantified the growth cone area of *Ptbp2* knockdown motoneurons expressing exogenous hnRNP R (*shPtbp2*+EGFP-hnRNP R) (new Fig. 8e,d). We observed the hnRNP R expression could rescue the reduced growth cone size induced by *Ptbp2* knockdown. We updated the main text as following:

"We found that re-expressing hnRNP R could restore axon length and growth cone area in *Ptbp2*-depleted motoneurons (Fig. 8d,e)."

Fig. 8d

Fig. 8e

6. In the discussion, the authors elaborate on the possibility that *Ptbp2* may influence synaptic maintenance in the axon. Could the author provide an initial experiment (e.g. immunostaining of synaptic markers) or at the least discuss whether this might be the case in *Hnrnp1-/-* mice as well?

Author's response: We performed immunostaining for the early presynaptic marker protein Synapsin 1 and the late synaptic marker Piccolo (*Pclo*) in *Ptbp2* knockdown motoneurons to assess their synaptic maturation. Interestingly, we observed reduction of *Pclo* but not Synapsin 1 in *Ptbp2*-depleted growth cones. *Pclo* is involved in presynaptic actin assembly at late stages of development whereas Synapsin 1 is a marker of synaptic vesicles that is already found in axon terminals of motoneurons at very early stages of development. Thus, *Ptbp2* appears to be involved in the late steps of axon terminal maturation. These additional data were inserted in (Fig. 1i-l). We updated the main text as following:

“To investigate growth cone maturation of Ptpb2 knockdown motoneurons, we performed immunostaining for the early presynaptic marker protein Synapsin 1 and the late synaptic marker Piccolo (Pclo). Interestingly, we observed reduction of Pclo but not Synapsin 1 in Ptpb2-depleted growth cones. Pclo is involved in presynaptic actin assembly at late stages of development^{15,16} whereas Synapsin 1 is a marker of synaptic vesicles that is already found in axon terminals of motoneurons at very early stages of development¹⁷. Thus, Ptpb2 appears to be involved in the late steps of axon terminal maturation.”

7. The methods section needs a general part on microscopy and image analysis. For instance, how were the FISH experiments imaged, how were the FISH spots quantified, how was the Puro-PLA intensity measured? This is only briefly mentioned for immunostainings.

Author’s response: We thank the reviewer for the comment. We prepared a separate Methods section for microscopy and image analysis:

“Image acquisition and data analysis

Images were acquired on an Olympus Fluoview 1000 confocal system equipped with the following objectives: 10× (NA: 0.25), 20× (NA: 0.75), 40× (oil differential interference contrast, NA: 1.30), or 60× (oil differential interference contrast, NA: 1.35). Fluorescence excitation was achieved with using 405,

473, 559, and 633 nm lasers. Images were obtained with the corresponding Olympus FV10-ASW (RRID:SCR_014215) imaging software for visualization. The resulting images (Olympus .oib format) were processed using ImageJ software.

For FISH, maximum intensity projections were created from 0.3 μm z-stacks. To measure the percentage of colocalization between *Ptbp2* and *Hnrnp1* punctae, *Ptbp2* and *Hnrnp1* punctae were identified in the cell body outside the DAPI-stained area and in 50 μm -long proximal axonal regions semi-automatically utilizing the ImageJ threshold and particle analysis plugin after background subtraction.

Hnrnp1 punctae were quantified in the nucleus, soma and 50 μm -long proximal axonal regions of each neuron semi-automatically utilizing the ImageJ threshold and particle analysis plugin after background subtraction. The nucleus was defined as the DAPI-stained area. In the soma, all punctae located in the cell body outside the DAPI-stained area were quantified. Axons were defined as the processes with a length at least three times as that of dendrites.

For intensity measurements of hnRNP R-PLA signals in the somata, mean gray values of images were measured from unprocessed raw data after background subtraction using ImageJ software. In the 50 μm -long proximal axonal regions, the number of punctae was quantified semi-automatically using the ImageJ threshold and particle analysis plugin after background subtraction. For quantification of immunofluorescence signals of Synapsin1 and Piccolo in growth cones, raw images were projected using ImageJ and mean grey values were measured after background subtraction. Puromycin intensity was measured in somata and 20 μm -long proximal and distal regions of axons.

For axon length measurements, motoneurons transduced with lentiviruses were immunostained at DIV 2, 4 and 6 with anti-tau and anti-EGFP antibodies. The images were acquired with a Keyence BZ-8000K fluorescence microscope equipped with a standard colour camera using a 20 \times 0.7-NA objective. The length of the longest axon branch was quantified using ImageJ software. Axon collaterals were not considered for the analysis. Motoneurons were only scored when designated axons were at least three times longer than the corresponding dendrites ensuring an unambiguous distinction between axons and dendrites.

For growth cone size analysis, cells were plated on laminin-221 for 6 DIV and stained against tau and phalloidin. The area of the growth cone was measured using ImageJ software.

Images from control and *Ptbp2* knockdown motoneurons were acquired with identical settings (laser intensity and photomultiplier voltage)."

8. The authors may want to consider increasing the brightness on some of their microscopy pictures to increase visibility, e.g. in Fig.2a.

Author's response: We have originally avoided to make changes to brightness and to present our data as best as possible in the original form. However, we have now adjusted brightness and contrast for some of our new imaging data (for example, new Supplementary Fig. 1c) to improve the visibility of axonal stainings.

9. The order of panels is at times confusing. Perhaps some panels can be reorganized to follow a more logic pattern.

Author's response: We followed the reviewer suggestion and re-organized several panels. For example, we split Fig. 1 into two figures, the first one (Fig. 1 in the revised manuscript) showing the role of *Ptbp2* for axon growth and the second one (Fig. 2 in the revised manuscript) showing the cytosolic localization of *Ptbp2*. We also re-arranged Fig. 5-8 to present the findings in a more logical and sequential manner.

Additional issues / editing:

- Fig.1a: The authors should clearly state in the text that mRNA levels were investigated, and *Ptbp2* should consistently be written in italics when refereeing to mRNA.

Author's response: We thank the reviewer for pointing this out. We changed the text and labelled *Ptbp2* in italics throughout when referring to the mRNA.

Moreover, could the authors specify the age or age-range that is considered to be adult?

Author's response: For the adult stage, we used samples from 6-12 weeks of age and we included this information now in the figure legend.

- In rescue experiments, it is not clear from the figure label/legend that the over expression condition is shRNA treated as well (e.g. Fig. 1e,g, Fig. 3e-l, Fig 7b). The plots would be more intuitive if this information were included. Moreover, could the authors specify if the co-expressed EGFP-*Ptbp2* used for rescue experiments was mutated to be resistant to sh*Ptbp2*?

Author's response: We thank the reviewer for pointing this out. We changed the label of the graphs to indicate that *Ptbp2* was co-expressed with the shRNA targeting *Ptbp2* as following: “sh1*Ptbp2*+EGFP-*Ptbp2*” (Fig. 1e,f,g,h; Fig. 4g,h and Fig. 5h,i in the revised manuscript). Similarly, we changed the labels for the hnRNP R rescue experiment to “sh1*Ptbp2*+EGFP-hnRNP R” (Fig. 8b,c,f-h in the revised manuscript).

Fig. 1e

Fig. 1f

Fig. 1g

Fig. 1h

Fig. 4g

Fig. 4h

Fig. 5h

Fig. 5i

Fig. 8b

Fig. 8f

Fig. 8c

Fig. 8g

Fig. 8h

- In Fig.3e-l, if *Ptbp2* mediates axonal *Hnrnpr* mRNA transport, would its depletion in the axon not result in the accumulation of the mRNA in other fractions? As *Hnrnpr* mRNA does not accumulate in

other fractions while it is depleted from axons, would a likely interpretation of this data not be a transcription or stability effect rather than transport? This may be in line with the reduction of mRNA levels seen in Fig. 4g. Could the authors discuss this option?

Author's response: The volume of the axon is very small relative to the volume of the soma such that *Hnrnpr* mRNA accumulation in the soma due to failed transport might be difficult to detect.

The reviewer also mentions that the *Hnrnpr* mRNA levels are mildly reduced in *Ptbp2* knockdown cells, as we have shown in the original Fig. 4g. We have now performed additional qPCR measurements of *Hnrnpr* levels in motoneurons depleted of *Ptbp2* using the same shRNA (sh1) as well as using a new shRNA (sh2) and observed that *Hnrnpr* levels are not changed upon *Ptbp2* knockdown (Fig. 5g in the revised manuscript). Thus, there is no general defect in *Hnrnpr* mRNA stability in *Ptbp2*-deficient motoneurons. However, it is still possible that *Ptbp2* regulates *Hnrnpr* stability in axons and we included this possibility in the Discussion as following:

“It remains to be determined in future studies whether the reduced axonal levels of *Hnrnpr* mRNA in *Ptbp2*-depleted motoneurons are a consequence of defective transport or reduced mRNA stability or both.”

Fig. 5g

Also to clarify this point, could the authors label Fig.3f with “soma” instead of cytoplasm as indicated in the text, if this is what was analyzed.

Author's response: For the figure mentioned by the reviewer (Fig. 4h in the revised manuscript), we counted all punctae in the cytoplasm of the soma (outside the nucleus). Therefore, we kept the labelling as “cytoplasm” but clarified the details of measurement in the Methods section “Image acquisition and data analysis” of the revised manuscript.

- The statement in line 330 that the translation of *Hnrnpr* is dependent on eIF5A1/2 is not fully supported at that point. We suggest to move this statement to the end of the paragraph after the relevant experiments have been presented.

Author's response: We deleted the statement mentioned by the reviewer at this position as we already have a similar sentence at the end of the final paragraph of this Results section.

- Could the author comment whether there is any known or observed phenotype for the *Hnrnpr*^{-/-} mouse. Are there overlapping defects with *Ptbp2* mutant mice?

Author's response: We are currently preparing a manuscript in which we provide a detailed description of the phenotype of *Hnrnpr*^{-/-} mice (Zare et al. in preparation). For this reason, we would prefer to omit this dataset in our current manuscript. We observed that *Hnrnpr*^{-/-} mice exhibit motor defects and motoneurons cultured from *Hnrnpr*^{-/-} also show axon growth defects.

- We ask that the authors reevaluate their discussion about axonal co-transport, as co-transport of *Ptbp2* and *Hnrnpr* mRNA is not directly shown. The authors may want to simply phrase this as localization instead.

Author's response: We followed the reviewer's suggestion and re-phrased the relevant section of the Discussion as following:

"We found that *Ptbp2* associates with *Hnrnpr* mRNA in the cytosol of motoneurons as part of mRNP particles, and that such particles are located in axons for local synthesis of hnRNP R."

Reviewer #3 (Remarks to the Author):

The manuscript entitled 'Ptbp2 modulates axon growth in motoneurons through axonal localization and translation of Hnrnp-r' reports an axon-localized pool of the splicing factor PTBP2 in motor neurons and explores its function in this compartment. The argument for PTBP2 to carry an axonal function is core to the study. Yet, the findings supporting the idea that the neuronal defect and hnrnp phenotype in Ptbp2 KD is due to lack of axonal pool are currently too weak. A series of essential controls and additional experiments are needed to support the conclusions proposed. The authors show that PTBP2 is required for axonal growth in cultured motor neurons and that PTBP2 binds to hnrnp-R mRNA 3'UTR (unclear if this binding is nuclear or also cytoplasmic). They show that in addition to co-localize in the nucleus, hnrnp mRNA and PTBP2 protein also co-localize in axons (although good control for the immunostaining is needed, see specific comments below). They do not provide experiments going beyond a correlation. Nuclear PTBP2 binds to 3'UTR of many mRNAs and regulates a lot of mRNAs coding for RNA-binding proteins, all likely to be important for transport of mRNAs. The loss of hnrnp-R transcripts in axons in the KD could therefore be very indirect. Same indirect correlation is true for the binding of PTBP2 with ribosomes and eIF5A, lacking findings supporting its functional significance. The manuscript is potential very important because it shows that PTBP2 is in the cytoplasm and axons and that in the cytoplasm, it can form a complex with the translation machinery. But is this binding meaningful or is the phenotypes observed an indirect consequence of PTBP2 known role in splicing of essential cytoplasmic proteins important for transport and local translation? Finally, results shown in Figure 7 contradicts the conclusions proposed by the authors.

All together this study needs a lot of additional data to provide a convincing set of evidence supporting the message promoted by the authors. I would recommend rethinking the study in depth; do a careful assessment of the nuclear and cytoplasmic PTBP2 variants and use variants (either natural or synthetic – ie, adding NES) to dissect the relative contribution of nuclear and cytoplasmic proteins in neuronal maturation.

Author's response: We thank the reviewer for the critical evaluation and suggestions for strengthening our study and are excited by the comment that our study potentially is very important. Based on the comments, we have now performed a series of additional experiments and controls to further clarify the function of Ptbp2 for regulating axonal hnRNP R levels. Among these and as outlined below, we have further validated the axonal localization of Ptbp2 and were able to rescue the axon growth defect of *Ptbp2* knockdown motoneurons using a cytoplasmic Ptbp2 variant. The latter finding, together with the rescue effect obtained by re-expressing hnRNP R in *Ptbp2* knockdown motoneurons, further supports the notion that Ptbp2 functions in axon growth are mediated by its cytoplasmic roles rather than its nuclear functions in splicing regulation.

Specific weaknesses:

- How quickly after shPTBP2 lentiviral treatment are the axons showing a defect in culture? The time of transfection/infection is unclear and imaging is done at DIV6. Published mouse mutant studies already showed that PTBP2 is required for neuronal survival during development and axonal integrity. Is this different? Different time of treatment and timelapse studies of neurons after treatment is required.

Author's response: Motoneurons were transduced with lentiviruses directly before plating on DIV 0. We now added this information to the Methods. This was necessary because the expression of the knockdown shRNA and the effects on Ptbp2 downregulation takes several days. In agreement with previous studies, we observed slight reduction of motoneuron survival on DIV 6 (new Fig. 1d). Additionally, to address the reviewer's comment on the time course of axon growth, we measured axon lengths of *Ptbp2* knockdown motoneurons at DIV 2, 4 and 6 (new Supplementary Fig. 1b). We updated the main text as following:

"Following *Ptbp2* depletion, cell survival was mildly reduced on day in vitro (DIV) 6 in agreement with previous studies on cortical *Ptbp2*-deficient neurons^{11,13} (Fig. 1d). Also at this time point, a reduction in axon length upon *Ptbp2* depletion became apparent (Fig. 1e,f and Supplementary Fig. 1b)."

Fig. 1d

Supplementary Fig. 1b

Fig. 1e

Fig. 1f

- There are at least 8 splice variants of PTBP2 in mouse. It is surprising that the Western is detecting only one band in neurons. Is it possible that the antibody is missing a few due to location of its epitope?

Author's response: We thank the reviewer for the comment. We found that there are eight *PTBP2* splice variants in human (<https://www.ncbi.nlm.nih.gov/gene/58155>) but only two *Ptbp2* splice variants in mouse (<https://www.ncbi.nlm.nih.gov/gene/56195>). These two *Ptbp2* splice isoforms (RefSeq: NM_001310711.1 and NM_019550.2) differ only by three base pairs corresponding to one amino acid and, thus, cannot be distinguished at the protein level by Western blotting. Nevertheless, we provided a full Western blot of an uncut nitrocellulose membrane to show that no additional bands are detectable (new Supplementary Fig. 1a). The epitope of our antibody is located in the range 25 – 75 aa of the PTBP2 protein, and is in the N-terminal region.

Supplementary Fig. 1a

- Is the EGFP-Ptpb2 protein transported in axons? Often GFP-tagged proteins show modified transport. Images of GFP fluorescence in the 'rescued' neurons and quantification of GFP puncta in axons in Fig. 1d-g is required. This is a key control.

Author's response: We thank the reviewer for this important suggestion and have now included additional immunofluorescence images in the revised manuscript of the subcellular distribution of EGFP-Ptpb2. We observed that, similar to endogenous Ptpb2, EGFP-Ptpb2 localized in axons. This new dataset has been inserted as Supplementary Fig. 1c.

Supplementary Fig. 1c

- Control of the specificity of immunostaining in neurons is required showing absence of reactivity in neurons carrying Ptpb2 deletion (in Fig. 3d, although nuclear signal is lower in shPtpb2 treated cells, cytoplasmic signal is still quite high). This can be done using CRISPR. If the deletion approach is too demanding, the authors need at least to compare GFP with anti-PTBP2 signals in the EGFP-PTBP2 expressing culture and ascertain co-localization.

Author's response: We addressed the reviewer's request for additional controls of the Ptpb2 antibody specificity in two ways. First, as suggested by the reviewer, we expressed EGFP-Ptpb2 and performed co-immunostaining with anti-Ptpb2 and anti-EGFP antibodies (new Supplementary Fig. 1c). We observed that, in axons, the distribution of Ptpb2 punctae correlated with EGFP punctae. Second, we designed an additional shRNA targeting the Ptpb2 transcript in exon 4 and, with this new knockdown

construct, we also observed a reduced Western blot signal with the Ptbp2 antibody (new Fig. 1b,c). We updated the manuscript as following:

“To further validate the specificity of the antibody, we immunostained motoneurons expressing EGFP-Ptbp2 with anti-Ptbp2 and anti-EGFP (Supplementary Fig. 1c). We observed that, in axons, the distribution of Ptbp2 punctae correlated with EGFP punctae.”

Supplementary Fig. 1c

Fig. 1b

Fig. 1c

- One possibility that would support the antibody staining results is that Ptbp2 mRNAs are transported in axons and locally translated. The authors need to perform a FISH against Ptbp2 mRNA as these RNA detection techniques are now routine.

Author’s response: We appreciate the reviewer’s feedback, however we respectfully disagree. Presence of a FISH signal for Ptbp2 mRNA in axons would not validate the specificity of the Ptbp2 antibody because Ptbp2 might also be transported into axons as a protein. To prove antibody specificity we performed the experiments as outlined in the previous comment.

- The data showing interaction between the hnRNP-R 3’UTR and PTBP2 are very solid. However, is this interaction happening in both nucleus and the axon? This needs to be checked doing IP on axonal extracts. Compartmentalized cultures are now routine in the field.

Author’s response: The reviewer raises the point that the interaction of Ptbp2 with *Hnrnp* mRNA in axons needs to be validated. In our manuscript, we have already provided data that Ptbp2 co-purifies *Hnrnp* mRNA from the soluble nuclear fraction as well as from the cytosolic fraction of motoneurons. In order to strengthen these data, we performed additional Ptbp2 immunoprecipitation on somatodendritic and axonal lysates from compartmentalized cultures of motoneurons (new Fig. 4d-f). We observed that Ptbp2 also associates with *Hnrnp* mRNA in axons and updated the manuscript as following:

“To further demonstrate the association between Ptp2 and *Hnrnpr* mRNA in axons, we cultured motoneurons in microfluidic chambers allowing the separation of axons from the somatodendritic compartment (Fig. 4d,e). Following Ptp2 immunoprecipitation from axonal and somatodendritic lysate, *Hnrnpr* mRNA co-purification was observed for both compartments (Fig. 4f).”

Fig. 4d

Fig. 4e

Fig. 4f

- In Fig 2f, IgG IP control shows a PTBP2 band too. Why is that? What is the reason for the PTBP2 bands to be different between NSC-34 and primary culture? How is the full Western looking (see question on variants above)?

Author’s response: We thank the reviewer for this comment. The signal previously detected in the IgG lane corresponded to background. However, to further clarify this, we have repeated the Ptp2 immunoprecipitation from NSC-34 cells, which now revealed less background in the IgG lane (new Fig. 3f). We also include the full Western blot here:

Fig. 3f

full Western blot:

- Fig 3 shows quite a lot of PTBP2 proteins left in the KD neurons, especially in the cytoplasm.

Author's response: According to our Western blot data in Fig. 1, the shRNA we used (now termed sh1) reduced Ptbp2 protein levels by ~60%, which is reflected by the reduced immunosignal in Fig. 3 (now Fig. 4). We now performed additional Ptbp2 immunostaining as part of other experiments requested and could also detect a clear reduction in the Ptbp2 immunosignal in both nucleus and cytoplasm (new Fig. 1i and k, Fig. 8a) by at least 60%. Thus, the immunostaining results are in line with the reduction detected by Western blotting.

Authors need to show whether rescue can be obtained with a cytoplasmic version of the PTBP2 protein (deletion of NLS and addition of NES element to the GFP construct and imaging of GFP localization as well as hnrnp-R mRNA).

Author's response: We thank the reviewer for this important suggestion, which was also raised by another reviewer. We tested whether Ptbp2-depleted motoneurons can be rescued with a cytoplasmic Ptbp2 variant by preparing a new lentiviral Ptbp2 knockdown construct that co-expresses a GFP-Ptbp2 fusion protein deficient in the NLS. Ptbp2 contains a NLS composed of two basic stretches (KR and KKFK) at its N-terminal end (Pina et al. 2018 Biochemistry, doi: 10.1021/acs.biochem.8b00256). Deletion of these stretches has been shown to abolish the nuclear localization of Ptbp2 (Romanelli et al. 2002 FEBS J, doi: 10.1046/j.1432-1033.2002.02942.x). We found that GFP-Ptbp2-ΔNLS could rescue the effect of *Ptbp2* knockdown on axon growth in motoneurons (new Fig. 2d-j). This also indicates that defects in splicing due to lack of Ptbp2 in the nucleus are not responsible for the axonal phenotype observed in Ptbp2-deficient motoneurons. We updated the manuscript as following:

“To test whether the effect of *Ptbp2* knockdown on axon growth is mediated by the cytosolic function of Ptbp2, we prepared a lentiviral *Ptbp2* knockdown construct that co-expresses an GFP-Ptbp2 NLS deletion mutant (Fig. 2d,e). Ptbp2 contains a NLS composed of two basic stretches (KR and KKFK) at its N-terminal end¹⁸. Deletion of these stretches has been shown to abolish the nuclear localization of Ptbp2¹⁹. We confirmed the cytosolic localization of GFP-Ptbp2-ΔNLS in transduced motoneurons by immunostaining (Fig. 2f). Expression of GFP-Ptbp2-ΔNLS in *Ptbp2* knockdown motoneurons could rescue the defect in axon growth and growth cone size indicating that cytosolic but not nuclear Ptbp2 functions mediate its role in axon growth (Fig. 2g-j).”

Fig. 2d**Fig. 2e****Fig. 2f****Fig. 2g****Fig. 2h****Fig. 2i****Fig. 2j**
- Fig 4 shows drop in proteins in the axons. As the authors show that the number of mRNA is reduced substantially in the KD, the drop in protein is not surprising. This does not mean 'that Ptbp2 is required for the local translation of hnRNP R in axons of motoneurons.' This is a vast over-interpretation of the results.

Author's response: We agree with the reviewer that the reduced axonal hnRNP R synthesis detected by Puro-PLA in *Ptbp2*-knockdown motoneurons might partially be caused by reduced axonal *Hnrnp* mRNA localization. Therefore, we performed additional experiments and measured the level of *Hnrnp* mRNA in the somatodendritic and axonal compartments of control and *Ptbp2*-depleted motoneurons cultured in microfluidic chambers (Fig. 4i in the revised manuscript). Axonal levels of *Hnrnp* mRNA were reduced by ~44% in *Ptbp2* knockdown compared to control motoneurons. This reduction in axonal *Hnrnp* mRNA is less than that observed by Puro-PLA (~64%). Additionally, there was no reduction or a tendency to increased level of *Hnrnp* mRNA levels in the somatodendritic compartment (Fig. 4i in the revised manuscript) or in whole *Ptbp2*-depleted cells (Fig. 5g in the revised manuscript), while we observed a significant reduction of hnRNP R Puro-PLA signal intensity also in the somata of *Ptbp2*-depleted motoneurons. Thus, our data indicate that *Ptbp2* regulates the axonal translation of *Hnrnp* mRNA in addition to its axonal localization. We adjusted the main text and included the

possibility that the *Ptbp2*-dependent *Hnrnpr* mRNA transport might contribute to its effects of local translation in axons as following:

“Additionally, while our Puro-PLA data indicate reduced axonal translation of hnRNP R in *Ptbp2* knockdown motoneurons, we cannot rule out the possibility that the reduced axonal localization of *Hnrnpr* mRNA we observed in *Ptbp2*-deficient axons contributes to this effect. However, according to our qPCR data from compartmentalized motoneurons, the reduction at the *Hnrnpr* mRNA level is less than that observed by hnRNP R Puro-PLA in axons of *Ptbp2*-depleted motoneurons. Additionally, we also observed reduced hnRNP R translation in the somata of *Ptbp2* knockdown motoneurons. Since total *Hnrnpr* mRNA levels are not affected by *Ptbp2* knockdown, this finding supports the notion that *Ptbp2* regulates axonal hnRNP R protein levels by promoting its local translation in addition to modulating its axonal mRNA localization.”

Fig. 4i

Fig. 5g

The observation that mRNA level is not as reduced as protein level can here again be very indirect as ‘sick’ neurons do show overall reduced translation. Only the experiment done later shows that overall polysomes are not generally affected in the KD.

Author’s response: The reviewer raises the important possibility that *Ptbp2* knockdown motoneurons might exhibit a general defect in translation. We tested this possibility by puromycin labelling followed by immunostaining with an anti-puromycin antibody to monitor global protein synthesis. We did not observe any changes of the puromycin immunosignal in the somata, proximal or distal axons of *Ptbp2*-depleted motoneurons compared to controls. This indicates that *Ptbp2* does not have a general function in translation, which also confirms our polysome fractionation results that did not detect any overall translation defects. We included these new puromycin immunostaining data in the new Supplementary Fig. 4a,b and adjusted the main text as following:

“Our Puro-PLA results indicate that Ptbp2 promotes hnRNP R translation in axons. To exclude the possibility that Ptbp2 deficiency induces a general defect in translation, we treated Ptbp2 knockdown and control motoneurons with puromycin followed by immunostaining with an anti-puromycin antibody. We did not observe any changes in the puromycin immunosignal in the somata, proximal or distal axons of Ptbp2 knockdown motoneurons relative to controls indicating that Ptbp2 depletion does not affect overall translation (Supplementary Fig. 4a,b).”

Supplementary Fig. 4a

Supplementary Fig. 4b

- Binding of PTBP2 with ribosomes is very interesting but do not infer a functional requirement of this binding for translation. The authors need to be much more careful in their conclusions.

Author’s response: We thank the reviewer for pointing this out. Our data show that Ptbp2 regulates the association between eIF5A1/2 with ribosomes (Fig. 6i and j in the revised manuscript). Given that eIF5A1/2 is involved in translation elongation (doi: 10.1002/jcb.20658, doi: 10.1016/j.molcel.2017.03.003, doi: 10.1038/nature08034), our findings suggest that Ptbp2 regulates translation through eIF5A1/2. Additionally, we observed that knockdown of Ptbp2 reduces the association of *Hnrnp* mRNA with ribosomes (Fig. 6a in the revised manuscript) showing that Ptbp2 is required for ribosomal recruitment to *Hnrnp*. We now confirmed this observation with two additional experiments. First, we expressed *EGFP* mRNA fused to the 3’ UTR of *Hnrnp* with or without the Ptbp2 binding site and tested the association of these reporters with ribosomes by immunoprecipitation with the Y10b antibody. We observed that deletion of the Ptbp2 binding site reduced ribosome association of the *EGFP* reporter transcript (new Fig. 6b). Second, we prepared an additional *EGFP* reporter mRNA, in which we fused the *EGFP* coding sequence to the Ptbp2 binding site present in the *Hnrnp* 3’ UTR. We found that the association of this reporter with ribosomes was enhanced compared to the control *EGFP* transcript that lacked the Ptbp2 binding domain (new Fig. 6c-d). Thus, Ptbp2 binding is necessary

and sufficient to promote mRNA recruitment to ribosomes. Nevertheless, we rephrased the main text to avoid unintended overinterpretation of this finding. We updated the manuscript as following:

“To substantiate this finding, we transduced motoneurons with the EGFP-FL and Δ PBS *Hnrnpr* 3' UTR lentiviral constructs and assessed their *EGFP* mRNA co-purification following ribosome immunoprecipitation with Y10b. We observed that co-purification of *EGFP* containing the *Hnrnpr* Δ PBS 3' UTR was reduced compared to the FL 3' UTR, further demonstrating that Ptbp2 binding mediates the association of *Hnrnpr* with ribosomes (Fig. 6b). Second, we prepared an additional EGFP reporter mRNA, in which we fused the EGFP coding sequence to the Ptbp2 binding site present in the *Hnrnpr* 3' UTR. We found that the association of this reporter with ribosomes was enhanced compared to the control EGFP transcript (Fig. 6c-e). Thus, Ptbp2 binding is necessary and sufficient to promote mRNA recruitment to ribosomes.”

Fig. 6b

Fig. 6c

Fig. 6d

Fig. 6e

- The fact that PTBP2 KD affects more polysomes than monosomes suggests that it affects cell body translation rather than axonal (normal axons and synapses are richer in monosomes and the Schuman lab showed that monosomes translate mRNAs in these compartments). Ribosomal distribution needs to be measured in control and KD neurons.

Author’s response: We thank the reviewer for raising this point. To investigate the ribosomal distribution, we cultured motoneurons in microfluidic chambers and performed polysome profiling from both the somatodendritic and axonal compartment by sucrose gradient ultracentrifugation. We observed that the *Hnrnpr* transcript was present at higher levels in fractions corresponding to polysomes in axons (new Fig. 5d). Thus, our data suggest that polysomes rather than monosomes mediate axonal *Hnrnpr* translation. We updated the manuscript as following:

“To further validate the axonal translation of *Hnrnpr* mRNA, we cultured motoneurons in microfluidic chambers and performed polysome profiling from both the somatodendritic and axonal compartment by sucrose gradient ultracentrifugation. We observed that, in axons, the *Hnrnpr* transcript was present at higher levels in those fractions corresponding to polysomes (new Fig. 5d). Thus, our data suggest that polysomes rather than monosomes mediate axonal *Hnrnpr* translation.”

Fig. 5d

- The co-precipitation of eIF5A with PTBP2 is expected as PTBP2 is binding to 3'UTR of ribosome-decorated mRNAs. Translation requires folding of the mRNA and direct interaction between cap proteins and 3'UTR complexes.

Author's response: Our co-precipitation data (Fig. 7f in the revised manuscript) show that Ptbp2 interacts with eIF5A1/2 in an RNA-independent manner. If the association was merely due to being present in the same translating mRNP it would be RNA-dependent. Furthermore, eIF5A1/2 is involved in translation elongation rather than initiation (doi: 10.1002/jcb.20658, doi: 10.1016/j.molcel.2017.03.003, doi: 10.1038/nature08034). This speaks against the possibility that Ptbp2 associates with eIF5A1/2 due to circularization of mRNAs during translation initiation. Nevertheless, we thank the reviewer for pointing this out and helping us to clarify our conclusions. We have revised the corresponding part in the Discussion as following:

“Whilst we observed that Ptbp2 associates with translation initiation complexes, we also showed that Ptbp2 binds to eIF5A1/2 in an RNA-independent manner, promoting the association of eIF5A1/2 with ribosomes and of eIF5A1/2 with *Hnrnpr* mRNA. eIF5A1/2 has previously been shown to interact with elongating ribosomes rather than translation initiation complexes^{27,38,39}. This puts forward a model according to which Ptbp2 associates with the 3' UTR of *Hnrnpr* and possibly other mRNAs, and stimulates their translation through modulating their association with translating ribosomes.”

- If the authors are right and PTBP2 phenotype is not due so much to hnrnp-R mRNA reduction but rather caused by lack of translation of this mRNA, the rescue of Ptbp2 KD by overexpression of a hnrnp-R vector is contradicting the conclusion completely as this vector is producing transcripts that should not be properly translated in absence of PTBP2. Figure 7 contradicts completely the model proposed by the authors.

Author's response: Our data show that Ptbp2 interacts with the 3' UTR of *Hnrnp-r*. The hnRNP R rescue construct (Fig. 8 in the revised manuscript) does not contain the *Hnrnp-r* 3' UTR and, thus, is not regulated by Ptbp2.

REVIEWERS' COMMENTS

Reviewer #1 (Remarks to the Author):

The authors addressed all my suggestions thoroughly, including several challenging experiments. Especially valuable is the capacity to rescue the axon growth defect of Ptbp2 knockdown motoneurons using a cytoplasmic Ptbp2 variant, and that knockdown of eIF5A2 leads to no statistically significant difference in Ptbp2 binding to Hnrnpr. Together with other additional and clarifications, this makes the findings and the model convincing and clear, therefore I find the manuscript ready for publication.

Reviewer #2 (Remarks to the Author):

The authors thoroughly and completely addressed all comments, which is strongly appreciated. Particularly new experiments, validating the specificity and quantification of the FISH data, the new quantification of growth cones in Ptbp2 knockdown neurons, the investigation of synaptic markers in Ptbp2 knockdown neurons, and the validation of ribosome interaction using the deleted binding site in the Hnrnpr 3'UTR, substantially improve the impact of this manuscript. Moreover, the data presentation is now much clearer and more informative. Therefore, I strongly suggest publication.